# Gradient Descent as Loss Landscape Navigation: a Normative Framework for Deriving Learning Rules

John J. Vastola[1,2,3]        Samuel J. Gershman[2,3]        Kanaka Rajan [1,3]*

[1]Department of Neurobiology, Harvard Medical School
[2]Department of Psychology and Center for Brain Science, Harvard University
[3]Kempner Institute for the Study of Natural and Artificial Intelligence, Harvard University
`{john_vastola, kanaka_rajan}@hms.harvard.edu, gershman@fas.harvard.edu`

## Abstract

Learning rules—prescriptions for updating model parameters to improve performance—are typically assumed rather than derived. Why do some learning rules work better than others, and under what assumptions can a given rule be considered optimal? We propose a theoretical framework that casts learning rules as policies for navigating (partially observable) loss landscapes, and identifies optimal rules as solutions to an associated optimal control problem. A range of well-known rules emerge naturally within this framework under different assumptions: gradient descent from short-horizon optimization, momentum from longer-horizon planning, natural gradients from accounting for parameter space geometry, non-gradient rules from partial controllability, and adaptive optimizers like Adam from online Bayesian inference of loss landscape shape. We further show that continual learning strategies like weight resetting can be understood as optimal responses to task uncertainty. By unifying these phenomena under a single objective, our framework clarifies the computational structure of learning and offers a principled foundation for designing adaptive algorithms.

## 1 Introduction

A central concern in machine learning is identifying parameters that optimize model performance. Because directly searching for optimal parameters (e.g., via a grid search) is prohibitively costly in the high-dimensional parameter spaces characteristic of models based on artificial neural networks, optimization typically involves seeking iterative improvements in performance rather than directly searching for a global optimum. Procedures for iterative parameter improvement, or *learning rules*, are most commonly some variant of gradient descent [1, 2], with the backpropagation algorithm [3, 4] being a notable example. In biological neural networks, the plausibility of gradient descent is hotly debated [5, 6], and alternative rules that do not follow gradients have been proposed [7, 8].

Variants of gradient descent can largely be classified in terms of the presence or absence of three elements: momentum [9, 10], an adaptive learning rate [11–13], and loss approximation. From a geometric perspective, gradient descent corresponds to moving down the steepest part of the local loss landscape. In this view, momentum helps ensure smooth parameter changes, even when the loss landscape changes abruptly; an adaptive learning rate allows parameters to change more quickly when gradients are steady, or equivalently in regions where the loss landscape is flat; and computing the loss or its gradients approximately, for example over mini-batches, both improves efficiency and may add noise useful for generalization [14–16].

---

*Corresponding author

39th Conference on Neural Information Processing Systems (NeurIPS 2025).

Why prefer one learning rule over another? Is a variant of gradient descent always optimal? If so, which one? If not, by what criteria do we construct or decide on an alternative? These questions are usually answered empirically, with optimizers like Adam [13] popular because they have proven performant in a wide variety of contexts [17–20]. It would be helpful to have a normative framework for answering them in a principled fashion, which, given a set of assumptions, identifies some learning rule as 'optimal'. In this paper, our aim is to provide such a unifying framework.

Our three key insights are as follows. First, one can improve performance by optimizing not just over the next small parameter update, but over the *whole sequence* of future updates; this allows optimization to be less 'myopic' and more 'farsighted'. Second, optimal learning dynamics ought to be sensitive to structure in parameter space. This insight is related to, but slightly more general than, the line of thought that leads to natural gradient descent [21]. Third, one can view the loss landscape as being only *partially* observable, which implies optimal learning ought to depend on *beliefs* about the loss landscape. Assuming partial observability is a useful way to model the fact that the training loss is typically only a proxy for the test loss, which is the true optimization target. We show this idea naturally yields adaptive optimizers like Adam, which update parameters using inferred loss shape.

Our framework is significant for two reasons. First, it makes it easier to generate new learning rules from a principled starting point, and to justify them without empirical guesswork. Second, it helps clarify which features of existing learning rules are essential for performance, and which are incidental. In the following sections, we discuss in more detail how different classes of well-known learning rules—including gradient descent with momentum (Sec. 3), natural gradient descent (Sec. 4), and rules with adaptive learning rates (Sec. 5)—can all be derived from our framework. Finally, Sec. 6 uses our framework to justify recently-identified rules for continual learning [22].

## 2  Mathematical formulation: learning rules as loss landscape navigation

**Gradient descent and Newton's method as optimal single steps through parameter space.**   To motivate our framework, it is useful to observe that gradient descent and Newton's method (a second-order analogue [23]) minimize a certain objective. Given a loss $\mathcal{L}(\boldsymbol{\theta})$ that depends on a parameter vector $\boldsymbol{\theta}$, and a local first- or second-order approximation of it, we would like an update $\Delta\boldsymbol{\theta}$ that decreases the loss as much as possible. Since large enough updates would invalidate our local loss approximation, we want to do this subject to the constraint that $\|\Delta\boldsymbol{\theta}\|$ is not too large. We can do this by minimizing a combination of the loss and a step-size-related regularization term:

$$J(\Delta\boldsymbol{\theta}) = \frac{\|\Delta\boldsymbol{\theta}\|^2}{2\eta} + \mathcal{L}(\boldsymbol{\theta} + \Delta\boldsymbol{\theta}) \approx \begin{cases} \frac{\|\Delta\boldsymbol{\theta}\|^2}{2\eta} + \mathcal{L}(\boldsymbol{\theta}) + [\nabla_{\boldsymbol{\theta}}\mathcal{L}(\boldsymbol{\theta})]^T(\Delta\boldsymbol{\theta}) \\ \frac{\|\Delta\boldsymbol{\theta}\|^2}{2\eta} + \mathcal{L}(\boldsymbol{\theta}) + [\nabla_{\boldsymbol{\theta}}\mathcal{L}(\boldsymbol{\theta})]^T(\Delta\boldsymbol{\theta}) + \frac{1}{2}(\Delta\boldsymbol{\theta})^T \boldsymbol{H}(\boldsymbol{\theta})(\Delta\boldsymbol{\theta}) \end{cases}$$

where $\eta > 0$ is the learning rate (which determines the size of the 'trust region'), and $\boldsymbol{H}(\boldsymbol{\theta})$ is the Hessian of $\mathcal{L}$ at $\boldsymbol{\theta}$. Given the first-order loss approximation, the solution is $\Delta\boldsymbol{\theta} = -\eta\nabla_{\boldsymbol{\theta}}\mathcal{L}(\boldsymbol{\theta})$, i.e., gradient descent with a learning rate $\eta$. Geometrically, this step is down the 'steepest' part of the loss landscape near $\boldsymbol{\theta}$. Given the second-order loss approximation, the solution is $\Delta\boldsymbol{\theta} = -[\frac{1}{\eta}\boldsymbol{I} + \boldsymbol{H}(\boldsymbol{\theta})]^{-1}\nabla_{\boldsymbol{\theta}}\mathcal{L}(\boldsymbol{\theta})$, i.e., gradient descent with a curvature-sensitive *effective* learning rate $\eta_{\text{eff}} := [\frac{1}{\eta}\boldsymbol{I} + \boldsymbol{H}(\boldsymbol{\theta})]^{-1}$. One still takes a step down the steepest part of the loss landscape, but takes a larger step if the landscape is very flat; this avoids the slowdown gradient descent faces near a local minimum, where gradients get smaller. Newton's method technically only corresponds to the large $\eta$ limit, where $\eta_{\text{eff}} = \boldsymbol{H}(\boldsymbol{\theta})^{-1}$; more generally, we obtain a regularized version, which (given a specific Hessian approximation) is sometimes called the Levenberg-Marquardt algorithm [24].

**Learning rules as partially observable loss landscape navigation.**   The single-step view of loss optimization is arguably myopic, since the best short-term improvement of the loss may be suboptimal in the long term; famously, gradient descent finds *local* rather than *global* minima. An obvious way to address this issue is to instead optimize over multiple steps, and ask what *sequence* of steps should be taken in order to minimize the loss (Fig. 1a, b). In going from single- to multi-step optimization, we convert the problem of deciding on parameter updates into a *navigation* problem: what path through the loss landscape minimizes the loss, while simultaneously avoiding overly fast parameter changes?

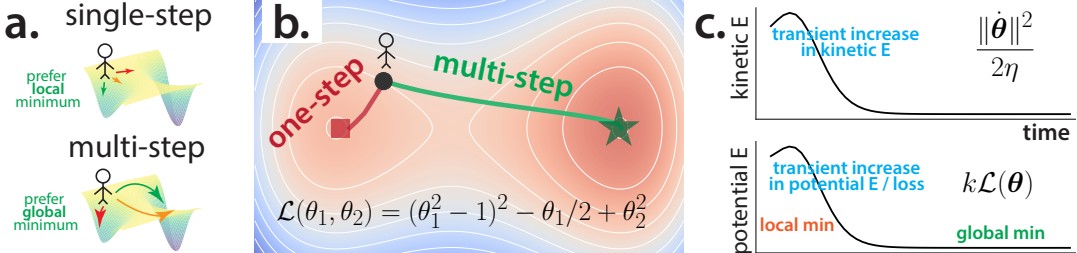

Figure 1: **Basic idea of our framework and a simple example. a.** Single-step approaches (top) optimize over short-term changes to the loss, while a multi-step approach (bottom) optimizes over longer-term changes. **b.** Gradient descent vs multi-step optimization for a double-well loss, with the optimal multi-step trajectory computed by directly minimizing the objective (see Appendix B for details). Note that the multi-step rule converges to the global rather than local minimum. **c.** Values of the kinetic (top) and potential (bottom) terms along the optimal trajectory from (b). The loss/potential does not decrease monotonically, since the learner must first escape a local minimum.

To formalize this, we define a continuous-time optimal control problem [25] over learning trajectories:

$$J(\{\boldsymbol{\theta}_t\}) = \mathbb{E}_{\text{loss landscape belief}} \{ [ \text{ parameter change cost } ] + [ \text{ loss cost } ] \} \tag{1}$$

$$= \mathbb{E}_{\{\hat{\mathcal{L}}_t\}} \left\{ \int_0^\infty \left( \frac{1}{2\eta} [\dot{\boldsymbol{\theta}}_t - \boldsymbol{f}(\boldsymbol{\theta}_t)]^T \boldsymbol{G}(\boldsymbol{\theta}_t) [\dot{\boldsymbol{\theta}}_t - \boldsymbol{f}(\boldsymbol{\theta}_t)] + k\hat{\mathcal{L}}_t(\boldsymbol{\theta}_t) \right) e^{-\gamma t} \, dt \right\}$$

where $\eta > 0$ is the learning rate, $k > 0$ weights the influence of the loss, and $\gamma \geq 0$ is the temporal discounting rate. This objective generalizes the single-step objective (see Appendix A) and defines a continuous-time reinforcement learning [26, 27] and control [25] problem. It penalizes a combination of abrupt parameter changes (through the first term) and high loss (through the second), while discounting temporally distant costs through the factor $e^{-\gamma t}$. This objective also effectively turns learning into a physics problem [28], with these two terms analogous to kinetic and potential energy, and $\boldsymbol{G}$ (the parameter space metric) and $\boldsymbol{f}$ (the 'drift' or 'bias') encoding assumptions about the ambient (parameter space) geometry. Note: the potential/loss term may not decrease monotonically, for example if optimal dynamics involves exiting a local minimum of the loss (Fig. 1c).

**Why partial observability?**    In most practical settings, the learner does not have full knowledge of the loss landscape. We model this by assuming the learner possesses a structured model $\hat{\mathcal{L}}_t$ of $\mathcal{L}$ that can evolve in time, which is informed by past observations (e.g., of gradients and curvature). This partial observability assumption is realistic for three reasons. First, due to the sheer size of parameter space, it is difficult to plan using more than a small part of the loss landscape at any given time. Second, the training loss is generally different from the test loss, but the two are not unrelated; it can be helpful to view the training loss as a *noise-corrupted version* of the test loss. Finally, the loss is usually evaluated on a batch of examples rather than the full training set. While these constraints make it difficult to plan an entire trajectory in one shot, one can instead work iteratively: take a step in the optimal direction, re-estimate your belief given new observations, and repeat.

**Deriving learning rules.**    Our objective (Eq. 1) takes as input **four pieces of data**: a loss function $\mathcal{L}_t$, a parameter space metric $\boldsymbol{G}$, a drift term $\boldsymbol{f}$, and a random variable $\hat{\mathcal{L}}_t$, which models the learner's belief about the loss landscape. This belief can evolve in time, e.g., via online Bayesian inference. Given this data, an **optimal learning trajectory** is a choice of $\{\boldsymbol{\theta}_t\}_{t \in [0,\infty)}$ that minimizes $J$. Given an initial state $\boldsymbol{\theta}_0$ and a desired time step $\Delta t > 0$, an **optimal learning rule** is the difference $\boldsymbol{\theta}_{\Delta t} - \boldsymbol{\theta}_0$ along the optimal trajectory, which is proportional to the initial velocity $\dot{\boldsymbol{\theta}}_0$ when $\Delta t$ is small. This defines a navigation policy and continuous-time analogue of the optimal 'next step'. The naive way to estimate an optimal trajectory is to consider a parameterized family of trajectories, and then directly minimize $J$ with respect to those parameters; Strang et al. [29] use this type of approach to numerically solve classical mechanics problems, and we followed their approach to generate Fig. 1b, c (see Appendix B for details). But there is a powerful alternative approach: framing learning rule optimization in terms of continuous-time learning trajectories allows us to leverage a result from the calculus of variations [28, 30] to identify optimal dynamics with solutions of the *Euler-Lagrange (EL) equations* (see Appendix C). These second-order ODEs are often solvable, and in our case, yield familiar learning rules like gradient descent and Adam under different assumptions. See Appendix D for discussion of technical and conceptual subtleties related to boundary conditions.

# 3 Momentum as a generic consequence of multi-step trajectory optimization

Unlike gradient descent, learning rules with momentum [9, 31] are more 'inertial': instead of following the current gradient, one follows a weighted combination of current and past gradients. Like second-order methods, this change helps speed up movement through flat regions of the loss landscape. In this section, we show that momentum is a generic consequence of multi-step optimization, and derive first- and second-order learning rules with momentum using our framework.

**Momentum as a generic consequence of multi-step optimization.** A straightforward, multi-step generalization of the objective we considered to justify gradient descent is (see Appendix E)

$$J(\{\boldsymbol{\theta}_t\}) = \int_0^\infty \left( \frac{1}{2\eta} \|\dot{\boldsymbol{\theta}}_t\|^2 + k\mathcal{L}(\boldsymbol{\theta}_t) \right) e^{-\gamma t} \, dt \, . \tag{2}$$

This objective is the simplest version of Eq. 1 ($\boldsymbol{G} = \boldsymbol{I}$, $\boldsymbol{f} \equiv \boldsymbol{0}$, and the loss is not approximated). It penalizes only two things: abrupt parameter changes, and the loss. Optimal learning dynamics follow the EL equations, which in this case take the simple form (see Appendix E)

$$\dot{\boldsymbol{\theta}}_t = \boldsymbol{p}_t \qquad\qquad \dot{\boldsymbol{p}}_t = \gamma \boldsymbol{p}_t + \eta k \nabla_{\boldsymbol{\theta}_t} \mathcal{L}(\boldsymbol{\theta}_t) \tag{3}$$

where we defined[2] **momentum** as $\boldsymbol{p}_t := \dot{\boldsymbol{\theta}}_t$. These equations have two interesting consequences. First, we obtain momentum essentially for free, simply from going from one-step to multi-step optimization. Second, the temporal discounting rate $\gamma$ allows one to interpolate between a momentum-based rule and standard gradient descent, since in the 'overdamped' (large $\gamma$) limit these equations become

$$\dot{\boldsymbol{\theta}}_t = -\frac{\eta k}{\gamma} \nabla_{\boldsymbol{\theta}_t} \mathcal{L}(\boldsymbol{\theta}_t) \, . \tag{4}$$

This objective has a straightforward analogy with a mechanical system [28]: a particle with a mass $1/\eta$ moves in a potential $k\mathcal{L}$ and experiences an amount of *friction* proportional to $\gamma$. When 'friction' is sufficiently high, dynamics become non-inertial, and dominated by the shape of the potential/loss.

**Deriving first- and second-order learning rules with and without momentum.** We can derive first- and second-order learning rules by tweaking Eq. 2 to locally approximate $\mathcal{L}$ near $\boldsymbol{\theta}_0$ as

$$\hat{\mathcal{L}}(\boldsymbol{\theta}_t) := \mathcal{L}(\boldsymbol{\theta}_0) + \boldsymbol{g}^T(\boldsymbol{\theta}_t - \boldsymbol{\theta}_0) + \frac{1}{2}(\boldsymbol{\theta}_t - \boldsymbol{\theta}_0)^T \boldsymbol{H}(\boldsymbol{\theta}_t - \boldsymbol{\theta}_0) \tag{5}$$

where $\boldsymbol{g}$ and $\boldsymbol{H}$ are the gradient and Hessian of $\mathcal{L}$ at $\boldsymbol{\theta}_0$. For any $\Delta t \geq 0$, we find (Appendix E) that

$$\boldsymbol{\theta}_{\Delta t} - \boldsymbol{\theta}_0 = -(\boldsymbol{I} - e^{\frac{\gamma}{2}\Delta t - \sqrt{\frac{\gamma^2}{4}\boldsymbol{I} + \eta k \boldsymbol{H}}\Delta t}) \boldsymbol{H}^{-1} \boldsymbol{g} \tag{6}$$

is the optimal update. See Fig. 2a for example $\theta_t$ traces and Fig. 2b for example loss traces. Importantly, we can derive three learning rules from this result. When the step size $\Delta t$ is sufficiently large, we recover the Hessian-conditioned rule (i.e., Newton's method) that 'jumps' straight to the minimum at $-\boldsymbol{H}^{-1}\boldsymbol{g}$. When $\Delta t$ is somewhat smaller than the other characteristic time scales, we either get gradient descent (in the overdamped large $\gamma$ limit) or a *ballistic* learning rule:

$$\boldsymbol{\theta}_{\Delta t} - \boldsymbol{\theta}_0 = \begin{cases} -(\boldsymbol{I} - e^{\frac{\gamma}{2}\Delta t - \sqrt{\frac{\gamma^2}{4}\boldsymbol{I} + \eta k \boldsymbol{H}}\Delta t}) \boldsymbol{H}^{-1} \boldsymbol{g} & \text{Newton's method; } \Delta t \text{ large} \\ -\frac{\eta k}{\gamma} \boldsymbol{g} \Delta t & \text{Gradient descent; } \Delta t \text{ small and } \gamma \gg 1 \\ -\sqrt{\eta k} \boldsymbol{H}^{-1/2} \boldsymbol{g} \Delta t & \text{Ballistic; } \Delta t \text{ small and } \gamma = 0 \end{cases} \cdot \tag{7}$$

The last learning rule is not usually considered, but has a a square-root normalization similar to Adam [13]. We call such a rule **ballistic** because it corresponds to the frictionless limit; it strikes a compromise between following the gradient, as gradient descent does, and jumping straight to the local minimum of the loss, as Newton's method does. Because it depends on the square root of the Hessian, differences in how quickly different parameter directions converge are reduced somewhat relative to gradient descent (Fig. 2c, d). A heuristic implementation of this rule performs well on standard datasets (Fig. 2e, f), which suggests its behavior may be reasonable even for non-quadratic losses. See Appendix B for experiment details.

---

[2]This convention matches machine learning practice. To better parallel physics, we would choose $\boldsymbol{p}_t := \dot{\boldsymbol{\theta}}_t/\eta$.

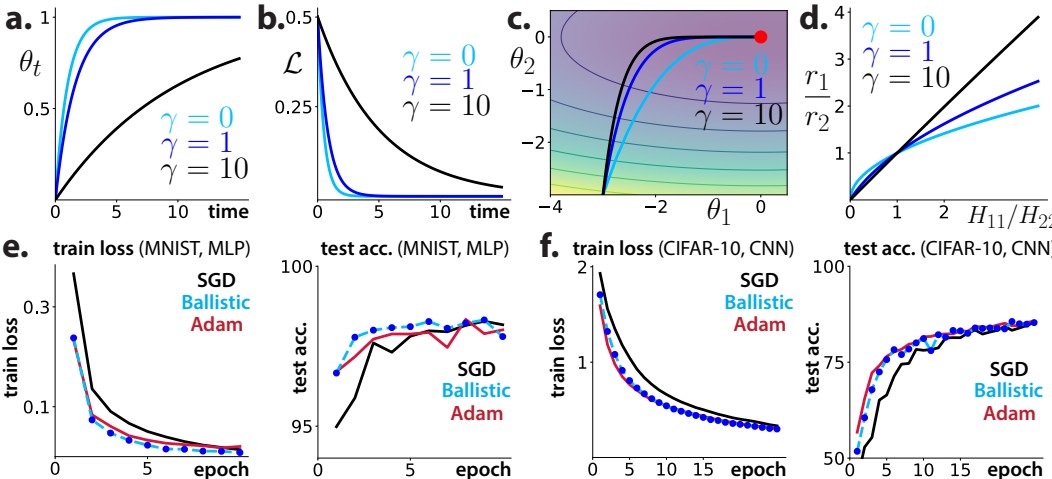

Figure 2: **Effect of modulating temporal discounting rate. a.** Example optimal $\theta_t$ traces for a 1D quadratic loss, assuming different values of the temporal discounting rate ($\gamma = 0, 1, 10$). **b.** Loss over time given the $\theta_t$ from (a), same values of $\gamma$. Note that lower values of $\gamma$ ('longer' planning horizon) produce loss curves that converge more quickly. **c.** Shape of trajectories for different $\gamma$ given a 2D anisotropic loss. In the gradient-descent-like regime ($\gamma \gg 1$), $\theta_2$ converges much more quickly than $\theta_1$ due to the anisotropy. In the ballistic ($\gamma \approx 0$) regime, the difference in convergence rates is not as extreme. **d.** Ratio of convergence rates $r_i := \sqrt{\gamma^2/4 + \eta k H_{ii}} - \gamma/2$ assuming a diagonal Hessian. In the gradient-descent-like regime, directions with four times as much curvature converge 4 times faster; in the ballistic regime, they only converge $\sqrt{4} = 2$ times faster. **e.** An Adam-like implementation of the ballistic ($\gamma \approx 0$) rule was used to train a small multilayer perceptron (MLP) to classify MNIST digits. Left: loss over training, right: test set accuracy over training. **f.** Same as in (e), but for a small convolutional neural network (CNN) trained to classify CIFAR-10 images. The ballistic rule generally performs better than SGD (black), and similarly to or worse than Adam (red).

## 4 Parameter space geometry modulates optimal learning dynamics

**Parameter space geometry modulates distances.** An insight due to Amari [21] is that learning rules ought to be sensitive to the structure of parameter space. If distances in parameter space follow a non-Euclidean metric $G(\theta)$ (e.g., the Fisher information matrix), the objective we used to derive gradient descent must be modified to involve a parameter change penalty $\frac{1}{2\eta}(\Delta\theta)^T G(\theta)(\Delta\theta)$, which causes the optimal first-order (one-step) rule to become $\Delta\theta = -\eta G^{-1}(\theta)\nabla_\theta \mathcal{L}(\theta)$. We can implement Amari's insight in the continuous-time, multi-step setting by adding a metric to Eq. 2:

$$J(\{\theta_t\}) = \int_0^\infty \left( \frac{1}{2\eta}\dot{\theta}_t^T G(\theta_t)\dot{\theta}_t + k\mathcal{L}(\theta_t) \right) e^{-\gamma t}\, dt\,. \tag{8}$$

The corresponding EL equations that describe optimal learning dynamics read (see Appendix F)

$$\dot{\theta}_t = p_t \qquad\qquad \dot{p}_t = \gamma p_t + \eta k G^{-1}(\theta_t)\nabla_{\theta_t}\mathcal{L}(\theta_t) \tag{9}$$

and by an argument analogous to the one in the previous section, we recover three types of rules:

$$\theta_{\Delta t} - \theta_0 = \begin{cases} -(I - e^{\frac{\gamma}{2}\Delta t - \sqrt{\frac{\gamma^2}{4}I + \eta k G^{-1}H}\Delta t})H^{-1}g & \text{Newton's method; } \Delta t \text{ large} \\ -\frac{\eta k}{\gamma}G^{-1}g\Delta t & \text{Nat. gradient; } \Delta t \text{ small and } \gamma \gg 1 \\ -\sqrt{\eta k}\sqrt{G^{-1}H}H^{-1}g\Delta t & \text{Nat. ballistic; } \Delta t \text{ small and } \gamma = 0 \end{cases} \tag{10}$$

These follow from making the replacements $g \to G^{-1}g$ and $H \to G^{-1}H$ in a model that assumes $G$ is locally $\theta$-independent (i.e., here, $G$ refers to $G(\theta_0)$). In addition to recovering *natural gradient descent* (middle), which modifies gradient descent to account for non-Euclidean parameter space geometry [21], we also recover two other learning rules. The first is Newton's method with a metric- and Hessian-dependent learning rate, and the last is a ballistic method which *looks* like Newton's method, but involves $\sqrt{G^{-1}H}H^{-1}$ instead of $H^{-1}$. This method compromises between using $G^{-1}$

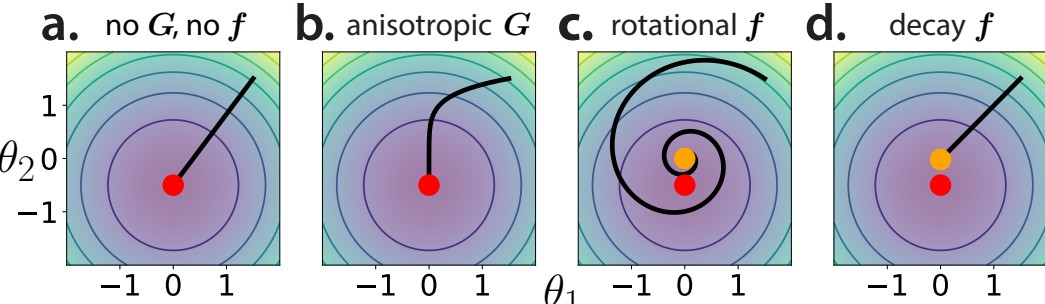

Figure 3: **Parameter space geometry affects optimal learning trajectories. a.** Optimal trajectory through $\theta_1$-$\theta_2$ space for an isotropic quadratic loss, assuming no nontrivial $G$ and $f \equiv 0$. The heatmap and contours show the value of the loss at each $(\theta_1, \theta_2)$ value. Black line: optimal trajectory, red dot: global minimum of loss. Note that, because the loss is isotropic, the optimal trajectory is too. **b.** Same as (a), but given a strongly anisotropic constant metric $G$. Note that the optimal trajectory is no longer the same along each direction, but converges much more quickly along the $\theta_1$ direction. **c.** Same as (a), but given $f$ that corresponds to purely rotational dynamics. Note two differences: it spirals about the origin, and no longer converges to the global minimum of the loss, but to a different point closer to the origin (orange dot). **d.** Same as (a), but given $f$ that corresponds to weight decay. There is no anisotropy, but the trajectory does not converge to the minimum of the loss.

and $H^{-1}$ as preconditioners, or equivalently between moving based on parameter space and loss landscape geometry. While prior work has attempted to combine natural gradients and momentum heuristically [32], our framework shows how these elements arise jointly from a principled objective. As is well-known, using an anisotropic $G$ can add anisotropy to learning dynamics (Fig. 3a, b).

**Natural gradient descent is not second-order optimization in disguise.** It is often argued that natural gradient descent behaves like a second-order method, with the metric $G$ acting as a surrogate for the Hessian $H$. Martens [33] explores this view in detail, and shows that natural gradient descent is sometimes equivalent to a Generalized Gauss-Newton method. But Martens also notes a variety of problems with this view, like the fact that existing theory (e.g., convergence rates) portrays the Hessian as more performant, while empirical work shows natural gradient descent is more performant. Our results suggest that the analogy between second-order methods and natural gradient descent is misguided, and that $G$ and $H$ play fundamentally different roles: $G$ governs the parameter velocity penalty, while $H$ measures loss landscape curvature. Physically, the former defines ambient geometry (as in general relativity [34]), whereas the latter pushes and pulls particles along that geometry. It is also clear when one examines the different roles of $G$ and $H$ in the rules we derived above.

**Optimizing partially controllable parameters.** Parameter space geometry can also influence what kind of learning rule is optimal in a different, less well-appreciated way: suppose an external 'force' pushes on the parameter in a state-dependent fashion, like a 'wind' that determines which directions are more or less difficult to travel in. This feature can arise from partial controllability; if our control of a parameter is partial, e.g., $\dot{\theta}_t = u_t + f(\theta_t)$, controlling the parameter via $u_t$ involves not just moving $\theta_t$ in a direction that improves the loss, but also *fighting against* the 'default' dynamics $f(\theta)$. Here, we show that this feature can produce *non-gradient* rules. If we add a drift term $f$ to Eq. 8,

$$J(\{\theta_t\}) = \int_0^\infty \left( \frac{1}{2\eta}[\dot{\theta}_t - f(\theta_t)]^T G(\theta_t)[\dot{\theta}_t - f(\theta_t)] + k\mathcal{L}(\theta_t) \right) e^{-\gamma t} \, dt . \qquad (11)$$

The presence of $G$ and $f$ can greatly complexify the EL equations, especially if they are state-dependent. Consider the effect of the drift $f$ by itself (i.e., assume $G = I$; see Appendix G):

$$\ddot{\theta}_t - [\gamma I + J(\theta_t) - J(\theta_t)^T] \, \dot{\theta}_t = [J(\theta_t)^T - \gamma I]f(\theta_t) + \eta k \nabla_\theta \mathcal{L}(\theta_t)$$

where $J(\theta_t)$ is the Jacobian of $f$ at $\theta_t$. One can interpret $J$ as contributing to dynamics in two ways: it contributes an effective 'potential-like' term $[J(\theta_t)^T - \gamma I]f(\theta_t)$, or equivalently an effective loss; and it produces an effective state-dependent discounting factor $\gamma_{\text{eff}}(\theta_t) := \gamma I + J(\theta_t) - J(\theta_t)^T$. Note that this reduces to the usual discounting factor if $J(\theta)^T = J(\theta)$ for all $\theta$, which is true if and only if $f$ is the gradient of some function.

In Appendix G, we consider two examples in more detail: the case where $\boldsymbol{f}$ corresponds to rotational dynamics (Fig. 3c), and the case where $\boldsymbol{f}$ corresponds to weight decay (Fig. 3d). The former case is 'nonconservative' (e.g., $\boldsymbol{J}(\boldsymbol{\theta}) \neq \boldsymbol{J}(\boldsymbol{\theta})^T$) but the second is not. Interestingly, rotational dynamics can make 'spiraling' trajectories optimal, and both choices of $\boldsymbol{f}$ make optimal trajectories converge to a point different from the global minimum of the loss.

**Optimal learning dynamics are generally non-gradient.** One significant consequence of including a drift term $\boldsymbol{f}$ is that it can cause the optimal learning rule to involve *non-gradient dynamics*, which cannot be described as gradient descent (with or without momentum) down any objective. In our $\boldsymbol{G} = \boldsymbol{I}$ example, this happens if and only if $\boldsymbol{f}$ itself is not the gradient of any function (or equivalently, if $\boldsymbol{J}(\boldsymbol{\theta})$ is not symmetric for all $\boldsymbol{\theta}$). In terms of a Helmholtz decomposition [35] $\boldsymbol{f}(\boldsymbol{\theta}) = \nabla_{\boldsymbol{\theta}} V(\boldsymbol{\theta}) + \boldsymbol{R}(\boldsymbol{\theta})$, where $V$ is some non-unique 'potential' function and $\boldsymbol{R}$ is divergence-free (i.e., $\nabla_{\boldsymbol{\theta}} \cdot \boldsymbol{R} = 0$), $\boldsymbol{R}$ is the interesting component of $\boldsymbol{f}$. See Appendix G for more discussion of this point. This interpretation suggests a concrete diagnostic: when learning rules involve update components that do not point downhill, such as decay terms or rotational drift, they may be optimal responses to implicit background dynamics.

# 5   Adaptive optimizers arise from beliefs about how gradients evolve in time

Adaptive optimizers like Adam [13] and RMSprop [12] use a learning rate that depends on the variance of recent gradients. If recent gradients were consistent, learn quickly; if not, make smaller updates. Doing this is known to work well in practice, with Adam variants outperforming most other known optimizers [17–20] in typical settings, but it is unclear why, or under what assumptions Adam is optimal. In this section, we show that an Adam-like strategy is optimal under a certain Bayesian model of how the local loss landscape shape evolves in time. More generally, we show adaptive optimizers arise from using past observations of landscape shape to estimate current and future shape.

**Modeling uncertainty in the time evolution of the local loss landscape.** One strategy for navigating a partially observable loss landscape is to maintain running estimates of quantities that characterize its local shape, like the gradient and curvature. Motivated by this idea, assume that the learner maintains a local model of loss landscape shape near their current location $\boldsymbol{\theta}_0$, i.e.,

$$\mathbb{E}[\hat{\mathcal{L}}(\boldsymbol{\theta}_t)] = \mathcal{L}(\boldsymbol{\theta}_0) + \boldsymbol{m}_t^T(\boldsymbol{\theta}_t - \boldsymbol{\theta}_0) + \frac{\kappa}{2}(\boldsymbol{\theta}_t - \boldsymbol{\theta}_0)^T \boldsymbol{V}_t(\boldsymbol{\theta}_t - \boldsymbol{\theta}_0) \tag{12}$$

where the gradient estimate $\boldsymbol{m}_t$ and Hessian estimate $\kappa \boldsymbol{V}_t$ are assumed to be time-varying. Here, $\kappa > 0$ is a fixed (known) scaling factor. Since $\boldsymbol{m}_t$ and $\boldsymbol{V}_t$ represent *average* values of gradients and curvature, respectively, they are not directly observable—but the learner can estimate them via their assumed link to observable gradients $\boldsymbol{g}_t$. Assume that the learner has a Bayesian model of how local landscape geometry evolves with two components: an *observation model*, which connects $\boldsymbol{g}_t$ to $\boldsymbol{m}_t$ and $\boldsymbol{V}_t$; and a *prior* belief about how landscape geometry evolves.

Motivated by a simple model of gradient drift and diffusion in a quadratic loss (see Appendix H), assume that $\boldsymbol{g}_t \sim \mathcal{N}(\boldsymbol{m}_t, \boldsymbol{V}_t/\Delta t)$, which implies that $\mathbb{E}[\boldsymbol{g}_t] = \boldsymbol{m}_t$ and $\mathrm{Cov}(\boldsymbol{g}_t)\Delta t = \boldsymbol{V}_t$. To make inference simpler, we will use a crude approximation[3] associated with method-of-moment-based strategies, and assume $\boldsymbol{g}_t$ and $(\boldsymbol{g}_t - \boldsymbol{m}_t)(\boldsymbol{g}_t - \boldsymbol{m}_t)^T$ provide *independent* observations of $\boldsymbol{m}_t$ and $\boldsymbol{V}_t$, i.e., $\boldsymbol{g}_t \sim \mathcal{N}(\boldsymbol{m}_t, (\sigma_1^2/\Delta t)\boldsymbol{I})$ and $(\boldsymbol{g}_t - \boldsymbol{m}_t)(\boldsymbol{g}_t - \boldsymbol{m}_t)^T \sim \mathcal{N}(\boldsymbol{V}_t, (\sigma_2^2/\Delta t)\boldsymbol{I})$. Here, $\sigma_1^2 > 0$ and $\sigma_2^2 > 0$ are 'observation noise' parameters.

One reasonable prior, which assumes that both parameters tend towards zero and become more uncertain in the absence of observations, is an *Ornstein-Uhlenbeck prior*. Assume

$$\dot{\boldsymbol{m}}_t = -\alpha_1 \boldsymbol{m}_t + \xi_1 \boldsymbol{\eta}_{1t} \qquad\qquad \dot{\boldsymbol{V}}_t = -\alpha_2 \boldsymbol{V}_t + \xi_2 \boldsymbol{\eta}_{2t} \tag{13}$$

where $\alpha_1 \geq 0$ and $\alpha_2 \geq 0$ parameterize decay, $\xi_1 > 0$ and $\xi_2 > 0$ parameterize the uncertainty growth rate, and $\boldsymbol{\eta}_{1t}$ and $\boldsymbol{\eta}_{2t}$ are Gaussian white noise terms. This means $\boldsymbol{m}_{t+1} \sim \mathcal{N}(\boldsymbol{m}_t - \alpha_1 \boldsymbol{m}_t \Delta t, \xi_1^2 \Delta t)$ and $\boldsymbol{V}_{t+1} \sim \mathcal{N}(\boldsymbol{V}_t - \alpha_2 \boldsymbol{V}_t \Delta t, \xi_2^2 \Delta t)$.

---

[3]See Appendix H for a discussion of why this is reasonable, and for a description of the somewhat more complicated rule one gets if one does not assume this.

**Time-evolving beliefs yield adaptive optimal learning rule.** Assume an objective that prioritizes a mix of small loss, smooth parameter changes, and good inference (implemented via a $\log p$ term):

$$J(\{\boldsymbol{\theta}_t\}) = \lim_{\Delta t \to 0} \int_0^\infty \left( \frac{\|\dot{\boldsymbol{\theta}}_t\|^2}{2\eta} - \frac{\log p(\boldsymbol{m}_{t+1}, \boldsymbol{V}_{t+1}|\boldsymbol{g}_t, \boldsymbol{m}_t, \boldsymbol{V}_t)}{\Delta t} + k\mathbb{E}[\hat{\mathcal{L}}(\boldsymbol{\theta}_t)] \right) e^{-\gamma t} dt$$

where $p(\boldsymbol{m}_{t+1}, \boldsymbol{V}_{t+1}|\boldsymbol{g}_t, \boldsymbol{m}_t, \boldsymbol{V}_t)$ is the learner's posterior belief about local landscape shape dynamics. This objective has a well-defined $\Delta t \to 0$ limit (see Appendix H) with EL equations

$$\ddot{\boldsymbol{\theta}}_t - \gamma \dot{\boldsymbol{\theta}}_t = \eta k \left[ \boldsymbol{m}_t + \kappa \boldsymbol{V}_t (\boldsymbol{\theta}_t - \boldsymbol{\theta}_0) \right]$$

$$\ddot{\boldsymbol{m}}_t - \gamma(\dot{\boldsymbol{m}}_t + \alpha_1 \boldsymbol{m}_t) = \alpha_1^2 \boldsymbol{m}_t + \xi_1^2 \left[ k(\boldsymbol{\theta}_t - \boldsymbol{\theta}_0) + \frac{1}{\sigma_1^2}(\boldsymbol{m}_t - \boldsymbol{g}_t) \right]$$

$$\ddot{\boldsymbol{V}}_t - \gamma(\dot{\boldsymbol{V}}_t + \alpha_2 \boldsymbol{V}_t) = \alpha_2^2 \boldsymbol{V}_t + \xi_2^2 \left[ k\frac{\kappa}{2}(\boldsymbol{\theta}_t - \boldsymbol{\theta}_0)(\boldsymbol{\theta}_t - \boldsymbol{\theta}_0)^T + \frac{1}{\sigma_2^2}[\boldsymbol{V}_t - (\boldsymbol{g}_t - \boldsymbol{m}_t)(\boldsymbol{g}_t - \boldsymbol{m}_t)^T] \right] .$$

These equations are somewhat complicated, but considerably simplify if one makes assumptions about the relative sizes of parameters. If we assume that parameter changes are ballistic ($\eta \gg \gamma$) but landscape beliefs change somewhat more slowly ($\xi_1^2, \xi_2^2 \ll \gamma$),

$$\ddot{\boldsymbol{\theta}}_t = \eta k \left[ \boldsymbol{m}_t + \kappa \boldsymbol{V}_t (\boldsymbol{\theta}_t - \boldsymbol{\theta}_0) \right]$$

$$\dot{\boldsymbol{m}}_t = -\alpha_1 \boldsymbol{m}_t - \frac{\xi_1^2}{\gamma} \left[ k(\boldsymbol{\theta}_t - \boldsymbol{\theta}_0) + \frac{1}{\sigma_1^2}(\boldsymbol{m}_t - \boldsymbol{g}_t) \right]$$

$$\dot{\boldsymbol{V}}_t = -\alpha_2 \boldsymbol{V}_t - \frac{\xi_2^2}{\gamma} \left[ k\frac{\kappa}{2}(\boldsymbol{\theta}_t - \boldsymbol{\theta}_0)(\boldsymbol{\theta}_t - \boldsymbol{\theta}_0)^T + \frac{1}{\sigma_2^2}[\boldsymbol{V}_t - (\boldsymbol{g}_t - \boldsymbol{m}_t)(\boldsymbol{g}_t - \boldsymbol{m}_t)^T] \right] .$$

To good approximation, this means that the optimal learning rule has

$$\boldsymbol{\theta}_{t+\Delta t} - \boldsymbol{\theta}_t = -\sqrt{\eta k} \boldsymbol{V}_t^{-1/2} \boldsymbol{m}_t \Delta t . \tag{14}$$

Note that the effective learning rate goes like $\boldsymbol{V}_t^{-1/2}$ rather than $\boldsymbol{V}_t^{-1}$. Our framework identifies the square root as a consequence of assuming highly inertial parameter changes, but gradient-descent-like landscape shape estimate dynamics. Moreover, contrary to other ideas about Adam [36], the square root is a feature rather than a bug, and need not be 'fixed'. Other features of Adam, like estimating $\boldsymbol{V}_t$ using the uncentered averages of $\boldsymbol{g}_t \boldsymbol{g}_t^T$ and approximating $\boldsymbol{V}_t$ as diagonal, appear to be approximations that improve efficiency and scalability, as is usually believed.

## 6 Noisy continual learning strategies reflect parsimony and task uncertainty

In continual learning settings, agents must balance their ability to learn new tasks with their ability to retain information about previous tasks [37, 38]. Poorly performing agents exhibit catastrophic forgetting, but even in the absence of such forgetting, many algorithms exhibit a progressive *loss in plasticity* as an ever larger number of tasks are learned [22]. Dohare et al. observed that periodically resetting weights that do not strongly respond to task gradients empirically seems to ameliorate this issue. Others have observed that injecting noise in different ways, like via randomly perturbing little-used weights, also helps address this issue [39]. Thus far, it has been somewhat unclear why any of these strategies ought to work, and to what extent various details (e.g., how noise is injected) matter. Our framework provides qualitative guidance here: the reason different noise injection strategies empirically work is that injecting noise *at all* is more important than *how* that noise is injected.

**Modeling weight-uncertainty-sensitive learning dynamics.** Assume that the learner uses distributional estimates, rather than point estimates, of the model parameters $\boldsymbol{\theta}$. For simplicity, assume each $\theta_i$ is associated with a normal distribution $\mathcal{N}(\mu_i, v_i)$, where $\mu_i$ denotes the mean estimate of $\theta_i$, and $v_i$ denotes the posterior variance. Instead of penalizing abrupt weight changes, in this setting it makes more sense to penalize abrupt changes in weight *distribution*:

$$J(\{\boldsymbol{\theta}_t\}) = \lim_{\Delta t \to 0} \int_0^\infty \left[ \frac{D_{KL}(p(\boldsymbol{\theta}|\boldsymbol{\mu}_{t+1}, \boldsymbol{v}_{t+1})\|p(\boldsymbol{\theta}|\boldsymbol{\mu}_t, \boldsymbol{v}_t))}{\eta(\Delta t)^2} - \mathcal{H}(p(\boldsymbol{\theta}|\boldsymbol{\mu}_t, \boldsymbol{v}_t)) + k\mathcal{L}(\boldsymbol{\mu}_t, \boldsymbol{v}_t) \right] e^{-\gamma t} dt$$

where we have also included an entropy term to explicitly penalize 'model complexity'. When written more explicitly, this objective reduces to our standard form (Eq. 1) with a nontrivial metric $\boldsymbol{G}$ (see

Appendix I). Given a local (quadratic) loss approximation with gradient $\boldsymbol{g}$ and Hessian $\boldsymbol{H}$, the EL equations read

$$\ddot{\mu}_i - \left[\frac{\dot{v}_i}{v_i} + \gamma\right]\dot{\mu}_i = \eta k v_i \left[g_i + \sum_j H_{ij}(\mu_j - \mu_{j0})\right] \tag{15}$$

$$\ddot{v}_i - \gamma\dot{v}_i = 2\eta\left[\frac{k}{2}H_{ii}v_i^2 - \frac{1}{2}v_i\right] + \frac{\dot{v}_i^2}{v_i} - \dot{\mu}_i^2 \; . \tag{16}$$

The equation for $\mu_i$ (Eq. 15) has an interesting feature: it involves a *variance-dependent* effective discounting rate $\gamma_{\text{eff}} := \gamma + \frac{\dot{v}_i}{v_i}$ and effective learning rate $\eta_{\text{eff}} := \eta v_i$. One can interpret the former as saying that planning should become more short-term when weight variances are changing quickly (i.e., $\dot{v}_i/v_i$ is high), and that learning should speed up when uncertainty (i.e., $v_i$) is high.

**Variance reflects both parsimony and task-driven instability.** The equation for $v_i$ (Eq. 16) specifies how variance ought to increase or decrease along the optimal learning trajectory. The two terms on the right-hand side tell us how this happens: variance *decreases* if the nearby loss landscape is very curved (intuitively, the nearest minimum is easier to find); the entropic term incentivizes *increases* in variance, in order to favor the 'simplest' model associated with a given loss value; and the final term affects variance in a more subtle way. It increases variance if the mean is changing more quickly than the variance, something that can happen in continual learning settings when there is a transition between tasks. If the variance is changing quickly but the mean is not changing, it decreases variance somewhat. See Appendix I for more discussion.

The entropic term enforces behavior analogous to the weight resetting of Dohare et al. [22]: when the loss is not particularly sensitive to a given weight's value (i.e., $H_{ii}$ is small), uncertainty about that weight should increase, with one possible mechanism being a reset. The last term on the right-hand side, which compares the speed of mean and variance changes, appears sensitive to task uncertainty. Together, these two effects—one tied to parsimony, the other to volatility—help explain the behavior of continual learning algorithms that inject noise or reset unused weights, like Dohare et al.'s method.

## 7 Discussion

We proposed a unifying framework that treats learning rules as solutions to an optimal control problem. By varying assumptions about geometry, planning horizon, and uncertainty, it recovers a wide range of familiar rules—including gradient descent, momentum, natural gradient descent, Adam, and continual learning strategies—from a single objective. This framework rests on three core ideas: learning unfolds over multiple steps, not one; parameter space can have nontrivial geometry and dynamics; and the loss landscape may only be partially observable, in which case it must be inferred. Our framework not only recovers these components individually, but also shows how they can be combined, yielding principled algorithms that integrate elements like momentum and natural gradients. It also suggests a shift in how learning rules should be evaluated: rather than focusing only on convergence rates or optimization guarantees, we should consider what assumptions a rule implicitly encodes and what problem it is actually solving (e.g., one related to generalization). This perspective is especially relevant for adaptive optimizers like Adam, which our framework portrays as optimizing an inferred test loss based on a specific Bayesian model of loss landscape shape dynamics. In many deep learning settings, performant optimizers ought to get things slightly 'wrong' to generalize [14, 40–42].

**Connection to physics.** Our optimal control formulation of learning dynamics can be precisely mapped to a classical mechanics problem, which may allow physics tools to be adapted to study learning algorithms. For example, Noether's theorem [43] implies that (quasi-) symmetries of objectives like Eq. 1 imply conserved quantities along optimal trajectories. Analogous ideas linking the consequences of symmetry to noisy recurrent dynamics [44], learning dynamics [45], and optimal dimensionality reduction [46] have already begun to be explored.

**Biological relevance.** Neural circuits in the brain are thought to implement learning rules—such as Hebbian plasticity, homeostatic mechanisms, or spike-timing-dependent plasticity—that often lack a clear gradient-descent interpretation [7, 8, 47]. These rules may include time-asymmetric

or rotational dynamics, operate under metabolic or architectural constraints, or adjust weights in response to activity thresholds rather than loss functions. Our framework suggests that such rules may still be optimal under constraints imposed by biological dynamics, such as synaptic decay, intrinsic drift, or limited access to global error signals [5, 48].

By treating learning as constrained control in a partially observable environment, our framework offers a normative lens on how non-gradient rules can potentially emerge as efficient strategies in biologically realistic regimes. This perspective is analogous to and consistent with observations that complex behavioral strategies can emerge from a mix of simple rewards and naturalistic constraints (e.g., partial observability), for example in foraging tasks [49].

**Related work.**    Several recent efforts aimed to unify learning rules under broader frameworks. Khan and Rue [50] portray various optimizers as natural gradient descent with respect to a fairly general objective, and Shoji et al. [51] similarly observe that many learning rules can be viewed as instances of natural gradient descent. However, both works take the idea that natural gradient descent is optimal for granted, and do not incorporate multi-step planning, belief updating, or partial observability.

Our framework has parallels with earlier foundational work by Wibisono et al. [52] which relates accelerated optimization methods to a continuous-time variational objective. However, there are also important qualitative differences between their framework and ours, with the most important being that their objective cannot be interpreted as a sum of costs, unlike ours. Our objective contains two terms—a quadratic parameter velocity penalty (or 'kinetic energy'), and a loss term (or 'potential energy')—which are *added* together. In their objective, as in classical mechanics, the potential term is *subtracted* from the kinetic term. While this sign difference means that their approach is more directly related to classical mechanics, it also means that it is more distantly related to optimal control. A practical benefit of our sign choice is that optimal trajectories *minimize* the objective, rather than make it stationary in general.

Concurrent work by Orvieto and Gower [53] also proposes a view of Adam related to loss landscape shape inference, and also formalizes it in terms of an objective with a $-\log p$ term, but our picture and theirs differ in certain details. Perhaps the most important is that we link the appearance of the square root of the Hessian to operating in the 'ballistic' regime, or equivalently longer-term planning.

Our work is related in spirit to recent efforts to unify training objectives in deep learning, such as the framework proposed by Alshammari et al. [54].

**Limitations.**    We consider a setting in which our notion of 'optimal' does not factor in concerns which often affect optimization in practice, like memory requirements, computational simplicity, and efficiency. Relatedly, even if a rule is identified as optimal given our framework, it is unclear how useful it may be in practice, especially if it requires potentially expensive matrix operations. On the other hand, given that our formulation is in terms of an objective function, it may be possible to penalize things like efficiency explicitly in order to partially address this issue.

Finally, we do not claim to be able to explain all phenomena related to learning dynamics or learning rules; our goal in this work is merely to propose a useful framework for thinking about why features of well-known learning rules (like momentum) might be useful. The most important consequence of our framework is that, given an assumed objective (e.g., with a certain amount of temporal discounting, and a particular model of landscape shape belief updating), any two learning rules can be compared, and one or more learning rules can be shown to be optimal. Whether and how to link this framework to specific empirical circumstances is a different question which we expect to be more difficult.

## Acknowledgments and Disclosure of Funding

SJG was funded by the Kempner Institute for the Study of Natural and Artificial Intelligence, and a Polymath Award from Schmidt Sciences. KR was funded by the NIH (RF1DA056403, U01NS136507), James S. McDonnell Foundation (220020466), Simons Foundation (Pilot Extension-00003332-02), McKnight Endowment Fund, CIFAR Azrieli Global Scholar Program, NSF (2046583), a Harvard Medical School Neurobiology Lefler Small Grant Award, and a Harvard Medical School Dean's Innovation Award. This work has been made possible in part by a gift from the Chan Zuckerberg Initiative Foundation to establish the Kempner Institute for the Study of Natural and Artificial Intelligence at Harvard University.

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

# A Motivating our continuous-time objective function

In this appendix, we describe in detail how the most general form of our objective function (Eq. 1) can be viewed as a direct generalization of the objective we used to justify gradient descent (see Sec. 2).

## A.1 From a single-step objective to a multi-step objective

Recall that the objective we used to justify gradient descent (and Newton's method) had the form

$$J(\Delta\boldsymbol{\theta}) = \frac{\|\Delta\boldsymbol{\theta}\|^2}{2\eta} + \mathcal{L}(\boldsymbol{\theta} + \Delta\boldsymbol{\theta}) , \tag{17}$$

where we have neglected to Taylor expand $\mathcal{L}$ to keep our objective more general. Note that $J$ involves two terms: one which penalizes large parameter changes, and another which penalizes high values of the loss. The former term is especially necessary if we consider a local approximation (e.g., an expansion in powers of $\Delta\boldsymbol{\theta}$) of $\mathcal{L}$, since the approximation may no longer be valid if we consider sufficiently large steps.

We would like to go from this single-step objective to an analogous multi-step objective. The most obvious way to do this is to define the objective over a sum of $K$ terms, each of which involves a parameter change penalty and a loss term:

$$J_{multi}(\Delta\boldsymbol{\theta}_0, \Delta\boldsymbol{\theta}_1, ..., \Delta\boldsymbol{\theta}_{K-1}) := \sum_{t=0}^{K-1} \frac{1}{2\eta}\|\Delta\boldsymbol{\theta}_t\|^2 + \mathcal{L}(\boldsymbol{\theta}_t) . \tag{18}$$

Note the philosophy of including the loss at each step: it implies that we would like a path through parameter space that involves decreases to the loss at each step, rather than just at the end. It is also possible to penalize the loss only at the end, but the resulting learning rules would look somewhat different than, e.g., variants of gradient descent.

## A.2 From discrete time to continuous time

Although we could directly study the optimization of Eq. 18, we can instead exploit the fact that objectives like this tend to be easier to analyze in continuous time, since determining the optimal sequence $\Delta\boldsymbol{\theta}_0, \Delta\boldsymbol{\theta}_1, ...$ becomes a well-studied calculus of variations [28, 30] problem. If each step takes an amount of 'time' $\Delta t$, the cost of each step is scaled to be proportional to $\Delta t$, and we adjust $\eta \to \eta(\Delta t)^2$ for dimensional reasons, we obtain a continuous-time objective that directly generalizes Eq. 18:

$$J(\{\boldsymbol{\theta}_t\}) = \sum_{t=0}^{K-1} \left[ \frac{1}{2\eta} \left\| \frac{\Delta\boldsymbol{\theta}_t}{\Delta t} \right\|^2 + \mathcal{L}(\boldsymbol{\theta}_t) \right] \Delta t \xrightarrow{\Delta t \to 0} \int_0^T \frac{1}{2\eta}\|\dot{\boldsymbol{\theta}}_t\|^2 + \mathcal{L}(\boldsymbol{\theta}_t)\, dt \tag{19}$$

where $T$ is the total optimization 'time'. Note that our optimization problem now looks like an optimal control problem (with a control cost $\propto 1/\eta$ and state cost $\mathcal{L}$) or classical mechanics problem (where a particle with mass $m := 1/\eta$ moves in a potential determined by $\mathcal{L}$).

To make it slightly clearer how our solution depends on $\mathcal{L}$, and to more explicitly control the influence of the loss, we will add a prefactor $k > 0$ (which has units of inverse time) in front of it:

$$J(\{\boldsymbol{\theta}_t\}) = \int_0^T \frac{1}{2\eta}\|\dot{\boldsymbol{\theta}}_t\|^2 + k\mathcal{L}(\boldsymbol{\theta}_t)\, dt . \tag{20}$$

This change does not meaningfully affect our optimization problem; it only amounts to a change in the units of $J$.

## A.3 Incorporating temporal discounting

Our objective (Eq. 21) now has the form of an optimal control problem. Motivated by this observation, we incorporate a temporal discounting factor $e^{-\gamma t}$, which causes the learner to overemphasize rewards and costs that are nearer in time:

$$J(\{\boldsymbol{\theta}_t\}) = \int_0^T \left[ \frac{1}{2\eta}\|\dot{\boldsymbol{\theta}}_t\|^2 + k\mathcal{L}(\boldsymbol{\theta}_t) \right] e^{-\gamma t}\, dt . \tag{21}$$

We could have included a more general discounting factor, as some prior work does [52]. We restrict ourselves to exponential discounting because it is a canonical choice [26, 27] and yields relatively simple EL equations. As an aside, allowing a more general discounting factor would allow us to derive learning rules that involve Nesterov momentum rather than just standard (Polyak) momentum.

Since we are not generally interested in any specific learning time $T$, and since taking the $T \to \infty$ substantially simplifies some EL-equation-related math, we will consider the $T \to \infty$ version of the objective in everything that follows:

$$J(\{\boldsymbol{\theta}_t\}) = \int_0^\infty \left[ \frac{1}{2\eta} \|\dot{\boldsymbol{\theta}}_t\|^2 + k\mathcal{L}(\boldsymbol{\theta}_t) \right] e^{-\gamma t} \, dt \, . \tag{22}$$

This objective is *infinitely* forward-looking, in the sense that trajectories which optimize it depend on considering $\boldsymbol{\theta}_t$ at arbitrary distant future times.

### A.4  Incorporating parameter space geometry

Our original single-step objective (Eq. 17) implicitly assumes that parameter space is Euclidean, or at least that a Euclidean distance metric is most appropriate for penalizing large parameter changes. As researchers like Amari [21] have observed, this is not necessarily true. More generally, we might want to penalize distances according to a less trivial metric like the Fisher information metric.

We can account for this fact by modifying Eq. 17 to involve a metric $\boldsymbol{G}$:

$$J(\Delta\boldsymbol{\theta}) = \frac{1}{2\eta} (\Delta\boldsymbol{\theta})^T \boldsymbol{G}(\boldsymbol{\theta})(\Delta\boldsymbol{\theta}) + \mathcal{L}(\boldsymbol{\theta} + \Delta\boldsymbol{\theta}) \, . \tag{23}$$

Note that, for all $\boldsymbol{\theta}$, we will assume $\boldsymbol{G}(\boldsymbol{\theta})$ is symmetric and positive definite (and hence invertible). To account for partial controllability (see Sec. 4), we could make the similarly minor change of penalizing not the size of $\Delta\boldsymbol{\theta}$, but the size of $\Delta\boldsymbol{\theta} - \boldsymbol{f}(\boldsymbol{\theta})$ (i.e., the size of deviations from the 'default' dynamics determined by the drift function $\boldsymbol{f}$):

$$J(\Delta\boldsymbol{\theta}) = \frac{1}{2\eta} [\Delta\boldsymbol{\theta} - \boldsymbol{f}(\boldsymbol{\theta})]^T \boldsymbol{G}(\boldsymbol{\theta})[\boldsymbol{\theta} - \boldsymbol{f}(\boldsymbol{\theta})] + \mathcal{L}(\boldsymbol{\theta} + \Delta\boldsymbol{\theta}) \, . \tag{24}$$

We can generalize this objective to something which operates in continuous-time over multiple steps by the same argument as behavior. We must only make the small change $\boldsymbol{f} \to \boldsymbol{f}\Delta t$ so that the continuous-time limit is well-defined. We obtain

$$J(\{\boldsymbol{\theta}_t\}) = \int_0^\infty \left[ \frac{1}{2\eta} [\dot{\boldsymbol{\theta}}_t - \boldsymbol{f}(\boldsymbol{\theta}_t)]^T \boldsymbol{G}(\boldsymbol{\theta}_t)[\dot{\boldsymbol{\theta}}_t - \boldsymbol{f}(\boldsymbol{\theta}_t)] + k\mathcal{L}(\boldsymbol{\theta}_t) \right] e^{-\gamma t} \, dt \, . \tag{25}$$

### A.5  Accounting for loss approximation

Lastly, we must account for our assumption that the true loss $\mathcal{L}$ is generally only partially observable. Since we have already posed learning as a reinforcement learning and optimal control problem, this change is easy to make. In those settings, one accounts for random variables by averaging the objective over them; we will do the same here. We finally have

$$J(\{\boldsymbol{\theta}_t\}) = \mathbb{E}_{\{\hat{\mathcal{L}}_t\}} \left\{ \int_0^\infty \left( \frac{1}{2\eta} [\dot{\boldsymbol{\theta}}_t - \boldsymbol{f}(\boldsymbol{\theta}_t)]^T \boldsymbol{G}(\boldsymbol{\theta}_t)[\dot{\boldsymbol{\theta}}_t - \boldsymbol{f}(\boldsymbol{\theta}_t)] + k\hat{\mathcal{L}}_t(\boldsymbol{\theta}_t) \right) e^{-\gamma t} \, dt \right\} \, . \tag{26}$$

# B  Experiment details

In this appendix, we provide details relevant to understanding the numerical experiments mentioned in the main text. See `https://github.com/john-vastola/lossnav-neurips25` for code that reproduces all figures.

**Direct optimization of the objective.**   In Fig. 1, we numerically estimate the minimizer of the objective

$$J(\{\boldsymbol{\theta}_t\}) = \int_0^\infty \left( \frac{1}{2\eta}\|\dot{\boldsymbol{\theta}}_t\|^2 + k\mathcal{L}(\boldsymbol{\theta}_t) \right) e^{-\gamma t}\, dt = \int_0^\infty \left( \frac{\dot{\theta_1}^2}{2\eta} + \frac{\dot{\theta_2}^2}{2\eta} + k\mathcal{L}(\theta_1, \theta_2) \right) e^{-\gamma t}\, dt \tag{27}$$

given a double-well loss

$$\mathcal{L}(\theta_1, \theta_2) = a\,(\theta_1^2 - b)^2 + c\,\theta_2^2 + d\,\theta_1 \tag{28}$$

with $a = 1$, $b = 1$, $c = 1$, and $d = -1/2$, assuming $k = \eta = 1$ and $\gamma = 0.1$. We do this optimization directly (rather than via the EL equations) by discretizing $\theta_1(t)$ and $\theta_2(t)$, i.e.,

$$J \approx \sum_{k=1}^N \left( \frac{[\theta_1(t_{k+1}) - \theta_1(t_k)]^2}{2\eta(\Delta t)^2} + \frac{[\theta_2(t_{k+1}) - \theta_2(t_k)]^2}{2\eta(\Delta t)^2} + k\mathcal{L}(\theta_1(t_k), \theta_2(t_k)) \right) e^{-\gamma t_k}(\Delta t) \tag{29}$$

where we choose $N + 1$ equally spaced time points $t_0, t_1, ..., t_N$ for simplicity. This means that $t_k := k\Delta t$ for all $k = 0, ..., N$, where $\Delta t := T/N$. Here, the cutoff time $T > 0$ is chosen to be large enough that the optimal trajectory is near the global minimum of the loss at the final time point $t_N$. (This means that, even though we do not integrate all the way until $t = \infty$, not much interesting happens beyond time $T$.)

By fixing the initial point $\boldsymbol{\theta}(t_0) = (\theta_1(t_0), \theta_2(t_0))^T$, final point $\boldsymbol{\theta}(t_N) = (\theta_1(t_N), \theta_2(t_N))^T$, and the cutoff time $T$, we can vary the remaining $2(N-1)$ degrees of freedom to determine the optimal trajectory. Following Strang et al. [29], we minimize $J$ with respect to these variables using a PyTorch-based gradient descent approach.

Given the solution, we can estimate the 'kinetic energy' throughout a trajectory by computing

$$\text{KE}(t_k) := \left( \frac{[\theta_1(t_{k+1}) - \theta_1(t_k)]^2}{2\eta(\Delta t)^2} + \frac{[\theta_2(t_{k+1}) - \theta_2(t_k)]^2}{2\eta(\Delta t)^2} \right) \tag{30}$$

and the 'potential energy' by computing

$$\text{PE}(t_k) := k\mathcal{L}(\theta_1(t_k), \theta_2(t_k)) . \tag{31}$$

**Application of ballistic rule to MNIST and CIFAR-10 image classification.**   For Fig. 2, we implement the 'ballistic' rule that emerges from one of our exact solutions in the $\gamma = 0$ limit (see Eq. 7), and use it to train classifiers on the MNIST [55] and CIFAR-10 [56] image datasets. At each step, the ballistic rule prescribes a parameter update proportional to

$$\Delta\boldsymbol{\theta} \propto -\boldsymbol{H}^{-1/2}\boldsymbol{g} \tag{32}$$

where $\boldsymbol{g}$ is the current loss gradient and $\boldsymbol{H}$ is the current Hessian of the loss. Since Hessian computation is expensive and difficult to scale, we implement the ballistic rule by using an Adam-like [13] approach: at each step, we update a running average of (uncentered) gradient variances along each direction. This involves three crude approximations: we use this running average instead of directly computing the Hessian; we only compute the diagonal entries of the Hessian proxy; and we do not center the variance estimates. Despite these approximations, we still believe that this heuristic approach captures the spirit of the ballistic rule. Furthermore, using an Adam-like approach is theoretically reasonable given the link we identify between Adam and our ballistic rule (Sec. 5).

Let $v_i$ denote the running gradient variance associated with the parameter $\theta_i$. For each $i$, the precise updates per step are

$$\begin{aligned} v_i &\leftarrow \beta_2 v_i + (1 - \beta_2)g_i^2 \\ \Delta\theta_i &= -\eta g_i/\sqrt{v_i} \end{aligned} \tag{33}$$

where $g_i$ denotes the current gradient along the $\theta_i$ direction, $\eta$ denotes the learning rate, and $\beta_2 \in [0, 1]$ controls the time scale on which gradient observations are averaged. Note that this usage of $\beta_2$ is intended to match Adam's; like in Adam, values like $\beta_2 = 0.9$ and $\beta_2 = 0.999$ (i.e., values close to one) appear to work well.

We consider only two simple architectures to illustrate the ballistic rule's behavior:

- a multilayer perceptron (MLP) with two hidden layers, which we trained on MNIST; and
- a small convolutional neural network (CNN) with four convolutional layers, two max pooling layers, and a final fully-connected layer, which we trained on CIFAR-10.

We generally found that, especially since this implementation of the ballistic rule is similar to the standard implementation of Adam, hyperparameter settings that work well for Adam also work well for it. For example, learning rates around `1e-3` and `1e-4` worked well. For MNIST, $\beta_2 = 0.999$ performed reasonably well, but for CIFAR-10 a lower value ($\beta_2 = 0.9$) appeared to be necessary for good performance.

# C   Deriving learning rules via the Euler-Lagrange equations

Suppose that the parameter vector $\boldsymbol{\theta}$ is $D$-dimensional. After averaging over the loss landscape belief model in Eq. 1, our objective has the form

$$J(\{\boldsymbol{\theta}_t\}) = \int_0^\infty \left[ \frac{1}{2\eta}[\dot{\boldsymbol{\theta}}_t - \boldsymbol{f}(\boldsymbol{\theta}_t)]^T \boldsymbol{G}(\boldsymbol{\theta}_t)[\dot{\boldsymbol{\theta}}_t - \boldsymbol{f}(\boldsymbol{\theta}_t)] + k\mathcal{L}(\boldsymbol{\theta}_t) \right] e^{-\gamma t}\, dt \tag{34}$$

for some $\mathcal{L}$, $\boldsymbol{G}$, and so on.[4] Given an objective like this, how do we go about (analytically) deriving learning rules?

We are looking for a trajectory $\{\boldsymbol{\theta}_t\}_{t\in[0,\infty)}$ which minimizes $J$. This trajectory is not necessarily unique, for example due to symmetry, but is often unique in practice. By the calculus of variations, subject to the relevant boundary conditions (and smoothness-related technical conditions), the optimal trajectory can be shown [28, 30] to satisfy the *Euler-Lagrange (EL) equations*.

The idea is to use the objective to define a Lagrangian

$$L(\boldsymbol{\theta}, \dot{\boldsymbol{\theta}}, t) := \left[ \frac{1}{2\eta}[\dot{\boldsymbol{\theta}} - \boldsymbol{f}(\boldsymbol{\theta})]^T \boldsymbol{G}(\boldsymbol{\theta})[\dot{\boldsymbol{\theta}} - \boldsymbol{f}(\boldsymbol{\theta})] + k\mathcal{L}(\boldsymbol{\theta}) \right] e^{-\gamma t} . \tag{35}$$

The Euler-Lagrange equations are the $D$ equations

$$\frac{d}{dt}\left( \frac{\partial L}{\partial \dot{\boldsymbol{\theta}}} \right) = \frac{\partial L}{\partial \boldsymbol{\theta}} \qquad , \text{i.e.,} \qquad \frac{d}{dt}\left( \frac{\partial L}{\partial \dot{\theta}_i} \right) = \frac{\partial L}{\partial \theta_i} \qquad \text{for } i = 1, ..., D . \tag{36}$$

More explicitly, we have $D$ second-order ODEs

$$\frac{d}{dt}\left( \boldsymbol{G}(\boldsymbol{\theta})[\dot{\boldsymbol{\theta}} - \boldsymbol{f}(\boldsymbol{\theta})] \right) - \gamma \boldsymbol{G}(\boldsymbol{\theta})[\dot{\boldsymbol{\theta}} - \boldsymbol{f}(\boldsymbol{\theta})] = \eta k \nabla_{\boldsymbol{\theta}} \mathcal{L}(\boldsymbol{\theta}) + \frac{\partial}{\partial \boldsymbol{\theta}}\left( \frac{1}{2}[\dot{\boldsymbol{\theta}} - \boldsymbol{f}(\boldsymbol{\theta})]^T \boldsymbol{G}(\boldsymbol{\theta})[\dot{\boldsymbol{\theta}} - \boldsymbol{f}(\boldsymbol{\theta})] \right) . \tag{37}$$

Since they are second-order, **specifying solutions to these ODEs** (up to symmetry-related degeneracies) **requires two pieces of boundary data**. In classical mechanics [28], one often looks for solutions with a specified *initial position* and *initial momentum* (or equivalently, initial velocity). In our context, this does not quite make sense. We *do* want a solution with a specified initial parameter vector $\boldsymbol{\theta}_0$, but requiring some particular initial velocity or momentum is less obviously meaningful.

The second piece of boundary data follows from the fact that we would like $J$ to be *minimized*. This is equivalent to requiring that the asymptotic ($t \to \infty$) endpoint of the trajectory corresponds to the global minimum of the loss, at least if $\boldsymbol{f} \equiv \boldsymbol{0}$. If $\boldsymbol{f}$ is nonzero, the drift contributes a term to the 'effective' loss, and it is the global minimum of *this* function which must be asymptotically approached instead.

**In short, we can find optimal trajectories $\{\boldsymbol{\theta}_t\}_{t\in[0,\infty)}$ by solving the EL equations** (Eq. 37) **subject to the constraints that:**

- $\boldsymbol{\theta}_0$ is fixed, and corresponds to the 'current' parameter vector.
- The remaining degree of freedom (e.g., the trajectory endpoint or initial momentum) is chosen so that $J$ is minimized.

Below, we provide two simple but instructive one-dimensional examples that provide intuition about how this process plays out in practice.

## C.1   Example: quadratic loss

Suppose $D = 1$, $\boldsymbol{G} = \boldsymbol{I}$, $\boldsymbol{f} \equiv \boldsymbol{0}$, and

$$\mathcal{L}(\theta) = \frac{h}{2}(\theta - \theta_*)^2 , \tag{38}$$

---

[4]If the loss model $\hat{\mathcal{L}}$ involves parameters (e.g., estimated gradients and curvature) which are themselves dynamic, these parameters should be considered components of $\boldsymbol{\theta}$, and can hence contribute terms to an 'effective' metric $\boldsymbol{G}$, an effective drift $\boldsymbol{f}$, etc. This is easiest to see in the context of specific examples, like the ones that appear in Sec. 5.

i.e., the loss is quadratic with a global minimum at $\theta_*$. Eq. 34 becomes

$$J(\{\theta_t\}) = \int_0^\infty \left[ \frac{1}{2\eta} \dot{\theta}_t^2 + \frac{kh}{2} (\theta_t - \theta_*)^2 \right] e^{-\gamma t} \, dt \,. \tag{39}$$

The corresponding Lagrangian is

$$L(\theta, \dot{\theta}, t) = \left[ \frac{1}{2\eta} \dot{\theta}^2 + \frac{kh}{2} (\theta - \theta_*)^2 \right] e^{-\gamma t} \,, \tag{40}$$

and the corresponding EL equation is

$$\ddot{\theta}_t - \gamma \dot{\theta}_t = \eta kh (\theta_t - \theta_*) \implies \ddot{\theta}_t - \gamma \dot{\theta}_t - \eta kh \theta_t = -\eta kh \theta_* \,. \tag{41}$$

This is a linear, second-order ODE with constant coefficients, and so can be solved in the usual way[5]. This ODE's characteristic equation is

$$r^2 - \gamma r - \eta kh = 0 \tag{42}$$

and has roots

$$r_\pm = \frac{\gamma}{2} \pm \sqrt{\frac{\gamma^2}{4} + \eta kh} \,. \tag{43}$$

Importantly, one of these roots is positive and one is negative, a fact which we will return to shortly. The full solution can hence be written

$$\theta_t = \theta_* + c_+ e^{r_+ t} + c_- e^{r_- t} \tag{44}$$

where $\theta_*$ is the (obvious) particular solution. Enforcing the initial condition,

$$\theta_0 = \theta_* + c_+ + c_- \implies c_- = \theta_0 - \theta_* - c_+ \,. \tag{45}$$

Hence,

$$\begin{aligned}
\theta_t &= \theta_* + c_+ e^{r_+ t} + (\theta_0 - \theta_* - c_+) e^{r_- t} \\
\dot{\theta}_t &= c_+ r_+ e^{r_+ t} + (\theta_0 - \theta_* - c_+) r_- e^{r_- t} \,.
\end{aligned} \tag{46}$$

We can determine $c_+$ by substituting these into $J$ and minimizing it with respect to $c_+$. Note,

$$\begin{aligned}
(\theta_t - \theta_*)^2 &= c_+^2 e^{2r_+ t} + (\theta_0 - \theta_* - c_+)^2 e^{2r_- t} + 2c_+ (\theta_0 - \theta_* - c_+) e^{(r_+ + r_-)t} \\
\dot{\theta}_t^2 &= c_+^2 r_+^2 e^{2r_+ t} + (\theta_0 - \theta_* - c_+)^2 r_-^2 e^{2r_- t} + 2c_+ r_+ r_- (\theta_0 - \theta_* - c_+) e^{(r_+ + r_-)t} \,.
\end{aligned} \tag{47}$$

Since

$$\begin{aligned}
r_+^2 &= \gamma r_+ + \eta kh \\
r_-^2 &= \gamma r_- + \eta kh \\
r_+ r_- &= -\eta kh \,,
\end{aligned} \tag{48}$$

the $\dot{\theta}_t^2$ expression can be simplified to

$$\begin{aligned}
\dot{\theta}_t^2 &= \eta kh \left[ c_+^2 e^{2r_+ t} + (\theta_0 - \theta_* - c_+)^2 e^{2r_- t} - 2c_+ (\theta_0 - \theta_* - c_+) e^{(r_+ + r_-)t} \right] \\
&+ \gamma \left[ c_+^2 r_+ e^{2r_+ t} + (\theta_0 - \theta_* - c_+)^2 r_- e^{2r_- t} \right] \,.
\end{aligned} \tag{49}$$

After some algebra, the integrand of $J$ is

$$\left( kh + \frac{\gamma}{2\eta} r_+ \right) c_+^2 e^{(2r_+ - \gamma)t} + \left( kh + \frac{\gamma}{2\eta} r_- \right) (\theta_0 - \theta_* - c_+)^2 e^{(2r_- - \gamma)t} \,. \tag{50}$$

Integrating from $t = 0$ to $t = T$, $J$ equals

$$J = \lim_{T \to \infty} \left( kh + \frac{\gamma}{2\eta} r_+ \right) c_+^2 \frac{\left( e^{(2r_+ - \gamma)T} - 1 \right)}{2r_+ - \gamma} + \left( kh + \frac{\gamma}{2\eta} r_- \right) (\theta_0 - \theta_* - c_+)^2 \frac{\left( e^{(2r_- - \gamma)T} - 1 \right)}{2r_- - \gamma} \,. \tag{51}$$

---

[5]Physically, it is analogous to a damped harmonic oscillator, but with the 'wrong' sign on the $\theta_t$ coefficient. This difference makes sense, since a learning trajectory ought to eventually settle down into a global minimum rather than oscillate.

Recall that $r_+$ is positive. The quantity

$$2r_+ - \gamma = \sqrt{\frac{\gamma^2}{4} + \eta k h} \tag{52}$$

is also positive, which means that the term with $e^{(2r_+-\gamma)T}$ approaches infinity in the $T \to \infty$ limit we're interested in. Inspecting Eq. 51, the only way to remove the offending term is to set $c_+ = 0$.

But this outcome was foreseeable if we note that the positive-root term $e^{r_+t}$ 'blows up' in the long-time limit, whereas the other term doesn't. For this reason, in future derivations we will bypass this argument and simply set $c_+$ (or its higher-dimensional analogue) to zero.

Incidentally, if our objective involved a finite time horizon $t \in [0, T]$ rather than $t \in [0, \infty)$, in general we would have $c_+ \neq 0$, which would somewhat complicate our expressions for optimal learning trajectories.

Setting $c_+ = 0$, we finally find that the optimal learning trajectory has

$$\begin{aligned} \theta_t &= \theta_* + (\theta_0 - \theta_*)e^{r_-t} \\ \dot{\theta}_t &= (\theta_0 - \theta_*)r_- e^{r_-t} \ , \end{aligned} \tag{53}$$

i.e., it approaches the global minimum exponentially quickly, at a rate $r_-$.

## C.2 Example: double-well loss

While the previous example is instructive, the loss function we considered was convex, and involved only one (local/global) minimum. It is somewhat more interesting to see what happens with a double-well loss

$$\mathcal{L}(\theta) = \frac{h}{4}(\theta^2 - \theta_*^2)^2 - \frac{q}{3}\theta^3 - \mathcal{L}_{min} \tag{54}$$

where $h > 0$ and $q > 0$, and where the additive constant $\mathcal{L}_{min}$ is chosen so that $\mathcal{L}$ equals zero at its global minimum. This loss has two local minima, as is clear from its derivative:

$$\mathcal{L}'(\theta) = h\theta(\theta^2 - \theta_*^2) - q\theta^2 \ . \tag{55}$$

In particular, since

$$\mathcal{L}'(\theta) = 0 \implies h\theta\left[\theta^2 - \frac{q}{h}\theta - \theta_*^2\right] \ , \tag{56}$$

the two minima are at

$$\theta_\pm := \pm\sqrt{\theta_*^2 + \frac{q^2}{4h^2}} + \frac{q}{2h} \ , \tag{57}$$

and are near $\pm\theta_*$ if $q$ is small. The lower minimum is at $\theta_+$; the $h$ term is identical at both $\theta_+$ and $\theta_-$, but the $q$ term is negative at $\theta_+$.

For this loss, the objective $J$ is

$$J(\{\theta_t\}) = \int_0^\infty \left[\frac{1}{2\eta}\dot{\theta}_t^2 + k\left(\frac{h}{4}(\theta^2 - \theta_*^2)^2 - \frac{q}{3}\theta^3\right)\right] e^{-\gamma t} \ dt \tag{58}$$

and the EL equation is

$$\ddot{\theta}_t - \gamma\dot{\theta}_t = \eta k\mathcal{L}'(\theta_t) = \eta k h\theta_t(\theta_t - \theta_+)(\theta_t - \theta_-) \ . \tag{59}$$

This second-order ODE is highly nonlinear, and it is not obvious if it is analytically solvable. If we assume $\gamma = 0$ (i.e., no temporal discounting), the EL equation becomes

$$\ddot{\theta}_t = \eta k\mathcal{L}'(\theta_t) \ . \tag{60}$$

This can be simplified somewhat by using a trick. If we multiply the left-hand side and right-hand side by $\dot{\theta}_t$, we can integrate both with respect to time; we obtain

$$\dot{\theta}_t^2 = 2\eta k\mathcal{L}(\theta_t) + 2\eta E \tag{61}$$

where $E$ is a constant. Since the left-hand side is nonnegative, $E \geq -k\mathcal{L}(\theta_{min})$, where $\theta_{min}$ is the argument for which $\mathcal{L}$ achieves its global minimum. (Here, due to the additive offset we included, $\mathcal{L}(\theta_{min}) = 0$.)

We label this constant $E$ since it corresponds to this setting's notion of energy, as we could figure out from an analysis of this problem's Hamiltonian[6].

Eq. 61 implies that

$$J = \lim_{T\to\infty} \int_0^T \frac{1}{2\eta}\dot{\theta}_t^2 + k\mathcal{L}(\theta_t)\, dt = \lim_{T\to\infty} \int_0^T 2k\mathcal{L}(\theta_t) + E\, dt = \lim_{T\to\infty} \int_0^T 2k\mathcal{L}(\theta_t)\, dt + ET \quad (62)$$

along the optimal trajectory. But this is a problem, since $ET \to \infty$ as $T \to \infty$ for most choices of $E$. We do best by setting energy equal to its minimum possible value $E = -k\mathcal{L}(\theta_{min}) = 0$, since the problematic term vanishes and the loss term asymptotically approaches zero (since the value of the loss at the global minimum is zero). That is,

$$J = \lim_{T\to\infty} \int_0^T 2k\mathcal{L}(\theta_t)\, dt < \infty \, . \quad (63)$$

Going back to Eq. 61, this means that

$$\dot{\theta}_t = \pm\sqrt{2\eta k\mathcal{L}(\theta_t)} \, . \quad (64)$$

This defines an optimal learning trajectory, and also an optimal learning rule.

What does this equation mean? To understand this expression, suppose $\theta_0 = \theta_-$, i.e., that the learner begins in the shallower minimum. Clearly, the optimal learning dynamics must move right (towards $\theta_+$); the optimal trajectory in this case has

$$\dot{\theta}_t = \sqrt{2\eta k\mathcal{L}(\theta_t)} \, , \quad (65)$$

i.e., we take the plus sign. If the learner begins at a parameter $\theta > \theta_+$, we would instead take the minus sign.

Although this example is somewhat complicated, there are two important takeaways:

- Long-horizon optimal learning trajectories approach the global minimum rather than any local minima.
- Even if the functional form of $\mathcal{L}$ is more complicated than quadratic, it is in some cases possible to derive optimal learning rules and trajectories.

In the rest of this paper, partly because analyses like the above are difficult, and partly because our goal is to derive well-known learning rules, we essentially only consider quadratic (local) approximations of the loss.

---

[6]See Vastola [44] for the details of how to follow this line of thought in an analogous theoretical setting. At least if $\gamma = 0$, this notion of energy is conserved along the optimal trajectory.

# D   More on the boundary conditions of the Euler-Lagrange equations

The boundary conditions for the objective minimization problem we describe in Sec. 2 are reasonably clear: of all the possible well-behaved (e.g., smooth, bounded) trajectories $\{\boldsymbol{\theta}_t\}_{t\in[0,\infty)}$ through the $D$-dimensional parameter space which begin at a prescribed initial parameter vector $\boldsymbol{\theta}_0$, the 'optimal' trajectory is the one which makes $J$ smallest. 'Smallest' here is a well-defined notion, since $J$ is bounded from below as long as the loss $\mathcal{L}$ is bounded from below. If multiple trajectories make $J$ as small as possible, then each of them is optimal.

However, the boundary conditions for the Euler-Lagrange (EL) equations are less obvious. The EL equations answer the following question: given all possible smooth trajectories that begin at $\boldsymbol{\theta}_0$ and reach $\boldsymbol{\theta}(T)$ at time $T > 0$, which of them makes the Lagrangian (i.e., the integrand of the objective) *stationary*? This is three steps removed from our original minimization problem, since (i) stationary points may not correspond to local minima, (ii) local minima may not be global minima, and (iii) a given $\boldsymbol{\theta}(T)$ may not correspond to an optimal trajectory.

We think about the boundary conditions of the EL equations we encounter in this paper in the following way. First, we assume that the trajectory of interest has a prescribed initial point $\boldsymbol{\theta}_0$, since this is a boundary condition of the original minimization problem. Second, since (according to standard calculus of variations results, under mild assumptions[7]) the Lagrangian $L$ is stationary at the global minimizer of $J$, satisfying the EL equations is a *necessary* (but not sufficient) condition for a trajectory to be optimal. Third, any two trajectories can be compared (i.e., which corresponds to a lower value of $J$?).

Together, the first two insights tell us that the EL equations must be satisfied for a specific initial point $\boldsymbol{\theta}_0$ and some other point $\boldsymbol{\theta}(T)$. The third insight tells us that, if we do not know which $\boldsymbol{\theta}(T)$ to use, we can compare any two possibilities by comparing the corresponding values of $J$. This yields a two-level optimization strategy: for many endpoints $\boldsymbol{\theta}(T)$, solve the EL equations; then, choose the solution whose corresponding $J$ is smallest.

This strategy is not circular, and is useful because it reduces an optimization problem defined over an infinite-dimensional function space (i.e., the infinite-dimensional space of all possible parameter trajectories) to an optimization problem over a $D$-dimensional space (since each possible solution corresponds to a particular choice of $\boldsymbol{\theta}(T) \in \mathbb{R}^D$).

Finally, it is worth noting that the rather abstract conditions under which a calculus of variations problem is interesting and well-defined are not that useful for understanding many of the extremely simple objectives we consider in this paper. Especially given a quadratic loss, we can often verify directly (through algebra rather than numerics) that solutions of the EL equations correspond to global minimizers of $J$, and even directly compute $J$ as a function of any $\boldsymbol{\theta}_0$ and $\boldsymbol{\theta}(T)$.

---

[7]For example, we may want to assume that the kinetic term is convex and coercive. These two properties hold for the simple quadratic kinetic terms we consider throughout.

# E   Deriving learning rules with momentum: more details

In this appendix, we derive the results presented in Sec. 3. The relevant objective is

$$J(\{\boldsymbol{\theta}_t\}) = \int_0^\infty \left( \frac{1}{2\eta} \|\dot{\boldsymbol{\theta}}_t\|^2 + k\mathcal{L}(\boldsymbol{\theta}_t) \right) e^{-\gamma t} \, dt \ . \tag{66}$$

The corresponding Lagrangian is

$$L(\boldsymbol{\theta}, \dot{\boldsymbol{\theta}}, t) = \left( \frac{1}{2\eta} \|\dot{\boldsymbol{\theta}}\|^2 + k\mathcal{L}(\boldsymbol{\theta}) \right) e^{-\gamma t} \tag{67}$$

and the corresponding EL equations are

$$\ddot{\boldsymbol{\theta}}_t - \gamma\dot{\boldsymbol{\theta}}_t = \eta k \nabla_{\boldsymbol{\theta}_t} \mathcal{L}(\boldsymbol{\theta}_t) \ . \tag{68}$$

We can rewrite these equations in a suggestive form by defining 'momentum' as $\boldsymbol{p}_t := \dot{\boldsymbol{\theta}}_t$, so that we have

$$\dot{\boldsymbol{\theta}}_t = \boldsymbol{p}_t \qquad\qquad \dot{\boldsymbol{p}}_t = \gamma\boldsymbol{p}_t + \eta k \nabla_{\boldsymbol{\theta}_t} \mathcal{L}(\boldsymbol{\theta}_t) \ . \tag{69}$$

Note that our convention for momentum matches machine learning practice, but does not necessarily match the physics convention for momentum, which has $\boldsymbol{p}_t := \frac{\partial L}{\partial \dot{\boldsymbol{\theta}}_t}$.

## E.1   Solution to the Euler-Lagrange equations for a quadratic loss

If we assume a quadratic loss

$$\hat{\mathcal{L}}(\boldsymbol{\theta}_t) := \mathcal{L}(\boldsymbol{\theta}_0) + \boldsymbol{g}^T(\boldsymbol{\theta}_t - \boldsymbol{\theta}_0) + \frac{1}{2}(\boldsymbol{\theta}_t - \boldsymbol{\theta}_0)^T \boldsymbol{H}(\boldsymbol{\theta}_t - \boldsymbol{\theta}_0) \ , \tag{70}$$

where $\boldsymbol{g}$ is the local gradient and $\boldsymbol{H}$ is the (symmetric, positive definite) local Hessian, the EL equations (Eq. 68) become

$$\ddot{\boldsymbol{\theta}}_t - \gamma\dot{\boldsymbol{\theta}}_t = \eta k \left[ \boldsymbol{g} + \boldsymbol{H}(\boldsymbol{\theta}_t - \boldsymbol{\theta}_0) \right] \implies \ddot{\boldsymbol{\theta}}_t - \gamma\dot{\boldsymbol{\theta}}_t - \eta k \boldsymbol{H}\boldsymbol{\theta}_t = \eta k \left[ \boldsymbol{g} - \boldsymbol{H}\boldsymbol{\theta}_0 \right] \ . \tag{71}$$

The above represents a system of coupled second-order ODEs. We can decouple it by exploiting an eigendecomposition of $\boldsymbol{H}$. Since $\boldsymbol{H} = \boldsymbol{Q}\boldsymbol{\Lambda}\boldsymbol{Q}^T$ for some orthogonal matrix $\boldsymbol{Q}$ and diagonal matrix $\boldsymbol{\Lambda}$ (whose diagonal entries are nonnegative), we can premultiply both sides of this equation by $\boldsymbol{Q}^T$ to obtain

$$\ddot{\boldsymbol{\phi}}_t - \gamma\dot{\boldsymbol{\phi}}_t - \eta k \boldsymbol{\Lambda}\boldsymbol{\phi}_t = \eta k \left[ \tilde{\boldsymbol{g}} - \boldsymbol{\Lambda}\boldsymbol{\phi}_0 \right] \ , \tag{72}$$

where we define $\boldsymbol{\phi}_t := \boldsymbol{Q}^T\boldsymbol{\theta}_t$ and $\tilde{\boldsymbol{g}} := \boldsymbol{Q}^T\boldsymbol{g}$. We now have many linear, second-order ODEs identical in form to the one from the first example in Appendix C. For a given $\phi_i$, we have

$$\ddot{\phi}_i - \gamma\dot{\phi}_i - \eta k \lambda_i \phi_i = \eta k \left[ \tilde{g}_i - \lambda_i \phi_{i0} \right] \ , \tag{73}$$

where $\lambda_i := \Lambda_{ii} \geq 0$. The corresponding characteristic equation reads

$$r^2 - \gamma r - \eta k \lambda_i = 0 \tag{74}$$

and has roots

$$r_\pm = \frac{\gamma}{2} \pm \sqrt{\frac{\gamma^2}{4} + \eta k \lambda_i} \ . \tag{75}$$

By the same argument we used in Appendix C, we throw away the positive root (since it does not minimize $J$) and keep the negative one. Denote the (negative) root that we keep as $r_i$. We have

$$\phi_i(t) = c_i e^{r_i t} + \phi_{i0} - \lambda_i^{-1} \tilde{g}_i \tag{76}$$

where $\phi_{i0} - \lambda_i^{-1}\tilde{g}_i$ is the particular solution of Eq. 73, and $c_i$ is a constant. We can determine $c_i$ by enforcing the initial condition. Doing so, we find

$$\phi_i(t) = \phi_{i0} - \lambda_i^{-1} \tilde{g}_i (1 + e^{r_i t}) \ . \tag{77}$$

Transforming back to $\boldsymbol{\theta}_t$ space via the relationship $\boldsymbol{\theta}_t = \boldsymbol{Q}\boldsymbol{\phi}_t$, we have

$$\theta_i(t) = \sum_j Q_{ij}\phi_{j0} - Q_{ij}\lambda_j^{-1}\tilde{g}_j(1 + e^{r_j t}) \ , \tag{78}$$

or equivalently

$$\boldsymbol{\theta}_t = \boldsymbol{Q}\boldsymbol{\phi}_0 - \boldsymbol{Q}\boldsymbol{\Lambda}^{-1}(\boldsymbol{I} + e^{\boldsymbol{R}t})\tilde{\boldsymbol{g}} = \boldsymbol{Q}\boldsymbol{\phi}_0 - \boldsymbol{Q}\boldsymbol{\Lambda}^{-1}\boldsymbol{Q}^T\boldsymbol{Q}(\boldsymbol{I} + e^{\boldsymbol{R}t})\boldsymbol{Q}^T\boldsymbol{Q}\tilde{\boldsymbol{g}}, \tag{79}$$

where we define the diagonal matrix $\boldsymbol{R}$ whose diagonal entries are the $r_i$. Noting that

$$\boldsymbol{Q}(\boldsymbol{I} - e^{\boldsymbol{R}t})\boldsymbol{Q}^T = \boldsymbol{I} - \exp\left\{-\left[\sqrt{\frac{\gamma^2}{4} + \eta k \boldsymbol{H}} - \frac{\gamma}{2}\boldsymbol{I}\right]\right\}, \tag{80}$$

we can simplify our result to

$$\boldsymbol{\theta}_t = \boldsymbol{\theta}_0 - \left(\boldsymbol{I} - \exp\left\{-\left[\sqrt{\frac{\gamma^2}{4}\boldsymbol{I} + \eta k \boldsymbol{H}} - \frac{\gamma}{2}\boldsymbol{I}\right]t\right\}\right)\boldsymbol{H}^{-1}\boldsymbol{g}. \tag{81}$$

This implies that, for any $\Delta t > 0$,

$$\boldsymbol{\theta}_{\Delta t} - \boldsymbol{\theta}_0 = -\left(\boldsymbol{I} - \exp\left\{-\left[\sqrt{\frac{\gamma^2}{4}\boldsymbol{I} + \eta k \boldsymbol{H}} - \frac{\gamma}{2}\boldsymbol{I}\right]\Delta t\right\}\right)\boldsymbol{H}^{-1}\boldsymbol{g}. \tag{82}$$

### E.2   Special cases of the quadratic loss solution

If $\Delta t$ is large, the exponential asymptotically vanishes, so

$$\lim_{\Delta t \to \infty} \boldsymbol{\theta}_{\Delta t} - \boldsymbol{\theta}_0 = -\boldsymbol{H}^{-1}\boldsymbol{g}, \tag{83}$$

and we recover Newton's method (i.e., one 'jumps' to the minimum). For large but not infinite $\Delta t$, Eq. 82 says that one ought to follow a Newton-like method, but with a possibly asymmetric learning rate in different directions, whose precise form depends on the argument of the exponential.

For small $\Delta t$, we obtain (to first order in $\Delta t$)

$$\boldsymbol{\theta}_{\Delta t} - \boldsymbol{\theta}_0 \approx \left[\sqrt{\frac{\gamma^2}{4}\boldsymbol{I} + \eta k \boldsymbol{H}} - \frac{\gamma}{2}\boldsymbol{I}\right]\boldsymbol{H}^{-1}\boldsymbol{g}\,\Delta t. \tag{84}$$

We can simplify this further if we make an assumption about the relative sizes of $\boldsymbol{H}$ and $\gamma$. If $\gamma \gg \sqrt{\eta k \lambda_i}$ for all eigenvalues $\lambda_i$ of $\boldsymbol{H}$, then

$$\frac{\gamma}{2}\left[\sqrt{\boldsymbol{I} + \frac{4\eta k}{\gamma^2}\boldsymbol{H}} - \boldsymbol{I}\right] \approx \frac{\gamma}{2}\frac{2\eta k}{\gamma^2}\boldsymbol{H} = \frac{\eta k}{\gamma}\boldsymbol{H}, \tag{85}$$

so in this limit (the 'overdamped' limit) we obtain the learning rule

$$\boldsymbol{\theta}_{\Delta t} - \boldsymbol{\theta}_0 \approx \frac{\eta k}{\gamma}\boldsymbol{H}\boldsymbol{H}^{-1}\boldsymbol{g}\,\Delta t = \frac{\eta k}{\gamma}\boldsymbol{g}\Delta t, \tag{86}$$

which just corresponds to gradient descent.

Meanwhile, if $\Delta t$ is small but $\gamma \ll \sqrt{\eta k \lambda_i}$ for all eigenvalues $\lambda_i$ of $\boldsymbol{H}$ (or if $\gamma = 0$), then

$$\boldsymbol{\theta}_{\Delta t} - \boldsymbol{\theta}_0 \approx \sqrt{\eta k \boldsymbol{H}}\boldsymbol{H}^{-1}\boldsymbol{g}\,\Delta t = \sqrt{\eta k}\boldsymbol{H}^{-1/2}\boldsymbol{g}\,\Delta t. \tag{87}$$

This 'ballistic' learning rule operates in a regime where momentum dominates learning dynamics.

Lastly, it's worth noting that if $\gamma$ is larger than some eigenvalues of $\boldsymbol{H}$ but smaller than others, one can obtain a learning rule that looks like gradient descent along some directions, and looks ballistic along other directions.

# F  Deriving learning rules in non-Euclidean parameter spaces

In this appendix, we derive the results presented in the first part of Sec. 4. The relevant objective is

$$J(\{\boldsymbol{\theta}_t\}) = \int_0^\infty \left( \frac{1}{2\eta} \dot{\boldsymbol{\theta}}_t^T \boldsymbol{G}(\boldsymbol{\theta}_t) \dot{\boldsymbol{\theta}}_t + k\mathcal{L}(\boldsymbol{\theta}_t) \right) e^{-\gamma t}\, dt \tag{88}$$

where $\boldsymbol{G}(\boldsymbol{\theta}_t)$ is a symmetric and positive definite matrix for all $\boldsymbol{\theta}_t$. The Lagrangian is

$$L(\boldsymbol{\theta}, \dot{\boldsymbol{\theta}}, t) = \left( \frac{1}{2\eta} \dot{\boldsymbol{\theta}}^T \boldsymbol{G}(\boldsymbol{\theta}) \dot{\boldsymbol{\theta}} + k\mathcal{L}(\boldsymbol{\theta}) \right) e^{-\gamma t} \tag{89}$$

and the EL equations are

$$\frac{d}{dt}\left( \boldsymbol{G}(\boldsymbol{\theta}_t) \dot{\boldsymbol{\theta}}_t \right) - \gamma \boldsymbol{G}(\boldsymbol{\theta}_t) \dot{\boldsymbol{\theta}}_t = \eta k \nabla_{\boldsymbol{\theta}_t} \mathcal{L}(\boldsymbol{\theta}_t) + \nabla_{\boldsymbol{\theta}_t} \left( \frac{1}{2} \dot{\boldsymbol{\theta}}_t^T \boldsymbol{G}(\boldsymbol{\theta}_t) \dot{\boldsymbol{\theta}}_t \right) . \tag{90}$$

If the metric is approximately independent of $\boldsymbol{\theta}_t$, we have the special form

$$\boldsymbol{G}(\boldsymbol{\theta}_t) \ddot{\boldsymbol{\theta}}_t - \gamma \boldsymbol{G}(\boldsymbol{\theta}_t) \dot{\boldsymbol{\theta}}_t \approx \eta k \nabla_{\boldsymbol{\theta}_t} \mathcal{L}(\boldsymbol{\theta}_t) \implies \ddot{\boldsymbol{\theta}}_t - \gamma \dot{\boldsymbol{\theta}}_t \approx \eta k \boldsymbol{G}(\boldsymbol{\theta}_t)^{-1} \nabla_{\boldsymbol{\theta}_t} \mathcal{L}(\boldsymbol{\theta}_t) \tag{91}$$

that appears in the main text. We can rewrite this in terms of the 'momentum' $\boldsymbol{p}_t := \dot{\boldsymbol{\theta}}_t$ to exactly reproduce the expression from Sec. 4.

## F.1  Solution to the Euler-Lagrange equations for a quadratic loss

Assume $\boldsymbol{G}$ is independent of $\boldsymbol{\theta}_t$ and that the loss is quadratic, i.e.,

$$\hat{\mathcal{L}}(\boldsymbol{\theta}_t) := \mathcal{L}(\boldsymbol{\theta}_0) + \boldsymbol{g}^T(\boldsymbol{\theta}_t - \boldsymbol{\theta}_0) + \frac{1}{2}(\boldsymbol{\theta}_t - \boldsymbol{\theta}_0)^T \boldsymbol{H}(\boldsymbol{\theta}_t - \boldsymbol{\theta}_0) , \tag{92}$$

where $\boldsymbol{g}$ is the local gradient and $\boldsymbol{H}$ is the (symmetric, positive definite) local Hessian. The EL equations become

$$\ddot{\boldsymbol{\theta}}_t - \gamma \dot{\boldsymbol{\theta}}_t = \eta k \boldsymbol{G}^{-1}\left[ \boldsymbol{g} + \boldsymbol{H}(\boldsymbol{\theta}_t - \boldsymbol{\theta}_0) \right] . \tag{93}$$

This is identical to what we considered in Appendix E, up to the changes $\boldsymbol{g} \to \boldsymbol{G}^{-1}\boldsymbol{g}$ and $\boldsymbol{H} \to \boldsymbol{G}^{-1}\boldsymbol{H}$. The solution, then, is also identical up to these changes.

# G  Non-gradient learning rules from partial controllability

In this appendix, we derive results related to the second part of Sec. 4, which concerns the influence of a 'drift' term (e.g., due to partial controllability) on optimal learning trajectories. The relevant objective is

$$J(\{\boldsymbol{\theta}_t\}) = \int_0^\infty \left( \frac{1}{2\eta} [\dot{\boldsymbol{\theta}}_t - \boldsymbol{f}(\boldsymbol{\theta}_t)]^T \boldsymbol{G}(\boldsymbol{\theta}_t)[\dot{\boldsymbol{\theta}}_t - \boldsymbol{f}(\boldsymbol{\theta}_t)] + k\mathcal{L}(\boldsymbol{\theta}_t) \right) e^{-\gamma t}\, dt\,. \tag{94}$$

The Lagrangian is

$$L(\boldsymbol{\theta}, \dot{\boldsymbol{\theta}}, t) = \left( \frac{1}{2\eta} [\dot{\boldsymbol{\theta}} - \boldsymbol{f}(\boldsymbol{\theta})]^T \boldsymbol{G}(\boldsymbol{\theta})[\dot{\boldsymbol{\theta}} - \boldsymbol{f}(\boldsymbol{\theta})] + k\mathcal{L}(\boldsymbol{\theta}) \right) e^{-\gamma t} \tag{95}$$

and the EL equations are

$$\frac{d}{dt}\left( \boldsymbol{G}(\boldsymbol{\theta}_t)[\dot{\boldsymbol{\theta}}_t - \boldsymbol{f}(\boldsymbol{\theta}_t)] \right) - \gamma \boldsymbol{G}(\boldsymbol{\theta}_t)[\dot{\boldsymbol{\theta}}_t - \boldsymbol{f}(\boldsymbol{\theta}_t)] = \eta k \nabla_{\boldsymbol{\theta}_t}\mathcal{L}(\boldsymbol{\theta}_t) + \nabla_{\boldsymbol{\theta}_t}\left( \frac{1}{2}[\dot{\boldsymbol{\theta}}_t - \boldsymbol{f}(\boldsymbol{\theta}_t)]^T \boldsymbol{G}(\boldsymbol{\theta}_t)[\dot{\boldsymbol{\theta}}_t - \boldsymbol{f}(\boldsymbol{\theta}_t)] \right)\,.$$

**Trivial metric.**  In the special case that $\boldsymbol{G} = \boldsymbol{I}$, we have

$$\ddot{\boldsymbol{\theta}}_t - \frac{d}{dt}\boldsymbol{f}(\boldsymbol{\theta}_t) - \gamma[\dot{\boldsymbol{\theta}}_t - \boldsymbol{f}(\boldsymbol{\theta}_t)] = \eta k \nabla_{\boldsymbol{\theta}_t}\mathcal{L}(\boldsymbol{\theta}_t) + \nabla_{\boldsymbol{\theta}_t}\left( \frac{1}{2}\|\dot{\boldsymbol{\theta}}_t - \boldsymbol{f}(\boldsymbol{\theta}_t)\|^2 \right)\,. \tag{96}$$

Recall that the Jacobian $\boldsymbol{J}$ of $\boldsymbol{f}$ is defined as[8]

$$J_{ij} := \frac{\partial f_i(\boldsymbol{\theta})}{\partial \theta_j}\,. \tag{97}$$

Using it, we can more explicitly write our expression as

$$\ddot{\boldsymbol{\theta}}_t - \boldsymbol{J}(\boldsymbol{\theta}_t)\dot{\boldsymbol{\theta}}_t - \gamma[\dot{\boldsymbol{\theta}}_t - \boldsymbol{f}(\boldsymbol{\theta}_t)] = \eta k \nabla_{\boldsymbol{\theta}_t}\mathcal{L}(\boldsymbol{\theta}_t) - \boldsymbol{J}(\boldsymbol{\theta}_t)^T[\dot{\boldsymbol{\theta}}_t - \boldsymbol{f}(\boldsymbol{\theta}_t)]\,. \tag{98}$$

Rearranging this, we obtain

$$\ddot{\boldsymbol{\theta}}_t - [\gamma\boldsymbol{I} + \boldsymbol{J}(\boldsymbol{\theta}_t) - \boldsymbol{J}(\boldsymbol{\theta}_t)^T]\,\dot{\boldsymbol{\theta}}_t = [\boldsymbol{J}(\boldsymbol{\theta}_t)^T - \gamma\boldsymbol{I}]\boldsymbol{f}(\boldsymbol{\theta}_t) + \eta k \nabla_{\boldsymbol{\theta}_t}\mathcal{L}(\boldsymbol{\theta}_t)\,, \tag{99}$$

the equation that appears in the main text.

**Trivial metric and linear drift.**  In the special case that $\boldsymbol{f}(\boldsymbol{\theta}) = \boldsymbol{J}\boldsymbol{\theta}$,

$$\ddot{\boldsymbol{\theta}}_t - [\gamma\boldsymbol{I} + \boldsymbol{J} - \boldsymbol{J}^T]\,\dot{\boldsymbol{\theta}}_t = [\boldsymbol{J}^T - \gamma\boldsymbol{I}]\boldsymbol{J}\boldsymbol{\theta}_t + \eta k \nabla_{\boldsymbol{\theta}_t}\mathcal{L}(\boldsymbol{\theta}_t)\,. \tag{100}$$

In the case that $\mathcal{L}$ is approximated as locally quadratic (as in, e.g., Appendix E), this becomes

$$\ddot{\boldsymbol{\theta}}_t - [\gamma\boldsymbol{I} + \boldsymbol{J} - \boldsymbol{J}^T]\,\dot{\boldsymbol{\theta}}_t = [\boldsymbol{J}^T - \gamma\boldsymbol{I}]\boldsymbol{J}\boldsymbol{\theta}_t + \eta k\,[\boldsymbol{g} + \boldsymbol{H}(\boldsymbol{\theta}_t - \boldsymbol{\theta}_0)]\,. \tag{101}$$

In the next two subsections, we will consider two explicit examples of dynamics of this form.

## G.1  Example: drift term with rotational dynamics

**Setup.**  For simplicity, assume that the Hessian is isotropic, i.e., $\boldsymbol{H} = h\boldsymbol{I}$. Assume that the default dynamics are *rotational* in the sense that $\boldsymbol{f}(\boldsymbol{\theta}) = \boldsymbol{J}\boldsymbol{\theta}$, where $\boldsymbol{J}$ is *skew-symmetric*, i.e., $\boldsymbol{J} = -\boldsymbol{J}^T$. If $\boldsymbol{J}$ is skew-symmetric, solutions of the default dynamics $\dot{\boldsymbol{\theta}} = \boldsymbol{f}(\boldsymbol{\theta})$ are purely rotational, since they have the form

$$\boldsymbol{\theta}_t = e^{\boldsymbol{J}t}\boldsymbol{\theta}_0 \tag{102}$$

where $\exp(\boldsymbol{J}t)$ is a rotation matrix.

The EL equations have the form

$$\ddot{\boldsymbol{\theta}}_t - [\gamma\boldsymbol{I} + 2\boldsymbol{J}]\,\dot{\boldsymbol{\theta}}_t = -[\boldsymbol{J}^2 + \gamma\boldsymbol{J}]\boldsymbol{\theta}_t + \eta k\,[\boldsymbol{g} + h(\boldsymbol{\theta}_t - \boldsymbol{\theta}_0)]\,. \tag{103}$$

---

[8]Note that we are using boldfaced $\boldsymbol{J}$ to denote the Jacobian of $\boldsymbol{f}$, and $J$ to denote the objective.

All skew-symmetric matrices are unitarily diagonalizable over the complex numbers, so we can write

$$\boldsymbol{J} = \boldsymbol{U}\boldsymbol{\Lambda}\boldsymbol{U}^\dagger \tag{104}$$

where $\boldsymbol{U}$ is unitary (i.e., $\boldsymbol{U}\boldsymbol{U}^\dagger = \boldsymbol{U}^\dagger\boldsymbol{U} = \boldsymbol{I}$) and $\boldsymbol{\Lambda}$ is diagonal with complex entries.

Our second-order ODEs become decoupled in the space of eigenvectors of $\boldsymbol{Q}$. The quantity $\boldsymbol{\phi}_t := \boldsymbol{U}^\dagger\boldsymbol{\theta}_t$ changes according to

$$\ddot{\boldsymbol{\phi}}_t - [\gamma\boldsymbol{I} + 2\boldsymbol{\Lambda}]\,\dot{\boldsymbol{\phi}}_t = -[\boldsymbol{\Lambda}^2 + \gamma\boldsymbol{\Lambda}]\boldsymbol{\phi}_t + \eta k\left[\tilde{\boldsymbol{g}} + h(\boldsymbol{\phi}_t - \boldsymbol{\phi}_0)\right] \tag{105}$$

where we define $\tilde{\boldsymbol{g}} := \boldsymbol{U}^\dagger\boldsymbol{g}$. Happily, each component $\phi_i$ of $\boldsymbol{\phi}_t$ evolves independently according to a linear second-order ODE, and each of these can be solved in the usual way.

**Decoupled ODEs.**    Each $\phi_i$ evolves according to

$$\ddot{\phi}_i - [\gamma + 2\lambda_i]\,\dot{\phi}_i - [\eta k h - \lambda_i^2 - \gamma\lambda_i]\phi_i = \eta k\left[\tilde{g}_i - h\phi_{i0}\right] \tag{106}$$

where $\lambda_i := \Lambda_{ii}$. The characteristic equation of this ODE is

$$r^2 - [\gamma + 2\lambda_i]r - [\eta k h - \lambda_i^2 - \gamma\lambda_i] = 0 \tag{107}$$

and its solution is

$$
\begin{aligned}
r_\pm &= \frac{\gamma + 2\lambda_i}{2} \pm \sqrt{\frac{(\gamma + 2\lambda_i)^2}{4} + \eta k h - \lambda_i^2 - \gamma\lambda_i} \\
&= \lambda_i + \frac{\gamma}{2} \pm \sqrt{\frac{\gamma^2}{4} + \eta k h}\ .
\end{aligned}
\tag{108}
$$

As before (see, e.g., Appendix C and Appendix E), we ignore the positive root since its associated solution asymptotically blows up. Define

$$r_i := \lambda_i + \frac{\gamma}{2} - \sqrt{\frac{\gamma^2}{4} + \eta k h} \tag{109}$$

and $\tilde{\boldsymbol{R}}$ as the diagonal matrix with $\tilde{R}_{ii} = r_i$. Combining the general and particular solutions of Eq. 106 yields

$$\phi_i(t) = \frac{\eta k h}{\eta k h - \lambda_i(\lambda_i + \gamma)}\left[\phi_{i0} - h^{-1}\tilde{g}_i\right] + ce^{r_i t} \tag{110}$$

for some constant $c$. Enforcing the initial condition,

$$\phi_i(t) = \frac{\eta k h}{\eta k h - \lambda_i(\lambda_i + \gamma)}\left[\phi_{i0} - h^{-1}\tilde{g}_i\right](1 - e^{r_i t}) + \phi_{i0}e^{r_i t}\ . \tag{111}$$

Define the diagonal matrix $\tilde{\boldsymbol{B}}$ via

$$\tilde{B}_{ii} := \frac{\eta k h}{\eta k h - \lambda_i(\lambda_i + \gamma)}\ , \tag{112}$$

so that the $\phi_i$ solution becomes

$$\phi_i(t) = (1 - e^{r_i t})\tilde{B}_i\left(\phi_{i0} - h^{-1}\tilde{g}_i\right) + e^{r_i t}\phi_{i0}\ , \tag{113}$$

or in vector form,

$$\boldsymbol{\phi}_t = (\boldsymbol{I} - e^{\tilde{\boldsymbol{R}}t})\tilde{\boldsymbol{B}}\left(\boldsymbol{\phi}_0 - h^{-1}\tilde{\boldsymbol{g}}\right) + e^{\tilde{\boldsymbol{R}}t}\boldsymbol{\phi}_0\ . \tag{114}$$

**Solution.**    The relationship $\boldsymbol{\theta}_t = \boldsymbol{U}\boldsymbol{\phi}_t$ tells us that

$$\boldsymbol{\theta}_t = e^{\boldsymbol{R}t}\boldsymbol{\theta}_0 + (\boldsymbol{I} - e^{\boldsymbol{R}t})\boldsymbol{B}(\boldsymbol{\theta}_0 - h^{-1}\boldsymbol{g}) \tag{115}$$

where we define the matrix $\boldsymbol{R}$ as

$$\boldsymbol{R} := \boldsymbol{U}\tilde{\boldsymbol{R}}\boldsymbol{U}^\dagger = \boldsymbol{J} + \left(\frac{\gamma}{2} - \sqrt{\frac{\gamma^2}{4} + \eta k h}\right)\boldsymbol{I} \tag{116}$$

and $\boldsymbol{B}$ as

$$\boldsymbol{B} := \boldsymbol{U}\tilde{\boldsymbol{B}}\boldsymbol{U}^\dagger = \eta k h \left[\eta k h - \boldsymbol{J}(\boldsymbol{J} + \gamma \boldsymbol{I})\right]^{-1} . \tag{117}$$

Interestingly, in the long-time limit this solution doesn't converge to the global minimum, but to a slightly different location controlled by the 'bias' matrix $\boldsymbol{B}$:

$$\lim_{t \to \infty} \boldsymbol{\theta}_t = \boldsymbol{B}(\boldsymbol{\theta}_0 - h^{-1}\boldsymbol{g}) . \tag{118}$$

An even more interesting feature of this solution is that, since the $r_i$ are complex-valued (through their dependence on the $\lambda_i$, which are pure imaginary for skew-symmetric matrices), they approach their asymptotic value in a spiraling-in fashion. To see this explicitly, consider the particular skew-symmetric matrix

$$\boldsymbol{J} := \begin{pmatrix} 0 & -1 \\ 1 & 0 \end{pmatrix} \tag{119}$$

which is the infinitesimal generator of a counterclockwise rotation in a two-dimensional plane. Then

$$e^{\boldsymbol{J}t} = \cos(t)\boldsymbol{I} + \sin(t)\boldsymbol{J} = \begin{pmatrix} \cos t & -\sin t \\ \sin t & \cos t \end{pmatrix} , \tag{120}$$

i.e., we obtain a matrix that performs a counterclockwise rotation by an angle $t$. The optimal learning trajectory approaches its final value according to

$$e^{\boldsymbol{R}t} = e^{\boldsymbol{J}t}e^{-\left(\sqrt{\frac{\gamma^2}{4} + \eta k h} - \frac{\gamma}{2}\right)t} , \tag{121}$$

which combines decay with rotation.

## G.2 Example: drift term representing weight decay

**Setup.** In this example, assume a general Hessian $\boldsymbol{H}$ and that $\boldsymbol{f}(\boldsymbol{\theta}) = -j\boldsymbol{\theta}$ for some weight decay rate $j > 0$. The EL equations are

$$\ddot{\boldsymbol{\theta}}_t - \gamma\,\dot{\boldsymbol{\theta}}_t = (j - \gamma)j\,\boldsymbol{\theta}_t + \eta k\left[\boldsymbol{g} + \boldsymbol{H}(\boldsymbol{\theta}_t - \boldsymbol{\theta}_0)\right] , \tag{122}$$

or equivalently

$$\ddot{\boldsymbol{\theta}}_t - \gamma\,\dot{\boldsymbol{\theta}}_t - \left[(j - \gamma)j\boldsymbol{I} + \eta k\boldsymbol{H}\right]\boldsymbol{\theta}_t = \eta k\left(\boldsymbol{g} - \boldsymbol{H}\boldsymbol{\theta}_0\right) . \tag{123}$$

We can solve this system of second-order ODEs using the strategy from Appendix E, by using an eigendecomposition $\boldsymbol{H} = \boldsymbol{Q}\boldsymbol{\Lambda}\boldsymbol{Q}^T$. The only differences are that the relevant rates are

$$r_i := \frac{\gamma}{2} - \sqrt{\frac{\gamma^2}{4} + (j - \gamma)j + \eta k\lambda_i} \tag{124}$$

and that we once again get biases

$$b_i := \frac{\eta k\lambda_i}{\eta k\lambda_i + (j - \gamma)j} . \tag{125}$$

Define the diagonal matrix $\tilde{\boldsymbol{R}}$ using the $r_i$, and the diagonal matrix $\tilde{\boldsymbol{B}}$ using the $b_i$. The solution has

$$\boldsymbol{\theta}_t = e^{\boldsymbol{R}t}\boldsymbol{\theta}_0 + (\boldsymbol{I} - e^{\boldsymbol{R}t})\boldsymbol{B}(\boldsymbol{\theta}_0 - \boldsymbol{H}^{-1}\boldsymbol{g}) \tag{126}$$

where we define the matrix $\boldsymbol{R}$ as

$$\boldsymbol{R} := \boldsymbol{Q}\tilde{\boldsymbol{R}}\boldsymbol{Q}^T = \frac{\gamma}{2}\boldsymbol{I} - \sqrt{\left(\frac{\gamma^2}{4} + (j - \gamma)j\right)\boldsymbol{I} + \eta k\boldsymbol{H}} \tag{127}$$

and $\boldsymbol{B}$ as

$$\boldsymbol{B} := \boldsymbol{Q}\tilde{\boldsymbol{B}}\boldsymbol{Q}^T = \frac{\eta k\boldsymbol{H}}{\eta k\boldsymbol{H} + (j - \gamma)j} . \tag{128}$$

As in the previous example, the form of $\boldsymbol{J}$ (here, just the scalar $j$) contributes to both a drift-related bias $\boldsymbol{B}$ and the rate $\boldsymbol{R}$ at which the optimal trajectory approaches its asymptotic value.

### G.3 Optimal learning trajectories generally do not follow gradients

The $G = I$ EL equations above can be written as

$$\dot{p}_t = [\gamma I + J(\theta_t) - J(\theta_t)^T] \, p_t + [J(\theta_t)^T - \gamma I] f(\theta_t) + \eta k \nabla_{\theta_t} \mathcal{L}(\theta_t) \,, \qquad (129)$$

where we define momentum (as before) as $p_t := \dot{\theta}_t$. Assume Helmholtz decomposition $f(\theta) = \nabla_\theta V(\theta) + R(\theta)$, where $V$ is some non-unique 'potential' function and $R$ is divergence-free (i.e., $\nabla_\theta \cdot R = 0$). This decomposition implies that the Jacobian of $f$ has entries

$$J_{ij} = \partial_{ij}^2 V(\theta) + \frac{\partial R_i(\theta)}{\partial \theta_j} \,. \qquad (130)$$

Note that the only possible source of asymmetry comes from $R$. Moreover, note that

$$\sum_j J_{ij}^T f_j = \sum_j \left[ \partial_{ij}^2 V + \frac{\partial R_j}{\partial \theta_i} \right] [\partial_j V + R_j] = \partial_i \left( \sum_j \frac{(\partial_j V)^2}{2} + (\partial_j V) R_j + \frac{R_j^2}{2} \right)$$
$$= \partial_i \left( \sum_j \frac{[\partial_j V + R_j]^2}{2} \right) \,. \qquad (131)$$

If $J_R$ denotes the Jacobian of $R$, the EL equations can be written

$$\dot{p}_t = [\gamma I + J_R(\theta_t) - J_R(\theta_t)^T] \, p_t - \gamma \nabla_{\theta_t} V(\theta_t) - \gamma R(\theta_t) + J(\theta_t)^T f(\theta_t) + \eta k \nabla_{\theta_t} \mathcal{L}(\theta_t)$$
$$= [\gamma I + J_R(\theta_t) - J_R(\theta_t)^T] \, p_t - \gamma R(\theta_t) + \nabla_{\theta_t} V_{eff}(\theta_t)$$

where we define the *effective loss/potential*

$$V_{eff}(\theta_t) := \eta k \mathcal{L}(\theta_t) + \frac{1}{2} \| \nabla_{\theta_t} V(\theta_t) + R \|^2 - \gamma V(\theta_t) \,. \qquad (132)$$

Hence, it is clear that a nontrivial $R$ contributes non-gradient dynamics to learning in two ways: first, through determining an asymmetric effective temporal discounting rate; and second, through the $\gamma R$ term.

# H   Deriving adaptive learning rules from dynamic loss landscape beliefs

In this appendix, we motivate and derive the Adam-like adaptive learning rule discussed in Sec. 5.

## H.1   Motivation: why gradient variance relates to the Hessian

First, we briefly motivate that the (average) loss landscape can be locally approximated as

$$\mathbb{E}[\hat{\mathcal{L}}(\boldsymbol{\theta}_t)] = \mathcal{L}(\boldsymbol{\theta}_0) + \boldsymbol{m}_t^T(\boldsymbol{\theta}_t - \boldsymbol{\theta}_0) + \frac{\kappa}{2}(\boldsymbol{\theta}_t - \boldsymbol{\theta}_0)^T \boldsymbol{V}_t(\boldsymbol{\theta}_t - \boldsymbol{\theta}_0) \tag{133}$$

where $\boldsymbol{m}_t$ is the average observed gradient and $\boldsymbol{V}_t$ is the covariance of these gradient observations. Why is it reasonable to suppose that $\boldsymbol{V}_t$ is proportional to the local Hessian $\boldsymbol{H}$?

Assume that the true loss landscape has a gradient $\boldsymbol{g}$ and Hessian $\boldsymbol{H}$. The idea is to view noise in gradients as due to an unobservable, noisy parameter vector $\tilde{\boldsymbol{\theta}}_t$ that explores the local loss landscape according to a stochastic process. Since $\tilde{\boldsymbol{\theta}}_t$ is noisy, its time derivative (i.e., gradients, which are observable) will also be noisy. We can view this as a change in perspective. Rather than assuming that $\boldsymbol{\theta}_t$ remains fixed but the landscape changes dynamically (and partly randomly) around us, we can assume that the landscape is fixed, but that *where we are in it* is randomly changing.

The simplest process we can assume is one that is unbiased (i.e., noisy gradients equal true gradients on average), and has a structureless (i.e., white and state-independent) noise term. Consider

$$\frac{d}{dt}\tilde{\boldsymbol{\theta}}_t = -\frac{1}{\tau}\nabla_{\tilde{\boldsymbol{\theta}}_t}\mathcal{L} + \sigma\boldsymbol{\eta}_t = -\frac{1}{\tau}\left[\boldsymbol{g} + \boldsymbol{H}(\tilde{\boldsymbol{\theta}}_t - \boldsymbol{\theta}_0)\right] + \sigma\boldsymbol{\eta}_t = \frac{1}{\tau}\boldsymbol{H}\left[\boldsymbol{\mu} - \tilde{\boldsymbol{\theta}}_t\right] + \sigma\boldsymbol{\eta}_t \tag{134}$$

where $\tau > 0$ is a decay time scale, $\sigma > 0$ controls the amount of noise, $\boldsymbol{\eta}_t$ is a Gaussian white noise term, and

$$\boldsymbol{\mu} := \boldsymbol{\theta}_0 - \boldsymbol{H}^{-1}\boldsymbol{g} . \tag{135}$$

At steady state (or at least, on time scales somewhat longer than $\tau$), we have

$$\tilde{\boldsymbol{\theta}} \sim \mathcal{N}(\boldsymbol{\mu}, \frac{\sigma^2}{2}\boldsymbol{H}^{-1}) . \tag{136}$$

Near steady state, this implies that

$$\mathbb{E}\left\{(\nabla_{\tilde{\boldsymbol{\theta}}_t}\mathcal{L})(\nabla_{\tilde{\boldsymbol{\theta}}_t}\mathcal{L})^T\right\} = \frac{1}{\tau^2}\boldsymbol{H}\,\mathbb{E}\left\{(\boldsymbol{\mu} - \tilde{\boldsymbol{\theta}}_t)(\boldsymbol{\mu} - \tilde{\boldsymbol{\theta}}_t)^T\right\}\boldsymbol{H} = \frac{\sigma^2}{2\tau^2}\boldsymbol{H} \tag{137}$$

or equivalently that the variance of observed gradients is proportional to $\boldsymbol{H}$. This result can be interpreted in the following way. For sharp/curved minima, $\boldsymbol{H}$ is by definition large, so small parameter changes can produce relatively large changes in gradients; on the other hand, in flat loss landscape regions, $\boldsymbol{H}$ is small, so even large parameter changes do not change gradients much.

## H.2   Deriving the Euler-Lagrange equations

**More complex observation model.**   Assume that $\boldsymbol{g}_t \sim \mathcal{N}(\boldsymbol{m}_t, \boldsymbol{V}_t/\Delta t)$. The relevant objective is

$$J(\{\boldsymbol{\theta}_t, \boldsymbol{m}_t, \boldsymbol{v}_t\}) = \lim_{\Delta t \to 0}\int_0^\infty \left(\frac{\|\boldsymbol{\theta}_t\|^2}{2\eta} - \frac{\log p(\boldsymbol{m}_{t+1}, \boldsymbol{V}_{t+1}|\boldsymbol{g}_t, \boldsymbol{m}_t, \boldsymbol{V}_t)}{\Delta t} + k\mathbb{E}[\hat{\mathcal{L}}(\boldsymbol{\theta}_t)]\right)e^{-\gamma t}dt$$

where $p(\boldsymbol{m}_{t+1}, \boldsymbol{V}_{t+1}|\boldsymbol{g}_t, \boldsymbol{m}_t, \boldsymbol{V}_t)$ is the learner's posterior belief about local landscape shape dynamics. The posterior term contains two types of terms: terms from the observation model, and terms from the prior. In particular, it contains the terms

$$\sum_i \frac{(g_i - m_i)^2}{2v_i} + \frac{1}{2}\log(2\pi v_i) + \frac{\|\dot{\boldsymbol{m}}_t + \alpha_1\boldsymbol{m}_t\|^2}{2\xi_1^2} + \frac{\|\dot{\boldsymbol{v}}_t + \alpha_2\boldsymbol{v}_t\|^2}{2\xi_2^2} + \text{const.} \tag{138}$$

where we neglect unimportant additive constants, and assume that $\boldsymbol{V}_t$ is diagonal. Here, $v_i$ denotes $V_{ii}$, and $\boldsymbol{v}$ denotes the vector containing the $v_i$. The corresponding Lagrangian is

$$L := \sum_i \left(\frac{\dot{\theta}_i^2}{2\eta} + \frac{(g_i - m_i)^2}{2v_i} + \frac{1}{2}\log(2\pi v_i) + \frac{(\dot{m}_i + \alpha_1 m_i)^2}{2\xi_1^2} + \frac{(\dot{v}_i + \alpha_2 v_i)^2}{2\xi_2^2} + k\mathbb{E}[\hat{\mathcal{L}}(\boldsymbol{\theta}_t)]\right)e^{-\gamma t} ,$$

and the corresponding EL equations are

$$\ddot{\theta}_i - \gamma\dot{\theta}_i = \eta k \left[ g_i + \kappa v_i(\theta_i - \theta_{0i}) \right]$$

$$\ddot{m}_i - \gamma(\dot{m}_i + \alpha_1 m_i) = \alpha_1^2 m_i + \xi_1^2 \left[ k(\theta_i - \theta_{0i}) + \frac{(m_i - g_i)}{v_i} \right] \tag{139}$$

$$\ddot{v}_i - \gamma(\dot{v}_i + \alpha_2 v_i) = \alpha_2^2 v_i + \xi_2^2 \left[ k\frac{\kappa}{2}(\theta_i - \theta_{0i})^2 + \frac{1}{2}\frac{1}{v_i} - \frac{1}{2}\frac{(g_i - m_i)^2}{2v_i^2} \right] .$$

The equation for $v_i$ is somewhat more complicated than the one presented in the main text, but note that it can be written as

$$\ddot{v}_i - \gamma(\dot{v}_i + \alpha_2 v_i) = \alpha_2^2 v_i + \xi_2^2 \left[ k\frac{\kappa}{2}(\theta_i - \theta_{0i})^2 + \frac{1}{2}\frac{1}{v_i^2}\left(v_i - (g_i - m_i)^2\right) \right] , \tag{140}$$

which matches the main text form up to the prefactor $1/(2v_i^2)$. If the variance $v_i$ is fairly stable (for example, because a reasonable amount of evidence about gradients has already been accumulated), then this prefactor is approximately constant, and the forms are identical.

**Simplified observation model.** If we instead treat gradients and squares of gradients as consisting of separate observations, i.e., $\boldsymbol{g}_t \sim \mathcal{N}(\boldsymbol{m}_t, (\sigma_1^2/\Delta t)\boldsymbol{I})$ and $(\boldsymbol{g}_t - \boldsymbol{m}_t)(\boldsymbol{g}_t - \boldsymbol{m}_t)^T \sim \mathcal{N}(\boldsymbol{V}_t, (\sigma_2^2/\Delta t)\boldsymbol{I})$, then the posterior contains the terms

$$\sum_i \frac{(g_i - m_i)^2}{2\sigma_1^2} + \frac{[v_i - (g_i - m_i)^2]^2}{2\sigma_2^2} + \frac{\|\dot{\boldsymbol{m}}_t + \alpha_1\boldsymbol{m}_t\|^2}{2\xi_1^2} + \frac{\|\dot{\boldsymbol{v}}_t + \alpha_2\boldsymbol{v}_t\|^2}{2\xi_2^2} + \text{const.} \tag{141}$$

The corresponding Lagrangian is

$$L := \sum_i \left( \frac{\dot{\theta}_i^2}{2\eta} + \frac{(g_i - m_i)^2}{2\sigma_1^2} + \frac{[v_i - (g_i - m_i)^2]^2}{2\sigma_2^2} + \frac{(\dot{m}_i + \alpha_1 m_i)^2}{2\xi_1^2} + \frac{(\dot{v}_i + \alpha_2 v_i)^2}{2\xi_2^2} + k\mathbb{E}[\hat{\mathcal{L}}(\boldsymbol{\theta}_t)] \right) e^{-\gamma t} ,$$

and the corresponding EL equations are

$$\ddot{\theta}_i - \gamma\dot{\theta}_i = \eta k \left[ g_i + \kappa v_i(\theta_i - \theta_{0i}) \right]$$

$$\ddot{m}_i - \gamma(\dot{m}_i + \alpha_1 m_i) = \alpha_1^2 m_i + \xi_1^2 \left[ k(\theta_i - \theta_{0i}) + \frac{(m_i - g_i)}{\sigma_1^2} \right] \tag{142}$$

$$\ddot{v}_i - \gamma(\dot{v}_i + \alpha_2 v_i) = \alpha_2^2 v_i + \xi_2^2 \left[ k\frac{\kappa}{2}(\theta_i - \theta_{0i})^2 + \frac{1}{\sigma_2^2}\left(v_i - (g_i - m_i)^2\right) \right] .$$

# I Deriving learning rules sensitive to weight uncertainty

In this appendix, we derive the weight-uncertainty-sensitive learning rule discussed in Sec. 6. Recall that we assume each model parameter $\theta_i$ (for $i = 1, ..., D$) is associated with a normal distribution $\mathcal{N}(\mu_i, v_i)$, where $\mu_i$ is the average value of $\theta_i$, and $v_i$ is its variance. The advantage of this setup is that it allows the learner to not just estimate what their parameters are, but also how certain they are about them. In principle, we could consider a model with a more general covariance matrix, but we restrict ourselves to a diagonal covariance for simplicity.

## I.1 Simplifying the objective

The relevant objective is

$$J(\{\boldsymbol{\mu}_t, \boldsymbol{v}_t\}) = \lim_{\Delta t \to 0} \int_0^\infty \left[ \frac{D_{KL}(p(\boldsymbol{\theta}|\boldsymbol{\mu}_{t+1}, \boldsymbol{v}_{t+1}) \| p(\boldsymbol{\theta}|\boldsymbol{\mu}_t, \boldsymbol{v}_t))}{\eta(\Delta t)^2} - \mathcal{H}(p(\boldsymbol{\theta}|\boldsymbol{\mu}_t, \boldsymbol{v}_t)) + k\mathcal{L}(\boldsymbol{\mu}_t, \boldsymbol{v}_t) \right] e^{-\gamma t} dt$$

where the first term does not penalize abrupt parameter changes, but abrupt changes in parameter distribution. Note that the Kullback-Leibler (KL) divergence term can be written

$$
\begin{aligned}
& D_{KL}(p(\boldsymbol{\theta}|\boldsymbol{\mu}_t + \dot{\boldsymbol{\mu}}_t\Delta t, \boldsymbol{v}_t + \dot{\boldsymbol{v}}_t\Delta t) \| p(\boldsymbol{\theta}|\boldsymbol{\mu}_t, \boldsymbol{v}_t)) \\
&= \mathbb{E}_{\boldsymbol{\theta}} \left\{ \log p(\boldsymbol{\theta}|\boldsymbol{\mu}_t + \dot{\boldsymbol{\mu}}_t\Delta t, \boldsymbol{v}_t + \dot{\boldsymbol{v}}_t\Delta t) - \log p(\boldsymbol{\theta}|\boldsymbol{\mu}_t, \boldsymbol{v}_t) \right\} \\
&= \sum_i \mathbb{E}_{\theta_i} \left\{ -\frac{[\theta_i - \mu_i - \dot{\mu}_i\Delta t]^2}{2(v_i + \dot{v}_i\Delta t)} - \frac{1}{2}\log[2\pi(v_i + \dot{v}_i\Delta t)] + \frac{[\theta_i - \mu_i]^2}{2v_i} + \frac{1}{2}\log[2\pi v_i] \right\} \\
&= \sum_i -\frac{1}{2} - \frac{1}{2}\log[2\pi(v_i + \dot{v}_i\Delta t)] + \frac{v_i + \dot{v}_i\Delta t}{2v_i} + \frac{\dot{\mu}_i^2}{2v_i}(\Delta t)^2 + \frac{1}{2}\log[2\pi v_i] \\
&= \sum_i \frac{\dot{v}_i\Delta t}{2v_i} + \frac{\dot{\mu}_i^2}{2v_i}(\Delta t)^2 - \frac{1}{2}\log\left(1 + \frac{\dot{v}_i}{v_i}\Delta t\right) \\
&\approx \sum_i \left( \frac{1}{2}\frac{\dot{\mu}_i^2}{v_i} + \frac{1}{4}\frac{\dot{v}_i^2}{v_i^2} \right)(\Delta t)^2
\end{aligned}
$$

and that the entropy term is

$$\mathcal{H}(p(\boldsymbol{\theta}|\boldsymbol{\mu}_t, \boldsymbol{v}_t)) = \sum_i \frac{1}{2}\log(2\pi e v_i) , \tag{143}$$

which means that our objective is effectively

$$J(\{\boldsymbol{\mu}_t, \boldsymbol{v}_t\}) = \int_0^\infty \left[ \sum_i \frac{1}{2\eta}\frac{\dot{\mu}_i^2}{v_i} + \frac{1}{4\eta}\frac{\dot{v}_i^2}{v_i^2} - \frac{1}{2}\log(2\pi e v_i) + k\mathcal{L}(\boldsymbol{\mu}_t, \boldsymbol{v}_t) \right] e^{-\gamma t} dt$$

and the corresponding Lagrangian is

$$L(\boldsymbol{\mu}, \boldsymbol{v}, \dot{\boldsymbol{\mu}}, \dot{\boldsymbol{v}}, t) = \left[ \sum_i \frac{1}{2\eta}\frac{\dot{\mu}_i^2}{v_i} + \frac{1}{4\eta}\frac{\dot{v}_i^2}{v_i^2} - \frac{1}{2}\log(2\pi e v_i) + k\mathcal{L}(\boldsymbol{\mu}_t, \boldsymbol{v}_t) \right] e^{-\gamma t} .$$

Here, the effective metric is the Fisher information metric for a normal distribution.

## I.2 Simplifying the Euler-Lagrange equations

Taking derivatives, the EL equations read

$$
\begin{aligned}
\frac{d}{dt}\left(\frac{\dot{\mu}_i}{v_i}\right) - \gamma\frac{\dot{\mu}_i}{v_i} &= \eta k\frac{\partial\mathcal{L}(\boldsymbol{\mu}, \boldsymbol{v})}{\partial\mu_i} \\
\frac{d}{dt}\left(\frac{1}{2}\frac{\dot{v}_i}{v_i^2}\right) - \gamma\frac{1}{2}\frac{\dot{v}_i}{v_i^2} &= \eta k\frac{\partial\mathcal{L}(\boldsymbol{\mu}, \boldsymbol{v})}{\partial v_i} - \frac{\eta}{2}\frac{1}{v_i} - \frac{\dot{\mu}_i^2}{2v_i^2} - \frac{\dot{v}_i^2}{2v_i^3} .
\end{aligned}
\tag{144}
$$

Simplifying, these become

$$\ddot{\mu}_i - \left[\frac{\dot{v}_i}{v_i} + \gamma\right]\dot{\mu}_i = \eta k v_i \frac{\partial \mathcal{L}(\boldsymbol{\mu}, \boldsymbol{v})}{\partial \mu_i}$$

$$\ddot{v}_i - \gamma \dot{v}_i = 2\eta \left[k v_i^2 \frac{\partial \mathcal{L}(\boldsymbol{\mu}, \boldsymbol{v})}{\partial v_i} - \frac{1}{2} v_i\right] + \frac{\dot{v}_i^2}{v_i} - \dot{\mu}_i^2 \ . \tag{145}$$

Consider a local (quadratic) approximation of the loss, i.e.,

$$\hat{\mathcal{L}}(\boldsymbol{\theta}_t) := \mathcal{L}(\boldsymbol{\theta}_0) + \boldsymbol{g}^T(\boldsymbol{\theta}_t - \boldsymbol{\theta}_0) + \frac{1}{2}(\boldsymbol{\theta}_t - \boldsymbol{\theta}_0)^T \boldsymbol{H}(\boldsymbol{\theta}_t - \boldsymbol{\theta}_0) \ , \tag{146}$$

where $\boldsymbol{g}$ is the local gradient and $\boldsymbol{H}$ is the local Hessian. Averaging this quantity over $\boldsymbol{\theta}_t$ and $\boldsymbol{\theta}_0$,

$$\mathcal{L}(\boldsymbol{\mu}, \boldsymbol{v}) := \mathbb{E}_{\boldsymbol{\theta}_t, \boldsymbol{\theta}_0}\{\hat{\mathcal{L}}(\boldsymbol{\theta}_t)\} := \boldsymbol{g}^T(\boldsymbol{\mu}_t - \boldsymbol{\mu}_0) + \frac{1}{2}(\boldsymbol{\mu}_t - \boldsymbol{\mu}_0)^T \boldsymbol{H}(\boldsymbol{\mu}_t - \boldsymbol{\mu}_0) + \sum_i \frac{1}{2} H_{ii} v_i + \text{const.}$$

where 'const.' denotes terms we can ignore. Using this particular $\mathcal{L}(\boldsymbol{\mu}, \boldsymbol{v})$, the EL equations become

$$\ddot{\mu}_i - \left[\frac{\dot{v}_i}{v_i} + \gamma\right]\dot{\mu}_i = \eta k v_i \left[g_i + \sum_j H_{ij}(\mu_j - \mu_{j0})\right]$$

$$\ddot{v}_i - \gamma \dot{v}_i = 2\eta \left[\frac{k}{2} H_{ii} v_i^2 - \frac{1}{2} v_i\right] + \frac{\dot{v}_i^2}{v_i} - \dot{\mu}_i^2 \ , \tag{147}$$

which are the equations that appear in the main text.

### I.3  The Euler-Lagrange equations in the overdamped limit

The EL equations simplify somewhat in the overdamped (large $\gamma$) limit[9]:

$$\dot{\mu}_i \approx -\frac{\eta k v_i}{\gamma}\left[g_i + \sum_j H_{ij}(\mu_j - \mu_{j0})\right]$$

$$\dot{v}_i \approx \frac{\eta}{\gamma}\left[v_i - k H_{ii} v_i^2\right] \ . \tag{148}$$

Note that, in this limit, $v_i$ influences how $\mu_i$ evolves (by modulating the effective learning rate), but $v_i$ evolves independently of $\mu_i$, at least on time scales where the quadratic loss landscape approximation remains valid. The entropic term $v_i$ tends to increase the variance, and the loss gradient term tends to decrease the variance; they balance when

$$v_i = \frac{1}{k H_{ii}} \ , \tag{149}$$

which implies that narrower loss landscape basins (high $H_{ii}$) produce small uncertainties, and broader basins (low $H_{ii}$) produce high uncertainties. This is intuitively reasonable, and reflects the selection of the 'simplest' model compatible with the data. Meanwhile, when the entropic term is absent, in this limit $v_i \to 0$.

Suppose that the learner has converged to a 'good' (i.e., deep) local or global minimum. When there is a task transition, one generally expects the local landscape to no longer be as curved (since the learner is probably no longer in a good local minimum), which means that $H_{ii}$ suddenly decreases. Eq. 148 says that optimal learning dynamics involves a sudden increase in variance, which persists until $v_i$ equilibrates to its new value. By Eq. 149, this value is proportional to the new $1/H_{ii}$.

---

[9]To justify this reduction more rigorously, we could have performed a singular perturbation analysis. This analysis is not particularly enlightening, so we merely report the result.

## I.4 Qualitative behavior outside of the overdamped limit

Outside of the overdamped limit, the EL equations for $\mu_i$ and $v_i$ influence each other: changes in $\mu_i$ contribute an effective 'force' that affects $v_i$, and $v_i$ affects both the effective discounting rate and learning rate in the $\mu_i$ equation. Because the $\dot{\mu}_i^2$ term has the same sign as the entropic term, it plays the same qualitative role, and can produce increases in variance.

Again suppose that there is a task transition after the learner has converged to a 'good' minimum, so that $H_{ii}$ suddenly decreases. Optimal learning dynamics again involves a sudden increase in variance, this time driven by both the entropic term and the $\dot{\mu}_i^2$ term.

Unlike in the overdamped case, if $H_{ii}$ does not change after a task transition but $g_i$ *does* (i.e., the location, but not size, of the basin has changed), the $\dot{\mu}_i^2$ produces a transient increase in variance. After $\mu_i$ equilibrates, the $\dot{\mu}_i^2$ term becomes zero, and $v_i$ approaches the same equilibrium value as in the overdamped case (Eq. 149).

