# OpenReview forum: "Gradient Descent as Loss Landscape Navigation: a Normative Framework for Deriving Learning Rules"
_NeurIPS.cc/2025/Conference — NeurIPS 2025 poster_

### Official Review · Reviewer_KkuV · 2025-06-30

**Clarity:** 4
**Significance:** 4
**Originality:** 4
**Rating:** 6
**Confidence:** 4

**Summary:**

Different learning rules, such as variants of gradient descent, are often compared empirically. Theoretical analysis generally looks at convergence rates or guarantees. This paper takes a slightly different perspective and tries to understand what problem each rule is implicitly solving. By framing parameter updates as a multi-step, continuous-time optimal control problem, the authors find they can derive many popular learning rules from a common framework. The objective for deriving the parameter updates is composed of two terms: a parameter change cost and the loss function. Different choices of the parameter change cost and approximations of the loss function give us different learning rules. Common tricks, like momentum, naturally fall out of the framework due to the multi-step optimization. Using a second-order approximation of the loss can give Netwon's method, gradient descent, or an in-between "ballistic" learning rule depending on the choice of time step and temporal discount factor. Incorporating different metrics in the parameter change cost can respect parameter geometry. For instance, using the Fisher information matrix gives us natural gradient descent. An important take-away from this analysis is that natural gradient descent is not actually second-order optimization. Rather, the metric governs the parameter velocity penalty and the Hessian measures loss landscape curvature. Making assumptions about partial controllability of parameters, such as existing external "forces" on parameters that may act in a state-dependent way, can produce non-gradient rules, such as rotational dynamics, and account for weight decay. Next, assuming the true loss is only partially observable and must be estimated, they show that adaptive optimizers, like Adam and RMSprop, can be derived. Finally, they show that some common continual learning strategies can fall out of modeling weight uncertainty and making the learning dynamics sensitive to this uncertainty.

**Questions:**

- Are there more subtle differences between Wibisono et al. [49] and the formulation considered in this paper, when not considering partial observability or controllability?
- What are some situations in which one would want to use the ballistic version of the learning rule? What are some trade-offs of these different choices besides Hessian computation?

**Ethical Concerns:**

["NO or VERY MINOR ethics concerns only"]

**Limitations:**

- The authors are clear that their analysis does not consider comptuational requirements and efficiency in its definition of optimal.

**Quality:**

3

**Strengths And Weaknesses:**

Strengths:
- Building a better theoretical understanding of what implicit assumptions different learning rules make and when to use them is an important problem. This paper provides a new perspective on this problem, deriving many different learning rules under a common framework.
- New learning rules can possibly be derived from this framework, making different assumptions on parameter geometry, partial controllability, and partial observability. In fact, the ballistic learning rules appear to be novel, although it is not evaluated empirically.
- The insight on the relationship between natural gradient descent and second order methods appears to be novel and makes a valuable distinction between the two.
- The view of adaptive learning rules as modeling partial observability also appears novel and provides a nice explanation for what these algorithms are doing under the hood.
- The discussion of continual learning algorithms, which may inject noise or reset unused weights, as being tied to a weight-uncertainty-aware learning rule is a nice justification of choices that are often empirically motivated.

Weaknesses:
- While the paper is primarily focused on theory, and is deriving known algorithms from a new perspective, there are some new variants of common learning rules put forth. It would be interesting and valuable to compare these different choices on some simple examples to better understand what trade-offs they make.
- The framing of momentum as falling out of a continuous-time formulation appears in Wibisono et al. [49], which is briefly mentioned in the related work section. While this work does not consider partial observability, partial controllability, weight uncertainty, or second-order methods, it does account for momentum falling out of a multi-step perspective on optimization. Moreover, it uses the Bregman divergence as a distance metric. Natural gradient descent can be viewed as gradient descent with a specific choice of Bregman divergence [1, 2]. Therefore, while I do think the contributions of this paper are substantial and novel, the paper could do a better job relating its innovations to this prior work and making the distinctions clear.

References:
[1] G. Raskutti and S. Mukherjee, "The information geometry of mirror descent," IEEE Transactions on Information Theory, 2015.
[2] N. Wagener, C. Cheng, J. Sacks, and B. Boots, "An online learning approach to model predictive control," RSS, 2019.

---

> ### Author Rebuttal · Authors · 2025-07-29
>
> We thank the reviewer for their support and useful feedback, which we think has greatly strengthened the paper. See below for detailed responses.
>
> **What is the relationship between our framework and the results from Wibisono et al.?**
>
> > The framing of momentum as falling out of a continuous-time formulation appears in Wibisono et al. [49], which is briefly mentioned in the related work section. While this work does not consider partial observability, partial controllability, weight uncertainty, or second-order methods, it does account for momentum falling out of a multi-step perspective on optimization. Moreover, it uses the Bregman divergence as a distance metric. Natural gradient descent can be viewed as gradient descent with a specific choice of Bregman divergence [1, 2]. Therefore, while I do think the contributions of this paper are substantial and novel, the paper could do a better job relating its innovations to this prior work and making the distinctions clear.
>
> > Are there more subtle differences between Wibisono et al. [49] and the formulation considered in this paper, when not considering partial observability or controllability?
>
> We agree that the relationship between our work and the work of Wibisono et al. could be made clearer, and **we will explicitly address this issue in the revised version by including a new appendix section**. In brief:
> - There is a *key conceptual difference* between our work and that of Wibisono et al. Our variational problem can be viewed as optimal loss landscape navigation, while theirs can't.
> - Their use of the Bregman divergence makes their Lagrangian more general in some ways, although, as you point out, our framework is more general in different ways (partial observability, partial controllability, weight uncertainty, second-order methods...).
>
> The first difference we mention is fairly interesting. To understand it, recall that our objective (ignoring partial controllability and observability, and explicit time-dependence in the loss) involves the Lagrangian
>
> $$ L(\mathbf{x}, \dot{\mathbf{x}}, t) = e^{- \gamma t} \left\\{  \frac{1}{2} \dot{\mathbf{x}}^T \mathbf{G}(\mathbf{x}, t) \dot{\mathbf{x}} + k \mathcal{L}(\mathbf{x})  \right\\} .$$
>
> Meanwhile, if we replace some of their variable labels with ours so that comparison is easier, their objective involves the Lagrangian
>
> $$L(\mathbf{x}, \dot{\mathbf{x}}, t) = e^{\alpha_t + \gamma_t} \left\\{  D_h(  \mathbf{x} + e^{-\alpha_t} \dot{\mathbf{x}}, \mathbf{x} )- e^{\beta_t} \mathcal{L}(\mathbf{x}) \right\\} $$
>
> where $D_h(\mathbf{y}, \mathbf{x})$ denotes a Bregman divergence, and where $\gamma_t$ has a slightly different meaning than in our Lagrangian. The use of the Bregman divergence is one obvious difference between our framework and theirs, although it is not crucial, since we could easily have *also* used it. (In the new aforementioned appendix, we will talk about what changes when we do this.)
>
> The more interesting difference is that their objective involves a *minus sign* (i.e., it has the form kinetic minus potential) whereas ours involves a *plus sign* (i.e., its form is kinetic plus potential).
>
> Why does this matter? With the plus sign, each increment $L( \mathbf{x}_t, \dot{\mathbf{x}}_t, t) \Delta t$ that contributes to the overall objective $J$ is can be assumed to be nonnegative (if we shift the loss so that it equals zero at its global minimum), and more importantly these increments have a simple interpretation. The first term represents the 'cost' of movement, while the second represents the 'cost' of the current loss value. The total cost of any single parameter update is the sum of these two.
>
> This fact implies that our framework's objective has the appealing interpretation that one is trying to navigate a loss landscape as efficiently as possible. The Wibisono et al. Lagrangian does not have the same interpretation, since the cost of movement and the cost incurred by the loss have different signs.
>
> One might expect that this subtle sign difference does not matter, and that one can recover our Lagrangian from theirs by some reparameterization, but this is not true. It turns out that the sign difference fundamentally changes the nature of the solutions one obtains. This is easiest to see in the simplest possible case, where
> - we take $\alpha_t = \beta_t = \gamma_t = 0$ in their Lagrangian, and we use $D_h(\mathbf{y},\mathbf{x}) = \Vert \mathbf{y} - \mathbf{x} \Vert_2^2$; and
> - we take $\gamma_t = 0$ and $\mathbf{G}$ to be the identity in our Lagrangian.
>
> Assuming a quadratic loss, one obtains two simple Lagrangians (quadratic kinetic term, quadratic potential) that differ only via the aforementioned sign difference. In this case, their Lagrangian yields sinusoidal solutions, while ours yields hyperbolic sines and cosines (or equivalently $e^{r t}$ and $e^{- r t}$ for some real $r$). These are very different!
>
>
> **When would one want to use the ballistic learning rule? More generally, when are different versions of the proposed learning algorithms useful?**
>
> > While the paper is primarily focused on theory, and is deriving known algorithms from a new perspective, there are some new variants of common learning rules put forth. It would be interesting and valuable to compare these different choices on some simple examples to better understand what trade-offs they make.
>
> > What are some situations in which one would want to use the ballistic version of the learning rule? What are some trade-offs of these different choices besides Hessian computation?
>
> This is a great question, and we agree that we have not clearly addressed this in the current version of the paper. **In the revised version, we will include simple experiments that empirically compare different algorithms, including the ballistic one.** In the simplest case, we will compare the algorithms for a toy quadratic loss. We will go on to compare them for simple image classification problems (like MNIST and CIFAR-10). We think this is appropriate, since these kinds of simple tasks are already used to benchmark algorithms like Adam.
>
> What do we expect to see? Theoretically, the ballistic algorithm is optimal when one prioritizes good performance in the distant future (low temporal discounting, $\gamma \approx 0$). Hence, we might expect the algorithm to take steps that are costlier in the short-term, but better (in terms of minimizing the loss) in the long-term.
>
> Consider the quadratic loss case. If $\lambda$ denotes an eigenvalue of the Hessian, Newton's method scales like $1/\lambda$ along a given eigendirection. The ballistic method scales like $1/\sqrt{\lambda}$ along the same direction, which can be both bigger (if $\lambda > 1$) and smaller (if $\lambda < 1$). Hence, it can make the learning rate both faster and slower along a given direction than if one followed Newton's method. The larger consequence of this is that although some directions will converge more slowly, overall, the global minimum will be reached more quickly.
>
> On the other hand, it is not obvious how this rule (which was derived as optimal assuming a simple quadratic loss) will perform given the highly non-convex loss landscapes associated with more realistic problems. One of the papers we cite ("Can We Remove the Square-Root in Adaptive Gradient Methods? A Second-Order Perspective" by Lin et al.) claims that removing the square root of Adam (which our framework identifies as related to the ballistic rule) actually *boosts* performance, which is contrary to the intuition we present here. There are probably many factors that make the real story more complicated.
>
> It is worth emphasizing that, although we find our framework theoretically interesting and helpful for thinking about learning rules, we agree with the claim of another reviewer that the ultimate test for any learning rule is a real experiment on the kind of task one is actually interested in (e.g., training some neural network to do image classification).

---

### Official Review · Reviewer_AiaD · 2025-07-02

**Clarity:** 3
**Significance:** 4
**Originality:** 4
**Rating:** 6
**Confidence:** 4

**Summary:**

The paper provides a general framework to model learning dynamics. The focus is not a parameter that minimize a loss, but rather a trajectory (on parameter space) that minimize a novel objective: an integral over the trajectory of a combination of loss and parameter change. The optimal solution is given by the Euler Lagrange equations providing two boundary conditions: an initial parameter, and the integral to be minimized.

Different hyperparameters characterize the weight of these two components, as well as their specific form. Importantly, for some specific choices of hyperparameters, the optimal solution can be expressed analitically, and the discrete time difference along such optimal trajectory corresponds to learning rules prescribed by established optimizers.
This motivates the relevance of this novel framework: it can recover and generalize multiple optimizers.

**Questions:**

Can the authors elaborate on the second boundary condition of the Euler Lagrange solutions?

In Line 1099 you state that Equation (133) "matches the main text form up to the prefactor $1/(2v_i^2$". Referring to the equation before line 231 in the main text. Can the authors expand on this mismatch? Why is there such difference? And how is the equation in the main text derived?

Can the authors provide an example where rotational forces exists?

**Ethical Concerns:**

["NO or VERY MINOR ethics concerns only"]

**Limitations:**

yes

**Paper Formatting Concerns:**

I didn't notice any major formatting issues.

**Quality:**

3

**Strengths And Weaknesses:**

The paper is well written an easy to read. The various elements are properly discussed and the Appendix provides several useful examples and a clear derivation of the results.

The proposed framework is, to my knowledge, novel. The derivation are correct and multiple learning rules are derived from the same framework. For this reason I consider this work very relevant and useful for the community.

What I personally find extremely interesting is the analysis in Section 4, where the loss is assumed to be quadratic and the parameter geometry is given by the Fisher. This lead to a proper distinction of Hessian and Generalized Gauss Newton which, as the authors also notice, are too often simply viewed as surrogates of each other. The authors framework provides a rigorous derivation of their role.

I found the assumption in Section 5 on gradient noise to be a bit arbitrary. It would be very helpful if the authors can show some references motivating (at least partially) this assumption. Aren't there other works that motivates Adam through this kind of assumption?

I like the presence of the $f$ term in the general form, but I personally don't see how the rotational term can be of any use in practice. Is there a practical use case that can be shown?

Overall, I really like the proposed framework and the approach of finding solutions through Euler Lagrange. My only concern (which is actually my only main concern about the paper) is on the boundary conditions. First of all, this part is a bit swiped under the carpet in the Appendix, and I don't really like that. The authors should be more clear about this.
Then, more specifically, no problem about the initial parameter condition, that's very reasonable. My concern is about "The remaining degree of freedom" that "is chosen so that J is minimized". This feels circular. Isn't the whole point of Euler Lagrange solution to minimize J? My understanding is that Euler Lagrange provides LOCALLY optimal solutions, i.e. optimal among small perturbations. How is this compatible with your choice of boundary condition? Please elaborate on this.

Some minor details:

-In Line 42 I'd recommend changing "sensitive to structure in parameter space" to "sensitive to non-local structure in parameter space" to better clarify what is actually being generalized of the natural gradient descent methods

-In Line 855 there is a typo: "J to minimized" should be changed in "J to be minimized"

-In Equations 123 and 124, isn't that $J$ supposed to be a $\mathbf{J}$? Is it a typo or am I missing something

---

> ### Author Rebuttal · Authors · 2025-07-29
>
> We thank the reviewer for their support, close reading, and detailed comments. We think these comments have greatly helped strengthen the paper. Let's begin by addressing your most major concern.
>
> **On boundary conditions.**
>
> > Can the authors elaborate on the second boundary condition of the Euler Lagrange solutions?
>
> > Overall, I really like the proposed framework and the approach of finding solutions through Euler Lagrange. My only concern (which is actually my only main concern about the paper) is on the boundary conditions. First of all, this part is a bit swiped under the carpet in the Appendix, and I don't really like that. The authors should be more clear about this. Then, more specifically, no problem about the initial parameter condition, that's very reasonable. My concern is about "The remaining degree of freedom" that "is chosen so that J is minimized". This feels circular. Isn't the whole point of Euler Lagrange solution to minimize J? My understanding is that Euler Lagrange provides LOCALLY optimal solutions, i.e. optimal among small perturbations. How is this compatible with your choice of boundary condition? Please elaborate on this.
>
> This is indeed a subtle point, and we agree that it would be helpful to clarify it in the paper and SI. **We will add an extra appendix section to discuss this issue explicitly.**
>
> Our most general objective (Eq. 2) involves expectations, but once these are evaluated, one ends up with an objective of the form
>
> $$ J[ \mathbf{x}(t) ] = \int_0^{\infty} F[ \mathbf{x}(t), \dot{\mathbf{x}}(t), t ] \ dt = \int_0^{\infty} \frac{1}{2} \left[ \dot{\mathbf{x}}(t) - \mathbf{f}(\mathbf{x}(t), t) \right]^T \mathbf{G}(\mathbf{x}(t), t) \left[ \dot{\mathbf{x}}(t) - \mathbf{f}(\mathbf{x}(t), t) \right] e^{- \gamma t} + k \mathcal{L}(\mathbf{x}(t), t) e^{- \gamma t} \ dt$$
> where the scalar-valued function $F$ is smooth, and moreover equals a 'kinetic' term plus a 'loss' term. We would like to minimize $J$ over all possible smooth trajectories $\mathbf{x}(t)$ with a fixed initial point $\mathbf{x}(0) = \mathbf{x}_0$.
>
> The idea is to optimize $J$ via a *two-step* optimization: we find the function $\mathbf{x}(t)$ that minimizes $J$ for a specified initial point $\mathbf{x}(0)$ and final point $\mathbf{x}(\infty)$, and *then* (since our initial problem only prescribes $\mathbf{x}(0)$) we choose the solution whose final point corresponds to the smallest value of $J$. Note that the latter minimization—over a finite set of $D$ numbers $\mathbf{x}(\infty) = ( x_1(\infty), ..., x_D(\infty) )^T$—is different than the more challenging optimization over a function space that we started with. The argument is not circular, since it reduces a minimization over functions to a (much easier) minimization over a finite number of scalars.
>
> How can we minimize $J$ for prescribed start- and endpoints? A standard result in the calculus of variations says that the stationary points of $J$ given these boundary conditions correspond to the solutions of the Euler-Lagrange equations. But a priori, these stationary points might be local maxima or saddle points. It turns out that if the matrix $\mathbf{G}$ is positive definite for all $\mathbf{x}$ and $t$, and the other functions appearing in $J$ are smooth in their arguments, *additional* results from the calculus of variations say that stationary points correspond to local minima, and that the global minimum is among them. This is true even if the loss is non-convex, as long as it is smooth.
>
> (If the loss is non-smooth, because of functions like ReLUs, the result still holds as long as the non-smoothness is not too bad.)
>
> It is possible that there is not a unique solution to the EL equations, as mentioned, e.g., in Appendix B (for example, one can navigate around an object either to the left or to the right, and these paths usually involve an equal amount of effort). But we usually have uniqueness, and uniqueness implies that the EL equation solution is the global minimum. If we do *not* have uniqueness, we simply choose the solution(s) with the lowest $J$ value, which must correspond to the global minimum.
>
> **In summary, the roadmap is this:** we start by hoping to minimize $J$ over a space of functions that correspond to trajectories through parameter space. This space of functions is only constrained by the condition that $\mathbf{x}(0)$ is fixed. Using some calculus of variations results, we can argue that (because of the special form of the objective) stationary points of $J$ correspond to minima, that the global minimum is one of these. Finally, getting this solution required us to specify $\mathbf{x}(\infty)$, so we actually obtained a family of possible solutions, each corresponding to a different value of $J$. The solution we ultimately care about is the one which makes $J$ smallest, and we can find it by directly minimizing $J$ with respect to the $D$ scalars contained in $\mathbf{x}(\infty)$.
>
> In the aforementioned new appendix, we will spell out the required technical conditions (e.g., the convexity and coercivity of the kinetic term) more precisely. But we absolutely agree that getting all of these details straight is crucial for being confident in our framework, and thank you for pointing out that we could be clearer about this.
>
> **On rotational forces.**
>
> > Can the authors provide an example where rotational forces exists?
>
> > I like the presence of the $\mathbf{f}$ term in the general form, but I personally don't see how the rotational term can be of any use in practice. Is there a practical use case that can be shown?
>
> This is a good question, and we will address it in a revised version of the paper. We think the $\mathbf{f}$ term is probably not that useful for modeling machine learning optimization, but may be useful for modeling optimization processes in biological neural networks. Processes like weight decay are known to occur, but more generally one can imagine dynamics that tend to drive a system towards some non-loss-related fixed point. In the simplest case, we can imagine that each parameter decays to zero at the same rate, but a more complicated case could involve each parameter going to zero (or some other value) with different rates, and possibly in a coupled fashion.
>
> An explicit but tractable model of this has
> $$\mathbf{f}(\mathbf{x}) = - \mathbf{A} \mathbf{x}$$
> where $\mathbf{A}$ has eigenvalues whose real parts are positive, so that in the long run $\mathbf{f}$ drives the system towards zero. In general, $\mathbf{A}$ is not necessarily symmetric (i.e., the effect of $x_j$ on $x_i$ may not be the same as the effect of $x_i$ on $x_j$), which means that it is possible for $\mathbf{A}$ to have complex eigenvalues, and hence it is possible that its dynamics include a rotational component.
>
> The pure rotation case that we use as an example is probably unrealistic (although we think it is cool), but a case involving a mix of decay and rotation appears to be more biologically plausible.
>
> **Gradient noise and Adam.**
>
> > I found the assumption in Section 5 on gradient noise to be a bit arbitrary. It would be very helpful if the authors can show some references motivating (at least partially) this assumption. Aren't there other works that motivates Adam through this kind of assumption?
>
> It is funny that you ask this. After the NeurIPS paper submission deadline, another paper appeared that proposes a way to motivate Adam's effectiveness: "In Search of Adam’s Secret Sauce" by Orvieto and Gower. It *also* motivates Adam as effectively implementing an online inference algorithm by assuming noisy observations of gradients. On the other hand, they are not as explicit about a connection between the local curvature of the loss landscape, and the size of the gradient observation noise.
>
> We will cite this paper and find some other references to support our approach.
>
> In any case, we definitely agree that this section can get confusing, and will think hard about how we might be able to simplify/clarify the derivation.
>
> **Miscellaneous and minor concerns.**
>
> > In Line 42 I'd recommend changing "sensitive to structure in parameter space" to "sensitive to non-local structure in parameter space"
>
> We can make this change, thanks for the suggestion.
>
> > In Line 855 there is a typo: "J to minimized" should be changed in "J to be minimized"
>
> Good catch, will fix.
>
> > In Equations 123 and 124, isn't that $J$
>  supposed to be a $\mathbf{J}$? Is it a typo or am I missing something
>
> Good catch. In those equations, we use $J_{ij}$ to denote the components of the Jacobian matrix $\mathbf{J}$. Although it is standard to use non-boldface to indicate components, we admit this may be confusing, and will try to clarify this better in the text. Sorry for the confusion.
>
> > In Line 1099 you state that Equation (133) "matches the main text form up to the prefactor $1/(2 v_i^2)$. Referring to the equation before line 231 in the main text. Can the authors expand on this mismatch? Why is there such difference? And how is the equation in the main text derived?
>
> Eq. 133 in the SI corresponds to the more conceptually simple but more technically annoying gradient noise model, which has
> $$ \mathbf{g}_t \sim \mathcal{N}( \mathbf{m}_t, \mathbf{V}_t / \Delta t) .$$
> It is a little annoying because the corresponding log-likelihood depends non-quadratically on $\mathbf{V}_t$, which complicates the EL equations a bit. In the main text, we consider the ad-hoc assumption that one effectively has *separate/uncoupled* noisy observations of the mean $\mathbf{m}_t$ and the variance $\mathbf{V}_t$. The point of the argument around Eq. 133 is that the ad-hoc assumption yields almost the same EL equations as the non-ad-hoc assumption, since the difference is
>
> $$\frac{[ v_i - (g_i - m_i)^2]}{2 v_i^2} \text{ versus } \frac{[ v_i - (g_i - m_i)^2]}{2 \sigma_2^2} . $$
>
> They approximately agree if $v_i$ equilibrates quickly and $v_i \approx \sigma_2$. We can clarify this point in the text.

---

> > ### Comment · Reviewer_AiaD · 2025-08-08
> >
> > I thank the authors for addressing my comments, and even more for the super interesting discussion with reviewer WGtm.
> >
> > I do personally believe that, for a paper like this one, the lack of experiments and the lack of strong convergence proofs should NOT be a reason for rejection. The simplicity of the framework, together with the wide support of existing algorithms covered, is a good enough reason for acceptance. However, this is a subjective point of view and I can't force others to agree. I keep my score and leave the final decision to the AC.

---

### Official Review · Reviewer_WGtm · 2025-07-02

**Clarity:** 3
**Significance:** 2
**Originality:** 3
**Rating:** 3
**Confidence:** 5

**Summary:**

The paper presents a framework to derive generally learning rules retrieving known ones in the process. Namely, the authors propose to define learning rules by finding continuous-time parameter curves minimizing ${}^{(1)}$ some average future discounted costs comprising the loss but also some parameters-specific costs enforcing e.g. smoothness of parameter changes or beliefs in the process changes.
The authors illustrate their framework to retrieve (i) momentum SGD, (ii) natural gradient descent, (iii) adaptive algorithms à la Adam. They also apply their framework to get insights on continual learning approaches. A long appendix provides detailed computations and explanations for each setting


${}^{(1)}$ the authors solve the Euler-Lagrange equations associated to their objective, but the Euler-Lagrange equations only give stationary points for the objective in terms of continuous time curves.

**Questions:**

- How can the framework explain why Adam could work better than SGD with momentum? More generally the choice of objective to solve for with Euler-Lagrange equations appear as arbitrary as choosing an algorithm, and does not provide much insights on the choice of the algorithm.
- Could the authors use their framework to
  - derive Nesterov's accelerated method?
  - derive learning-rate tuners such as line-searches?
  - derive algorithms using exponential moving average?
  - derive sign-sgd (a variant of sgd that was linked to adam)
- A current difficulty of usual frameworks is to be able to take into account phenomena like edge of stability [1]. Does the proposed framework help to integrate such effects?
- Could the authors test some new algorithms they propose?


[1] Cohen, Jeremy M., et al. "Gradient descent on neural networks typically occurs at the edge of stability." arXiv preprint arXiv:2103.00065 (2021).

**Ethical Concerns:**

["NO or VERY MINOR ethics concerns only"]

**Final Justification:**

I fully acknowledge that I am not the right audience for this paper, while it appears that other reviewers like the paper. A fresh perspective on optimization is needed. This one may be wrong, and experiments (even simple ones like the snippet I proposed) would have tremendously helped. I cannot recommend this paper for acceptance, but I shall not block it. So I will increase my score to 3.

**Limitations:**

See weaknesses and questions.

**Quality:**

2

**Strengths And Weaknesses:**

**Strengths**
- The idea of considering "long term" optimization horizons could be interesting. The authors should integrate algorithms like Lookahead [1] in their framework since it bears similar motivations.
- The derivations in the appendix are clear.
- The paper is well written and pleasant to read (see small remark about tone below though).
- Some new algorithms are derived, though not tested.
- While the momentum and natural gradient viewpoints are very well known from a usual variational viewpoint, the viewpoint for Adam appears new.

**Weaknesses**

*Main*
- The field of optimization may be older than the authors themselves. There already exists frameworks in optimization to design algorithms in a principled way. An essential example is the one developed by e..g A. Nemirovisky and D. Rudin [2] that influenced optimization for years. Their framework defines what could the best possible algorithm for a given class of problems, read for example [3] for a gentler introduction. Such a framework had the benefits to understand how different algorithms compare given some preliminary information on it. It also bore fruits: the accelerated algorithm of Nesterov was motivated by finding the best algorithm for smooth and convex problems. In comparison, the framework proposed by the authors does not help us understand how these algorithms compare for a given class of problems. The framework provides ad-hoc explanations to well-known algorithms.
- "it it makes it easier to generate new learning rules from a principled starting point, and to justify them without empirical guesswork" Actually an algorithm is made to be applied. Experiments are the gold standard, the truth to understand the relevance of the theory. The ultimate test for the proposed framework is to get a new algorithm that perform better in practice (like sharpness aware minimization was designed after studying sharpening effects). The authors actually have some new algorithms like the "ballistic one", why aren't they tested empirically? At least each intuition that the framework provides could be tested empirically to see how relevant the framework is. Why aren't each intuition tested empirically to analyze their relevance? Is it left for other people to do this work?
- Numerous approximations are sometimes needed to find the desired algorithm in the given framework (see Adam for example).
- All these algorithms had previous justifications: momentum is well known to be motivated from a physics viewpoint (it's called Polyak's heavy ball), Adam is justified as a form of Adagrad, Adagrad itself is justified a s a form of a normalized gradient, there are many studies from those viewpoints that gave new intuitions like understanding how normalized gradient can adapt to the edge of stability phenomenon. The authors need to justify what brings their approach. "One rule to rule them all" is not a good motivation (especially given that their framework won't encompass all possible algorithms).

The paper presents many algorithms through the given framework. It could be much better to focus on one algorithm each time and provide numerical experiments illustrating either new algorithms that the framework gave rise to (like the ballistic one) or to illustrate empirically some intuitions that the framework gave. Even the "Newton's method" that the framework gives in section 3 is not the actual Newton method. Maybe it gives a much better algorithm, maybe not at all. Probably this framework completely misses the need to adjust for the learning rate once a learning rate rule is found. A Newton method needs a priori to be damped for example, is it the case for what the authors propose? There could be interesting fruits from the framework but currently trying to frame each algorithm through one lens does not add much relevance to the approach. Experiments will prove the relevance.

*Minor*
- "Natural gradient descent is not second-order optimization in disguise" The authors are presenting generic preconditioning methods. Indeed they are not presenting natural gradient descent which is one particular instance. Updates using pre-conditioners are well motivated from a variational viewpoint. Actually one can go beyond using simple Mahalanobis distances and consider Bregman divergences, or any gradient mapping coming from a partial linearization of the objective.

*Additional notes*:
- A humble tone may be best to invite the reader to understand the relevance of the idea rather than a forceful tone affirming for example that "[this] framework is significant". A paper is written to present an idea not to argue or convince about its significance, let the reader judge by themselves.
- There is a missing identity matrix in the "effective learning rate" of the Newton step and it's not an "effective learning rate", learning rates are scalars, a Newton method does more than multiplying with a scalar.

[1] Zhang, Michael, et al. "Lookahead optimizer: k steps forward, 1 step back." Advances in neural information processing systems 32 (2019).
[2] Nemirovskij, Arkadij Semenovič, and David Borisovich Yudin. "Problem complexity and method efficiency in optimization." (1983).
[3] Nesterov, Yurii. Introductory lectures on convex optimization: A basic course. Vol. 87. Springer Science & Business Media, 2013.

---

> ### Author Rebuttal · Authors · 2025-07-27
>
> We deeply appreciate the reviewer's time and detailed feedback, and think that your comments have improved the paper. We will respond to some of the major concerns first.
>
> **What is the value of a normative framework for learning rules?**
>
> > "it makes it easier to generate new learning rules from a principled starting point, and to justify them without empirical guesswork" Actually an algorithm is made to be applied. Experiments are the gold standard, the truth to understand the relevance of the theory.
>
> > Experiments will prove the relevance.
>
> This touches on an important philosophical point, which we will clarify in a revised version of the paper. In the end, most machine learning practitioners would like an algorithm that helps their models learn as quickly as possible. What's the point of theory here?
>
> We think neither theory nor experiment is the endpoint. Rather, the process is best viewed as a 'virtuous cycle': experiments identify which learning rules and hyperparameters work well in practice; theory can help clarify why those algorithms work as well as they do, and perhaps suggest improvements; those improvements can be tested in further experiments; and so on. Said differently, the ideal workflow is not theory -> experiment, nor is it experiment -> theory; it is theory -> experiment -> theory -> experiment ...
>
> Our statement "it makes it easier ..." does *not* mean that there is nothing more to be said, or that no experiments should be done. It means that we provide a principled framework for justifying learning rules from a theoretical perspective, which can *then* be tested in experiments. It is not the only framework possible, but it is fairly broad (for example, it encompasses a number of known learning rules, as we show in the paper) and has a number of advantages.
>
> One of these advantages is that the 'goodness' of a given learning rule, given a set of assumptions, can be quantified by a single scalar, rather than more complicated quantities like convergence rates. This means that there is an unambiguous way to compare learning rules; one rule is better than another if the corresponding number is lower (i.e., lower loss and more efficient navigation). Another advantage is that it generally identifies a unique rule and set of hyperparameters as optimal.
>
> We emphasize that the idea of this variational approach is not completely new, and has been successfully used in previous literature to derive interesting insights about optimization methods. The Wibisono et al. 2016 paper we cite is a good example.
>
> **Our framework is not ad-hoc.**
>
> > the framework proposed by the authors does not help us understand how these algorithms compare for a given class of problems. The framework provides ad-hoc explanations to well-known algorithms.
>
> > ... the choice of objective to solve for with Euler-Lagrange equations appear as arbitrary as choosing an algorithm, and does not provide much insights on the choice of the algorithm.
>
> We think it is inaccurate to describe our framework as ad-hoc. It is 'normative' in the sense that, given a set of assumptions, it identifies one or a family of learning rules as 'optimal'. Moreover, these assumptions are fairly reasonable, involving (among other things) minimizing the loss, efficiently navigating through the loss landscape, and temporal discounting. The framework says, *given some assumptions*, that one or a family of rules is best in a precise sense.
>
> One potential objection to this approach is, why these assumptions and not others? Equivalently, where does the objective function come from? There is admittedly infinite freedom here, which is why we use fairly simple assumptions throughout. For example, we only consider quadratic kinetic terms like $\dot{x}^2$, and an exponential temporal discounting function. One becomes more confident that these assumptions are reasonable by rederiving known rules.
>
> On the other hand, it is worth emphasizing that any approach of this type has to assume *something*, and even in classical work (e.g., by Nesterov) this is true. The assumptions are taken as primitive, and we can use theory tools to derive the consequences. Whether the resulting rules apply in more realistic situations is indeed a question best clarified by experiments, and by proceeding with the 'virtuous cycle' described above.
>
> **Not all algorithms are encompassed by the framework.**
>
> > (especially given that their framework won't encompass all possible algorithms).
>
> It is absolutely true that not all optimization algorithms can be derived via our framework, and we think this is a feature rather than a bug. In particular, this fact can be used to suggest changes to well-known algorithms. For example, although you point out that:
>
> > Numerous approximations are sometimes needed to find the desired algorithm in the given framework (see Adam for example)
>
> ...a different interpretation is that the framework identifies appropriate ways Adam could be modified. While it remains to be seen whether these modifications perform better in practice, these hypotheses are at least interesting consequences of the framework. The framework does not provide 'just-so' stories that can be used to 'derive' absolutely any algorithm.
>
> As a minor point, algorithms like line search (which do not necessarily involve smooth changes in a parameter, since it 'jumps' around to look for a minimum) are probably not encompassed by our framework, which only concerns optimizers that involve smooth parameter changes.
>
> **How is this framework's justifications different than other justifications?**
>
> > All these algorithms had previous justifications
>
> This is absolutely true. But one feature of our framework that we think is nice is that it provides quantitative guidance about how good an algorithm is through the aforementioned 'goodness' score (essentially, the objective $J$). This is somewhat different than how momentum is usually justified: while the physics analogy suggests including momentum is intuitively *reasonable*, our framework shows that it is *quantitatively optimal* under certain assumptions. A similar statement holds for Adam and the other algorithms we discuss. In short, the framework offers quantitative insight (that algorithm X is optimal, and *moreover* Y amount better than algorithm Z) that complements existing insight. Not all existing insight is qualitative---for example, convergence rates are not---but a single performance scalar, and a claim regarding optimality, is arguably somewhat simpler, or at the very least helpful.
>
> **Where are the experiments?**
>
> > The authors actually have some new algorithms like the "ballistic one", why aren't they tested empirically?
>
> > Could the authors test some new algorithms they propose?
>
> Although our aim is primarily to develop a unifying theoretical framework for various existing learning rules (in the spirit of other recent work in machine learning, like Alshammari et al. 2025), we agree that additional experiments will help demonstrate the utility of the framework and learning rules derived from it.
>
> In a revised version of the paper, we plan to explicitly test some of these algorithms (e.g., the ballistic one) in a few simple cases, including a toy convex quadratic loss and in simple image classification problems (e.g., MNIST, CIFAR-10). These types of problems are already used in many papers benchmarking algorithms like Adam, so although they are simple, they seem appropriate here. Our goal is not necessarily to derive a state-of-the-art algorithm that performs better than existing algorithms on a frontier problem.
>
> We cannot show the results here, since it is not possible to include links or PDFs during the rebuttal, but do not anticipate that these experiments will be difficult to perform.
>
> **Extensions of the framework.**
>
> > Could the authors use their framework to derive ...
>
> There isn't space to go into details, but in short, the framework definitely includes Nesterov's accelerated method and exponential moving average as examples. The accelerated method can be derived assuming a similar objective to the one from Sec. 3, and algorithms involving an EMA can be derived using an objective similar to Adam. (The insight is that EMA is optimal when past gradients are informative about current and future gradients.) We will add both derivations to the paper.
>
> As mentioned above, the framework probably doesn't encompass line searches or algorithms where parameter updates are 'jumpy', like sign-SGD (although maybe sparsity or compression constraints can help).
>
> **Miscellaneous and minor concerns.**
>
> - In the revised version, we will add many more numerical experiments and figures. This both helps show the framework is useful, and makes the paper easier to follow.
> - We are happy to modify the tone somewhat, and agree that the lack of humbleness in certain places could put off some readers.
> - We will add the identity matrix to our expression for $\eta_{eff}$. This wasn't a typo, just slightly sloppy notation. It's true that the term "learning rate" generally refers to a scalar, but we think it is a helpful and not-too-egregious abuse of terminology to speak about a 'non-scalar learning rate' in the context of preconditioning methods.
> - We agree that the Lookahead optimizer provides an interesting test case for our framework. We will cite it and show how to derive it using our framework.
> - We agree that the edge of stability phenomenon is extremely interesting, but think it would require a completely separate analysis to link our framework to it. The problem is that most learning rules are derived using simple assumptions (e.g., a convex and often quadratic loss); these simple assumptions are probably insufficient for modeling a phenomenon associated with the highly non-convex loss landscapes of neural networks. We will cite this paper and note this as a limitation of our work, but not necessarily as a limitation of the framework overall.

---

> > ### Comment · Reviewer_WGtm · 2025-08-06
> >
> > I beg your pardon I had difficult internet access. I thank you for your detailed answer.
> >
> > **What is the value of a normative framework for learning rules?**
> >
> > > philosophical point...
> >
> > Ideally the flow should be [authors ABC: theory+experiments] -> [authors DEF: theory+experiments]. It should not be  [authors ABC: theory] -> [authors DEF: experiments]. It's not just that it's leaving part of the work to others. It's just about humbly acknowledging that "all models are wrong, but some are useful". To know if they are useful, one needs to try them. There is joy in doing both theory and experimentation at the same time. On the other hand, doing only theory leaves others banging their heads on walls to understand why it does not work in practice.
> >
> > > The corresponding number is lower
> >
> > The loss is not capturing all you need. Look at phenomenons such as the edge of stability, grokking, etc... Try a line-search on one or even more steps to adjust the learning rate as a function of the loss in deep learning. It simply does not work, see [1]. One needs to try it experimentally to know.
> >
> > **Not all algorithms are encompassed by the framework.**
> >
> > > a different interpretation is that the framework identifies appropriate ways Adam could be modified
> >
> > Yes! But then please, please, please try empirically these modifications. Should I for example now do your work and try these modifications by myself, losing a few weeks of my time to maybe see that it does not work at all? Isn't it extremely unfair? Maybe your framework is great and I honestly feel bad about potentially rejecting it. Just please do some experiments and we'll see!
> >
> > > it is quantitatively optimal under certain assumptions
> >
> > The problem is that the assumptions vary from one algorithm to the other. So no comparison is possible.
> >
> >
> > Again thank you for your answer and sorry for the delay. The current framework could actually inspire for example to use model predictive control algorithms with the adequate models but really I cannot accept a paper that leaves out experiments to other researchers with the risk of leaving them do the dirty work.
> >
> > [1] Roulet, Vincent, et al. "Stepping on the edge: Curvature aware learning rate tuners." Advances in Neural Information Processing Systems 37 (2024): 47708-47740.

---

> ### Author Response · Authors · 2025-08-06
>
> > Ideally the flow should be [authors ABC: theory+experiments] -> [authors DEF: theory+experiments]. It should not be [authors ABC: theory] -> [authors DEF: experiments]. It's not just that it's leaving part of the work to others. It's just about humbly acknowledging that "all models are wrong, but some are useful". To know if they are useful, one needs to try them. There is joy in doing both theory and experimentation at the same time.
>
> We appreciate the broader point, but think it is worth clarifying our aims. We both agree that theory and experiment are both valuable. But different researchers have their preferred theory-experiment balance, and different motivations as well. Some researchers (and admittedly, the author writing this response) find unifying theoretical frameworks intrinsically interesting. Even in a hypothetical world where all possible learning rules have somehow been discovered and evaluated, I would *still* find a framework which 'unified' them (for example, by linking many learning rules to a single objective, as we do here) interesting. I think many other theory-focused ML researchers would feel the same way. My point is that a theory can have value independent of its empirical utility.
>
> Like you, others may be more interested in empirical performance. While we emphasize once again that our central goal was not to derive new learning rules that achieve state-of-the-art performance on frontier problems (otherwise, we would have written a very different kind of paper), **we reiterate that we plan to perform experiments to test some of the learning rules our theory suggests.**
>
> Due to the constraints of the rebuttal period—and the fact that NeurIPS policy prohibits including new results figures—we have not included figures here, but we will incorporate them in the revised version. We say this in our rebuttal (and in the rebuttals to the other reviewers) above:
>
> > In a revised version of the paper, we plan to explicitly test some of these algorithms (e.g., the ballistic one) in a few simple cases, including a toy convex quadratic loss and in simple image classification problems (e.g., MNIST, CIFAR-10).
>
> Hence, **we will include experiments trying out learning rule variants suggested by our theory** in a revised version.
>
> > The loss is not capturing all you need. Look at phenomenons such as the edge of stability, grokking, etc... Try a line-search on one or even more steps to adjust the learning rate as a function of the loss in deep learning. It simply does not work, see [1]. One needs to try it experimentally to know.
>
> We seem to be talking past each other here. All we are saying is that, from the *theoretical* point of view, our framework provides a nice way of comparing different algorithms for a fixed objective, and in particular identifies one or more as 'optimal' given fixed assumptions.
>
> Yes, in practice maybe the problem you'd like to solve is different than the theoretical one you set up earlier (i.e., maybe there is a theory-experiment mismatch). This does not change the fact that the theory provides unambiguous insight into which rule or rules is optimal, *with respect to certain assumptions*. Whether or not those assumptions apply to a given set of experiments is a different question.
>
> Finally, we agree that phenomena like the edge of stability are interesting, but it was not our goal here to address them.
>
> > Yes! But then please, please, please try empirically these modifications. Should I for example now do your work and try these modifications by myself, losing a few weeks of my time to maybe see that it does not work at all? Isn't it extremely unfair? Maybe your framework is great and I honestly feel bad about potentially rejecting it. Just please do some experiments and we'll see!
>
> See above response. We have already promised to do this in a revised version.
>
> > The problem is that the assumptions vary from one algorithm to the other. So no comparison is possible.
>
> To clarify again, the statement is fairly simple: given a set of assumptions, the objective $J$ can be computed for any possible learning rule. This follows straightforwardly from the structure of the framework. Different rules can then be compared, with respect to those fixed assumptions, and one or more of them can be shown to be optimal.
>
> Yes, *which* assumptions are used controls which rule is optimal. According to our work, a rule with momentum is optimal under different assumptions than a rule without momentum. **This does not change the fact that, given fixed assumptions, a comparison is always possible, and always meaningful.**
>
> > ... but really I cannot accept a paper that leaves out experiments to other researchers with the risk of leaving them do the dirty work.
>
> We understand and respect your emphasis on empirical validation. To that end, we’ve already committed to implementing the suggested experiments in the revision.

---

> > ### Author Response · Authors · 2025-08-07
> >
> > Below are some additional details about the experiments we plan to run in case they are helpful. The experiments are not intended to demonstrate benchmark performance improvements, but to characterize in slightly more realistic (albeit still relatively simplistic) settings the behavior of various rules we consider. This includes some of the novel rules we propose, like the "ballistic" rule (which involves the square root of the inverse Hessian; see Section 2).
> >
> > **Experiments planned.**
> >
> > 1. Toy examples using convex quadratic loss landscapes to compare gradient descent, momentum, Newton, natural gradient, ballistic, and Adam-like rules. We have analytic results for these, and based on them expect to see how different rules (assuming identical learning rates) have different-shaped loss curves, and converge to the global minimum at different rates. For example, gradient descent decreases the loss more quickly on short time scales, but rules with momentum (which via our framework is linked to longer-horizon optimization) reach the global minimum more quickly overall.
> >
> > 2. Trajectory visualizations in low-dimensional (e.g., 2D) settings to highlight differences in the curvature, speed, or smoothness of paths.
> >
> > 3. Small-scale classification tasks on MNIST and CIFAR-10 (with shallow architectures), to demonstrate that these behavioral differences persist even in non-idealized settings.
> >
> > These are not “method X is better than method Y” comparisons. Rather, they serve to clarify how structural assumptions—e.g., about geometry, planning horizon, or observability—translate into qualitative differences in learning behavior.

---

> > > ### Comment · Reviewer_WGtm · 2025-08-08
> > >
> > > Below is the code for the ballistic on a simple quadratic.
> > > Its behavior appears non-trivial and superior to a simple gradient descent.
> > > In the future, the authors should simply send such a code. It allows everyone to see the results. No need for a pdf, nor for promises.
> > >
> > > Some potential additional notes that could improve the paper:
> > > - Consider making a related work on "frameworks for optimization" or other ways to derive these algorithms.
> > > - Rephrasing the paper not as a framework but as "rederiving learning rules from a infinite-time horizon control perspective" could largely help tone down many of the claims
> > > - The insight on natural gradient descent is not new. Amari itself proposed natural gradient descent as a change of geometry and not an approximation of the loss. The "second-order optimization viewpoint" is useful to derive convergence proofs. What is new is how the proposed viewpoint mixes preconditioner and Hessian. A classical discrete viewpoint would end up adding hessian and preconditioner. The proposed viewpoint is quite different as it moves along a continuous curve. I would again like to know which one works better. Also the literature on natural gradient descent is long so it would be best to see what has been proposed as "natural gradient descent with a Hessian".
> > > - I do not believe the stochasticity of the loss is properly tackled. The noise makes the proposed integral a stochastic integral and requires appropriate tools to be solved. Maybe detail better how is the stochasticity taken into account in each scenario.
> > > - Figure 1 is overly simplistic in my opinion (compare for example to the description of the lookahead optimizer that presents an actual optimizer with the "long horizon" idea).
> > >
> > > There are details that will probably puzzle most of the mathematical audience such as:
> > > - l 234: "To good approximation, this means that the optimal learning rule ..." What approximation?
> > > - l 228: Why is there a "$\lim_{\delta t \rightarrow 0}$" for the derivation of the Adam rule?
> > > - l 228: How was it possible to move the outside expectation of the original objective inside the integral? If there is some stochasticity it should have been taken into account properly.
> > > - l1099: "If the variance is fairly stable (for example, because a reasonable amount of evidence about gradients has already been accumulated), then this prefactor is approximately constant, and the forms are identical" But if it is fairly stable why do you consider derivatives of v?
> > > - The loss landscape derived for Adam (with the variance) is done around a steady state. Why would this approximation be valid during optimization?
> > >
> > > Overall, I understand that all these approximations can be valuable to find a model. But then, either the paper proposes a framework that is clear and approachable to everyone, or it provides evidence of its relevance (and support for each of the approximations done). Neither is done here.
> > >
> > > I hope that in this message the authors will find my sincere beliefs in the value of the paper after its next major revision. For now, I don't believe it would be reasonable to accept the paper as is.
> > >
> > > ```
> > > # Test ballistic step on quadratic
> > >
> > > import jax.numpy as jnp
> > > import jax.random as jrd
> > > import jax
> > > from matplotlib import pyplot as plt
> > >
> > > # Setup function
> > > d = 4
> > >
> > > A = jrd.normal(jrd.key(3), (d, d))
> > > A = A.dot(A.T)
> > > b = jrd.normal(jrd.key(1), (d,))
> > >
> > > eigs = jnp.linalg.eigvals(A).real
> > > max_eig, min_eig = jnp.max(eigs), jnp.min(eigs)
> > > cond_number = min_eig/max_eig
> > > sqrt_cond_number = jnp.sqrt(cond_number)
> > >
> > > print(f'{max_eig=:<10} {min_eig=}')
> > >
> > > def f(x):
> > >   return 0.5*x.dot(A).dot(x) + b.dot(x)
> > >
> > > min_f = -0.5*b.dot(jnp.linalg.solve(A, b))
> > >
> > > # Setup algorithms
> > > eta = 1
> > > k = 1
> > > delta_t = 5*1e-1
> > >
> > > stepsize_gd = 1/max_eig
> > >
> > > def ballistic_step(x):
> > >   H = jax.hessian(f)(x)
> > >   g = jax.grad(f)(x)
> > >   sqrtH = jax.scipy.linalg.sqrtm(H).real
> > >   return x - delta_t*jnp.sqrt(eta*k)*jnp.linalg.solve(sqrtH, g)
> > >
> > > def grad_step(x):
> > >   return x - stepsize_gd*jax.grad(f)(x)
> > >
> > > ballistic_step = jax.jit(ballistic_step)
> > >
> > > # Run algorithm
> > > maxiter = 100
> > >
> > > def run_algo(algo_step):
> > >   # print(f'####\n{algo_step}\n####')
> > >   fun_vals = []
> > >   x = jnp.zeros(d)
> > >   init_fun_val = f(x) - min_f
> > >   for iter in range(maxiter):
> > >     fun_val = (f(x) - min_f)/init_fun_val
> > >     fun_vals.append(fun_val)
> > >     # print(f'{iter=:<10} {fun_val=}')
> > >     x = algo_step(x)
> > >   return fun_vals
> > >
> > > # Plot fun values
> > > fun_vals_ballistic = run_algo(ballistic_step)
> > > fun_vals_gd = run_algo(grad_step)
> > >
> > > plt.plot(fun_vals_ballistic, label='ballistic')
> > > plt.plot(fun_vals_gd, label='gd')
> > > plt.yscale('log')
> > > plt.legend()
> > > plt.show()
> > >
> > > # Estimate rates
> > > def estim_rate(fun_vals):
> > >   rates = jnp.diff(jnp.log(jnp.array(fun_vals)))
> > >   return -jnp.mean(rates[:10])
> > >
> > > rate_ballistic = estim_rate(fun_vals_ballistic)
> > > rate_gd = estim_rate(fun_vals_gd)
> > > print(f'{rate_ballistic=:<5.2e} {rate_gd=:<5.2e} {cond_number=:<5.2e} {sqrt_cond_number=:<5.2e} ')
> > > ```

---

> > > > ### Author Response · Authors · 2025-08-08
> > > > **Response, part 1**
> > > >
> > > > Thanks for continuing to engage with the work.
> > > >
> > > > **On experiments.**
> > > >
> > > > > Below is the code for the ballistic on a simple quadratic. Its behavior appears non-trivial and superior to a simple gradient descent. In the future, the authors should simply send such a code. It allows everyone to see the results. No need for a pdf, nor for promises.
> > > >
> > > > First, our theoretical results already exactly characterize the behavior of the ballistic learning rule (and various other rules) for a quadratic loss. While I agree that it is easy to write this code, I do not find this specific experiment that informative. As long as the assumptions of the theory match the assumptions of the experiment (e.g., full-batch updates are used), there ought to be agreement.
> > > >
> > > > What we would prefer to do later, both given the rule-related limitations of the rebuttal period (e.g., no links and PDFs are allowed) and the short time associated with the rebuttal period, is a more detailed set of numerical experiments that show the behavior of different rules on more complex losses (e.g., MNIST or CIFAR-10 image classification). This is what we are mainly talking about. If you would have been satisfied by simple code like the snippet you shared, then there was clearly a misunderstanding.
> > > >
> > > > The kind of experiments we're talking about are not easily reportable (or at least, usefully reported) in text form, since we are not that interested in final loss values. Especially given the simple nature of the tasks we will consider, what is much more interesting is the qualitative behavior of different rules, like how quickly loss curves go down, and how quickly the parameter approaches its final value along different parameter space directions. Once again, it's possible to summarize this with numbers, but much more opaque than just showing loss curves and trajectories.
> > > >
> > > > A priori, you may be concerned that something could go wrong with the experiments, and hence, it is unreasonable to accept the paper without them already having been done. But this concern does not square with the central aims of our work. **As we have repeated a few times, our goal is not to develop new SOTA rules here, but to develop a framework for unifying various different rules, which can provide intuition about how different rules may be useful** (e.g., Adam relates to long-horizon optimization and local loss approximation). This means that there isn't really anything that can go wrong. Maybe some novel rules work well or don't, but we are just characterizing their behavioral differences; it's unfortunate if a rule doesn't work well, but orthogonal to our aims.
> > > >
> > > > You may still want experiments done and disagree with our assessment. The last thing to note is that authors may have other commitments, and may not have the bandwidth to do adequate experiments during the short rebuttal period.
> > > >
> > > > (continued below)

---

> > > > > ### Author Response · Authors · 2025-08-08
> > > > > **Response, part 2**
> > > > >
> > > > > **On additional notes.**
> > > > >
> > > > > > Consider making a related work on "frameworks for optimization" or other ways to derive these algorithms.
> > > > >
> > > > > What does this mean? If you are asking us to include more discussion of related work, of course we are happy to do this.
> > > > >
> > > > > > Rephrasing the paper not as a framework but as "rederiving learning rules from a infinite-time horizon control perspective" could largely help tone down many of the claims
> > > > >
> > > > > What does this mean? It is a theoretical framework for deriving learning rules. This is a fair assessment, and one which is factually well-supported by the entire main text. It indeed involves an infinite-time horizon and ideas related to control theory, but I do not understand how your proposal would substantively change anything we have said. We already use those words throughout the paper (e.g., in the abstract, in Section 2).
> > > > >
> > > > > > The insight on natural gradient descent is not new. Amari itself proposed natural gradient descent as a change of geometry and not an approximation of the loss.
> > > > >
> > > > > We did not claim that it was new. Hence calling it "Amari's" insight. Our contribution is about the relationship between loss approximation (which involves a Hessian) and parameter space geometry (which involves a different positive definite matrix). See lines 158-168.
> > > > >
> > > > > > The "second-order optimization viewpoint" is useful to derive convergence proofs. What is new is how the proposed viewpoint mixes preconditioner and Hessian. A classical discrete viewpoint would end up adding hessian and preconditioner. The proposed viewpoint is quite different as it moves along a continuous curve. I would again like to know which one works better. Also the literature on natural gradient descent is long so it would be best to see what has been proposed as "natural gradient descent with a Hessian".
> > > > >
> > > > > We found it hard to find work that combines both in the sense we describe, but do cite a few related works (e.g., line 157, ref 31). We can add more references if you like.
> > > > >
> > > > > It is worth noting that the question of which works better is not well-posed. Probably one or the other works better depending on the problem and related details. It is not our goal to propose SOTA methods. It isn't our goal to write that kind of paper.
> > > > >
> > > > > > I do not believe the stochasticity of the loss is properly tackled. The noise makes the proposed integral a stochastic integral and requires appropriate tools to be solved. Maybe detail better how is the stochasticity taken into account in each scenario.
> > > > >
> > > > > We provide detail on how stochasticity is taken into account in each scenario, given that we have a detailed set of appendices and almost 150 equations. We can add more detail. There are no major mathematical errors, and no stochastic integral formalism is necessary here. This should be clear from studying any of the specific examples we discuss. In all cases, we always first take expectations of any relevant random variables, and *then* perform the optimization (which is always over non-random variables).
> > > > >
> > > > > > Figure 1 is overly simplistic in my opinion (compare for example to the description of the lookahead optimizer that presents an actual optimizer with the "long horizon" idea).
> > > > >
> > > > > We have already said that we agree that the paper would be improved by additional figures illustrating various conceptual/technical points. They are not necessary for correctness, but can help improve clarity. We have already mentioned in several places, including in our original response to you, that we will do this.
> > > > >
> > > > > (continued below)

---

> > > > > > ### Author Response · Authors · 2025-08-08
> > > > > > **Response, part 3**
> > > > > >
> > > > > > **On math-related comments.**
> > > > > >
> > > > > > > There are details that will probably puzzle most of the mathematical audience such as:
> > > > > >
> > > > > > This seems to be an exaggerated assessment. Other reviewers were not quite so puzzled. In any case, what you point out are not errors, but places where, at worst, additional clarification could be helpful.
> > > > > >
> > > > > > > l 234: "To good approximation, this means that the optimal learning rule ..." What approximation?
> > > > > >
> > > > > > The specific approximation is described immediately above the line you cite, in lines 232-233. To exactly quote those lines "If we assume that parameter changes are ballistic ($\eta \gg \gamma)$ but landscape beliefs change somewhat more slowly ($\xi_1^2, \xi_2^2 \ll \gamma$),". Similar approximations are used elsewhere throughout the paper, e.g., the ballistic learning rule is derived from the more general momentum-involving rule (Section 3, Eq. 7) assuming $\gamma = 0$.
> > > > > >
> > > > > > > l 228: Why is there a "$\lim_{\Delta t \rightarrow 0}$" for the derivation of the Adam rule?
> > > > > >
> > > > > > The objective function depends on assumptions about how beliefs change over time. Since we work in continuous time, the log of the transition probability scales with $\Delta t$ (this is typical for, e.g., transition probabilities related to SDEs). Dividing by $\Delta t$ is needed both to get a consistent continuous time limit, and to consistently interpret the objective as involving a log-likelihood maximization term. We thought it may be clearer to write the equation this way, since this shows where it comes from. The actual form of the integrand, given the assumed belief model and actually taking the $\Delta t \to 0$ limit explicitly, is written in Appendix F:
> > > > > >
> > > > > > $\sum_i \left( \frac{\dot{\theta}_i^2}{2 \eta}  +   \frac{( g_i - m_i )^2}{2 v_i} + \frac{1}{2} \log( 2 \pi v_i ) +  \frac{( \dot{m}_i + \alpha_1 m_i )^2}{2 \xi_1^2} + \frac{( \dot{v}_i + \alpha_2 v_i )^2}{2 \xi_2^2} + k \mathbb{E}[\hat{\mathcal{L}}(\boldsymbol{\theta}_t)]    \right) e^{- \gamma t} $
> > > > > >
> > > > > > We can provide an additional comment that clarifies this point.
> > > > > >
> > > > > > > l 228: How was it possible to move the outside expectation of the original objective inside the integral? If there is some stochasticity it should have been taken into account properly.
> > > > > >
> > > > > > Only the loss is treated as a random variable here, so that is the only place the expectation has an effect. Expectations are linear operators, and so can be moved inside the integral. This is very standard.
> > > > > >
> > > > > > > l1099: "If the variance is fairly stable (for example, because a reasonable amount of evidence about gradients has already been accumulated), then this prefactor is approximately constant, and the forms are identical" But if it is fairly stable why do you consider derivatives of v?
> > > > > >
> > > > > > The statement is about how much $v$ varies relative to its typical value $(g - m)^2$. If it is not far from its typical value, the two belief models we consider (a more complex one, and a more simplified one) agree, and both yield Adam. If it is far from its typical value, each belief model yields a slightly different learning rule, at least in that regime. This distinction isn't that important for obtaining Adam, since (again, lines 232-233) doing so involves an assumption that the loss landscape changes somewhat more slowly than typical parameter changes.
> > > > > >
> > > > > > > The loss landscape derived for Adam (with the variance) is done around a steady state. Why would this approximation be valid during optimization?
> > > > > >
> > > > > > It was not done around a steady state, but with respect to a local loss landscape approximation. This idea was used again and again throughout the paper. In optimization, this is standard, and relates to the idea of a 'trust region'. One takes small enough steps that the local loss approximation is reasonable. If one makes a step and a new local approximation is better, one should use that instead. It is kind of weird to take issue with this, since locally approximating a loss (via gradients, for example) is one of the cornerstone ideas of optimization.
> > > > > >
> > > > > > > Overall, I understand that all these approximations can be valuable to find a model. But then, either the paper proposes a framework that is clear and approachable to everyone, or it provides evidence of its relevance (and support for each of the approximations done). Neither is done here.
> > > > > >
> > > > > > The other reviewers appeared to generally appreciate the framework and find it reasonably clear. We provided evidence of relevance by deriving a large variety of well-known learning rules, and, as a bonus, noted that our framework naturally yields variants of well-known rules (like the ballistic rule). Your statement seems to be wrong, or at the very least an extreme exaggeration.
> > > > > >
> > > > > > (continued below)

---

> > > > > > > ### Author Response · Authors · 2025-08-08
> > > > > > > **Response, part 4**
> > > > > > >
> > > > > > > **On what remains unaddressed.**
> > > > > > >
> > > > > > > > I hope that in this message the authors will find my sincere beliefs in the value of the paper after its next major revision. For now, I don't believe it would be reasonable to accept the paper as is.
> > > > > > >
> > > > > > > Please consider the detailed responses we have provided to every one of your points. Specific issues you brought up (can framework include X rule? is there an error on line Y?) have all been addressed, and many were arguably not really issues to begin with. We have not yet done a detailed set of experiments, and for reasons of time (short rebuttal window) and goals (it is explicitly not our aim to derive SOTA rules, so whether a novel one works well or not is kind of beside the point) prefer to do this at a later time. This does not affect the correctness or value of the paper. It is mainly a theoretical work. It does not seem reasonable to heavily penalize a paper just because it is not the kind of paper you would have written.

---

> ### Comment · Reviewer_WGtm · 2025-08-09
>
> Thank you for your answers. I beg your pardon, my tone was sometimes inappropriate (like "puzzle most of the mathematical audience" or "No need for a pdf, nor for promises."). I wanted to answer before the deadline and I should have reread my messages. Again sincere apologies.
>
> **Comments.**
>
> > our theoretical results already exactly characterize the behavior of the ballistic learning rule (and various other rules) for a quadratic loss.
>
> I had not seen what convergence rate was obtained, nor what choice of parameters ensured convergence. I would sincerely appreciate it if you could point me to those if they are present.
>
> > The kind of experiments we're talking about are not easily reportable
>
> You mentioned toy examples, see different loss curves, and convergence to the global minimum at different rates. My code, if you run it, plots the loss curves. It also computes the instantaneous rates and compare it to theoretical ones known for gradient descent or accelerated gradient descent. My point was: yes, the ballistic rule behaves in a quite interesting way. There are regimes where the convergence appear independent of the condition number for example. Plotting distance to the solution or trying another toy function can also easily be added to the snippet above.
>
> Did you have a more complex plan in mind? Like testing the validity of the approximations that were necessary for each algorithm?
>
> > making a related work on "frameworks for optimization" What does this mean ?
>
> See the introduction to Nesterov's book "Lectures on convex optimization". There the author presents a framework to study optimization. The author defines what are classes of problems, then classes of algorithms and final complexities of algorithms for given classes of problems. Such a framework allowed the community to understand theoretically whether an algorithm may be better than another one in convex optimization. The authors present a way to derive algorithms from an infinite time horizon objective. That objective and its parameters provide intuitions on the algorithms but, compared to Nesterov's framework, does not give one-to-one comparisons like worst case convergence rates for a given class of problems.
>
> On that note, the cost per iteration of the algorithms could be quickly discussed.
>
> The word framework seemed in that sense too ambitious. But this is a matter of phrasing, and it is the decision of the authors not the reviewer.
>
> > We did not claim that it was new.
>
> Right, I beg your pardon. Other reviewers considered this as a contribution. Reviewer KkuV said for example "The insight on the relationship between natural gradient descent and second order methods appears to be novel and makes a valuable distinction between the two." My point here was to try to highlight how the proposed viewpoint differed compared to a usual optimization viewpoint. This was a constructive comment to offer more perspective on the contribution, not an argument against acceptance.
>
> > It is worth noting that the question of which works better is not well-posed. Probably one or the other works better depending on the problem and related details. It is not our goal to propose SOTA methods. It isn't our goal to write that kind of paper.
>
> Yes, I understood that your goal was not to propose SOTA methods, thanks. Here indeed, the viewpoints (your framework vs usual optimization framework) diverge. It is worth (i) noting the exact differences (which is not that preconditioner and Hessian can be decoupled but how they can be coupled) and (ii) studying when the proposed framework is more appropriate .This has nothing to do with "SOTA methods". It is a simple matter of understanding the relevance of the approach compared to previous work.
>
> > We provide detail on how stochasticity is taken into account ...
>
> > Only the loss is treated as a random variable
>
> Ok, sorry I think I see. Before solving the EL equations, parameters and loss are decoupled so typically the parameters are not random before taking the expectation. If I may ask a follow-up question: during a stochastic optimization algorithm, only a stochastic estimate of the loss is accessible, so we cannot derive an algorithm based on an expectation of the loss. So isn't there a mismatch between the derivation and the actual oracles we have on the function.
>
> > The specific approximation is described immediately above the line you cite,
>
> Could you give the whole details of the derivations? This does not have to be in this rebuttal but for the last version of the paper. I am sorry but from the equations above line 234 to the equation below line 234 there are no detailed derivations.
>
> > Similar approximations are used elsewhere
>
> Detailing each step will strengthen the paper everywhere.

---

> ### Comment · Reviewer_WGtm · 2025-08-09
>
> (continued)
>
> > It was not done around a steady state, but with respect to a local loss landscape approximation.
>
> I was referencing appendix F.1. To me steady state meant equilibrium, i.e., for t sufficiently large. You indeed say "At steady state (or at least, on time scales somewhat longer than τ )". In optimization, an approximation is usually done around the current parameters (for per-iteration complexity). Approximations around the solution (so what the author means by steady state? please correct me if I misunderstood) lead to local convergence rates. It seems to me peculiar to consider being close to a solution (both in usual convex optimization, but also in deep learning).
>
> > Your statement seems to be wrong, or at the very least an extreme exaggeration.
>
> Ok for the exaggeration. However, I would sincerely encourage the author to give detailed derivations (like the ones for Adam).
>
>
> **Other notes**
>
> - One argument to not make experiments has been that the algorithms are well known. In that case, a complete and detailed derivation of the final algorithm obtained by the framework is necessary. As the authors noted, these details could actually lead to slightly different algorithms. And so the experiments may actually also be slightly different. As a simple example, the "Newton" method proposed by the authors is not the usual Newton method. Note that it may also be costly to implement, require a linesearch etc... So currently the framework appears to provide ideal (i.e. independent of iteration cost) versions of some algorithms.
>
> - That said, the proposed framework can also rather lead to new relevant diagnostic tools when running experiments. Namely plotting the proposed objectives along time (as some form of lyapunov function) could be great (even on toy problems).

---

> > ### Comment · Reviewer_WGtm · 2025-08-09
> >
> > I acknowledge that this review led to a somewhat heated exchange that was unfortunate.
> >
> > I hope that the author will consider this exchange to add details in the paper, more discussion with related work, and relevant experiments.
> >
> > I fully acknowledge that I am not the right audience for this paper, while it appears that other reviewers like the paper. A fresh perspective on optimization is needed. This one may be wrong, and experiments (even simple ones like the snippet I proposed) would have tremendously helped. I cannot recommend this paper for acceptance, but I shall not block it. So I will increase my score to 3.

---

### Official Review · Reviewer_BRpX · 2025-07-12

**Clarity:** 1
**Significance:** 1
**Originality:** 2
**Rating:** 3
**Confidence:** 2

**Summary:**

In this work, the authors develop a framework for describing existing learning algorithms. Across a variety of algorithms, they work towards identifying rules for which each particular algorithm is the optimal algorithm.

**Questions:**

how do the results relate to practice? are there experiments that would made this paper more relevant to practice?

**Ethical Concerns:**

["NO or VERY MINOR ethics concerns only"]

**Limitations:**

yes

**Quality:**

2

**Strengths And Weaknesses:**

Strengths:

* the authors study the important problem of characterizing learning algorithms
* normative rules for learning algorithms is an interesting way to characterize learning algorithms

Weaknesses:

* no experiments for identifying whether or not the identified rules are actually realistic in practice (as a basic example: do continuous time approximations actually match what happens in practice? one would think a priori that infinitely many extraordinarily small learning rates would actually significantly change navigation of the loss landscape for DL optimization tasks).
* the paper is rather confusing, treating prior work as well known and obvious (e.g., the amari insight on line 141) and considered theory as obvious (e.g., line 118 or line 80). ideally, the authors would work to guide the reader better through their work.

---

> ### Author Rebuttal · Authors · 2025-07-27
>
> We thank the reviewer for their time and feedback, and think that it has improved the paper. We would first like to clarify a minor but conceptually important point.
>
> **Our theory does not require infinitesimally small learning rates.**
>
> > (as a basic example: do continuous time approximations actually match what happens in practice? one would think a priori that infinitely many extraordinarily small learning rates would actually significantly change navigation of the loss landscape for DL optimization tasks)
>
> The learning rules we derive in this paper, including well-known rules like gradient descent with momentum, and Adam, do not require infinitesimally small parameter updates. See, for example, our derivation of gradient descent with momentum (Sec. 2; see especially Eq. 7). In the final learning rules, the step size $\\Delta t$ can be arbitrarily large.
>
> (As a caveat, the precise *form* of the learning rule definitely depends on the step size. For example, in Eq. 7, small step sizes look more like gradient descent, and larger step sizes look more like Newton's method.)
>
> We agree that, if our theory proposed only learning rules with extraordinarily small learning rates, it would not be practical. However, it does not propose this. Rather, the continuous-time formulation should be mostly viewed as a trick for getting concrete math results (since it enables the use of the Euler-Lagrange equations, our central tool). In the final learning rules, updates do not have to be infinitesimally small. For concrete examples with finite $\Delta t$, see (for example) Eq. 7, Eq. 10, and Eq. 14.
>
> **On guiding the reader through our results.**
>
> > the paper is rather confusing, treating prior work as well known and obvious (e.g., the amari insight on line 141) and considered theory as obvious (e.g., line 118 or line 80). ideally, the authors would work to guide the reader better through their work.
>
> This is a fair point, and we could do a better job clarifying our approach throughout the paper. While we do think certain concepts may be familiar to some readers---for example, the continuous-time objectives we discuss on lines 118 and 80 may be familiar to readers familiar with control theory---we acknowledge that this will not be true for all readers. We have already tried to add context where possible (for example, when we discuss Amari's insight around line 141, we also go on to describe that insight using the terminology of this paper), but we will add more context to further improve clarity.
>
> To address this concern, we plan to improve the final version of the paper in two ways. First, we can introduce some of the math and ideas a bit more gently, and add additional context where possible. It is certainly true that some sections are quite dense in the current version. Second, we will add more figures and examples to provide readers another way to understand what is going on. We think these two changes will greatly improve the clarity of the paper.
>
> **On practical consequences of our framework.**
>
> > how do the results relate to practice? are there experiments that would made this paper more relevant to practice?
>
> This is a good question. First, our main aim with this paper is to provide a framework for unifying and better understanding existing learning rules. This aim is admittedly more modest than trying to derive new state-of-the-art learning rules, but we think we have achieved it reasonably well. Other recent machine learning work also tries to do this (see, for example, "I-Con: A Unifying Framework for Representation Learning" by Alshammari et al., which we cite).
>
> However, it is definitely true that one can be more confident in a framework like ours if it derives novel learning rules that perform well on nontrivial machine learning tasks. We have derived a few novel learning rules (for example, in Eq. 7 we mention a novel 'ballistic' learning rule which involves the square root of the Hessian), but have not seriously evaluated their performance in the current version of the paper.
>
> We are happy to perform new experiments to test the novel learning rules suggested by our framework, and think these experiments would improve the quality of the paper and more clearly demonstrate the power of the framework. We are open to suggestions for what experiments may be most interesting, but we think that since the main goal of the paper is to establish the framework and rederive known learning rules, we will try to keep things simple. We will show some simple toy examples (e.g., we will compare different learning rules for a convex, quadratic loss) and slightly more complex examples related to image classification (e.g., on simple data sets like MNIST and CIFAR-10). These are the kinds of examples used in a variety of benchmarking papers, for example to study the performance of Adam.
>
> We have not yet run these experiments, partly since the rebuttal period does not allow explicitly showing new results, but do not expect they will be difficult to perform.

---

### Decision · Program_Chairs · 2025-09-17

**Decision:**

Accept (poster)

**Comment:**

I have the following concerns:

(1) The main contribution is a unifying framework, but I am not convinced of its significance. For a fixed set of phenomena (or, a fixed set of learning rules), one can find infinitely many ways to unify them. The theory space is much larger than the space of reality. To actually propose a unifying framework, I think the authors are obliged to predict/design/do something new with it. But the only thing the authors did is to explain existing ones. This also relates to some reviewers' criticism that it lacks practical relevance and experiments

(2) The paper does not achieve what it suggests to achieve. A key question raised by the authors is "Why do some learning rules work better than others?" But this question is unanswered, and I do not feel a hint of answer after reading the paper. I think those who support the acceptance of this paper should explain to me using authors' logic the answer to this question. This point also links to two other problems: (1) lack of significance and (2) the paper is poorly written (in terms of overclaiming)

(3) Novelty. I find the paper of limited novelty. As other reviewers have also pointed out, the Lagrangian formalism has been proposed by Wibisono et al. The derivation of the second-order rules and Natural GD is very straightforward and involves no novel technique (given the standard derivations of them and other related works pointed out by reviewers). Not to mention that these rules are not really used. I think the primary novelty is about the partial observability, but this part is way too short and shallow. I would happier if this section is 5 page long with 2 experiments on transformers revealing novel phenomena about them (exaggerated but you get my point).

Also, I partially disagree with the authors' reply to KkuV. The authors claim that a disadvantage of Wibisono et al is that "that their objective involves a minus sign (i.e., it has the form kinetic minus potential)." This is not a valid criticism. First of all, the Lagrangian in physics has kinetic minus potential, thus this criticism applies to physics. Thus, the same logic implies that the authors' formalism is also a better theory for physics. This cannot be true. Second, if we transform the Lagrangian to a Hamiltonian, the Hamiltonian indeed has kinetic plus potential (usually). Thus, the problem is trivially avoided by well known results. In fact, the equivalent Bregman Hamiltonian has indeed been pointed out by Wibisono et al.